# Disparate macrophage responses are linked to infection outcome of Hantan virus in humans or rodents

Hongwei Ma[1,2,6], Yongheng Yang[1,6], Tiejian Nie[3,6], Rong Yan[1], Yue Si[1], Jing Wei[1,4], Mengyun Li[1], He Liu[1], Wei Ye [1], Hui Zhang[1], Linfeng Cheng[1], Liang Zhang[1], Xin Lv[1], Limin Luo[5], Zhikai Xu[1] ✉, Xijing Zhang[2] ✉, Yingfeng Lei[1] ✉ & Fanglin Zhang[1] ✉

Hantaan virus (HTNV) is asymptomatically carried by rodents, yet causes lethal hemorrhagic fever with renal syndrome in humans, the underlying mechanisms of which remain to be elucidated. Here, we show that differential macrophage responses may determine disparate infection outcomes. In mice, late-phase inactivation of inflammatory macrophage prevents cytokine storm syndrome that usually occurs in HTNV-infected patients. This is attained by elaborate crosstalk between Notch and NF-κB pathways. Mechanistically, Notch receptors activated by HTNV enhance NF-κB signaling by recruiting IKKβ and p65, promoting inflammatory macrophage polarization in both species. However, in mice rather than humans, Notch-mediated inflammation is timely restrained by a series of murine-specific long noncoding RNAs transcribed by the Notch pathway in a negative feedback manner. Among them, the lnc-ip65 detaches p65 from the Notch receptor and inhibits p65 phosphorylation, rewiring macrophages from the pro-inflammation to the pro-resolution phenotype. Genetic ablation of lnc-ip65 leads to destructive HTNV infection in mice. Thus, our findings reveal an immune-braking function of murine noncoding RNAs, offering a special therapeutic strategy for HTNV infection.

Hantaviruses are zoonotic pathogens distributed worldwide and have drawn public concern with the newly reported possibility of super spread[1]. They encompass at least 58 distinct genotypes classified in the genus *Orthohantavirus*, among which the Old World and New World lineages lead to two diseases in humans, namely, hemorrhagic fever with renal syndrome (HFRS) prevailing in Eurasia and hantavirus pulmonary syndrome (HPS) in the Americas, respectively[2,3]. Hantaan virus (HTNV), the prototype hantavirus naturally hosted by striped field

mice (*A. agrarius*), is transmitted to humans by the contaminated secreta or excreta, causing severe HFRS with a case fatality rate of 15%[4,5]. HTNV virions contain three negative single-stranded RNA genome segments encoding the nucleocapsid protein (NP), glycoprotein precursor, and viral RNA-dependent RNA polymerase, respectively[6]. Previous studies have demonstrated that cytokine storm syndrome contributes to the pathogenesis of HFRS[7–10], while only slight immunopathologic injury can be detected in natural reservoirs[11–13]. Less is

[1]Department of Microbiology & Pathogen Biology, School of Basic Medical Sciences, Air Force Medical University (the Fourth Military Medical University), Xi'an, Shaanxi 710032, China. [2]Department of Anaesthesiology & Critical Care Medicine, Xijing Hospital, Air Force Medical University (the Fourth Military Medical University), Xi'an, Shaanxi 710032, China. [3]Department of Experimental Surgery, Tangdu Hospital, Air Force Medical University (the Fourth Military Medical University), Xi'an, Shaanxi 710038, China. [4]Shaanxi Provincial Centre for Disease Control and Prevention, Xi'an, Shaanxi 710054, China. [5]Department of Infectious Disease, Air Force Hospital of Southern Theatre Command, Guangzhou, Guangdong 510602, China. [6]These authors contributed equally: Hongwei Ma, Yongheng Yang, Tiejian Nie. ✉e-mail: zhikaixu@fmmu.edu.cn; zhangxj918@163.com; yflei@fmmu.edu.cn; flzhang@fmmu.edu.cn

known about the mechanisms determining the magnitude of host anti-hantaviral responses.

Macrophages and their precursor monocytes belong to the host mononuclear phagocyte system (MPS), constituting the first defense line against microbial infection. Emerging evidence indicates that macrophages maintain high plasticity and heterogeneity, serving as a rheostat for immune actions[14]. Macrophage polarization is regulated by various cytokines or surveillance receptors[15–17]. Stimulation by Th1 cytokines or pathogen-associated molecular patterns strengthens the classical inflammatory status of macrophage (M1) through the Stat1 or NF-κB pathway. In contrast, Th2 cytokines or glucocorticoids can convert macrophages to an alternative pro-resolution state (M2) by activating Stat3 or GATA3. Similarly, the monocyte subsets are classified into classic, inflammatory (M1-like) or patrolling (M2-like) pattern[18,19]. Perturbing the MPS state transition contributes to the pathogenesis of multiple infectious diseases[20,21]. Recent studies reported that human monocytes and macrophages might be a determinant of hantavirus pathogenicity[22–24], while their role in rodents remains ambiguous.

Notch signaling is an evolutionarily conserved development pathway in vertebrates, and recent researches indicate it exerts pleiotropic action in immune regulation[25,26]. Mammalian Notch ligands include Delta-like and Jagged family members that interact with Notch receptors and promote their cleavage by γ-secretase, releasing the Notch intracellular domain (NICD). NICD translocates into the nucleus and binds to transcription factor (TF) RBP-J, enabling the expression of HES or HEY family members[27]. The dual effects of Notch signaling on macrophage polarization have been reported with a complex but elaborate mechanism in inflammatory diseases[28–31]. The Notch and TLR pathways cooperated synergistically to reinforce M1 activation by enhancing the synthesis and IRF8 and promoting the production of TNFα, IL-6, and IL-12[29,30]. On the other hand, the Notch pathway was indispensable for the expression of a series of M2 genes in chitin- or lymphocyte-derived DNA stimulation models[28,29,31]. Several studies have discovered that activation of the Notch pathway in monocytes or macrophages participates in the pathogenesis of acute viral infection, including the Dengue virus (DENV) and influenza A virus (IAV) infection[32,33], but the exact mechanisms have been largely underexplored.

NF-κB is a family of TFs that influence a broad range of physiological and pathological processes[34]. In the resting state, p65 is bound and sequestered by the inhibitor of NF-κB (IκB) in the cytoplasm. Upon infection or stress conditions, IκB proteins are phosphorylated by the IκB kinase (IKK) complex and undergo subsequent degradation, which releases p65 and potentiates its phosphorylation[35]. Phosphorylated p65 translocates into the nucleus and induces the expression of inflammatory cytokines, and aberrant p65 activation is highly involved in the cytokine storm syndrome during viral or bacterial sepsis[36,37]. The role of NF-κB signaling seems to be controversial during hantaviral infection. HTNV infection could trigger TLR4-dependent and p65-mediated inflammatory responses, which are responsible for endothelial dysfunction and viral pathogenicity[38–42]. Nevertheless, it has been reported that the NP of HTNV could interact with the karyopherin importin α, blocking the nuclear translocation of inflammatory TFs[43–45]. It is opaque whether there exists a disparate modulatory pattern of NF-κB pathway between different species.

Currently, increasing evidence suggests that long noncoding RNAs (lncRNAs) are critical immune regulators by acting as the modification switcher (e.g., lnc-DC), location guider (e.g., lincRNA-Cox2), or aggregation scaffolder (e.g., NEAT1)[46]. Lnc-DC directly binds to and prevents Stat3 dephosphorylation by SHP1, facilitating human dendritic cell differentiation[47]. LincRNA-Cox2 enhances the occupancy of RNA polymerase II on the *Il6* promoter to facilitate IL-6 production[48]. NEAT1 recruits transcriptional suppression proteins to form paraspeckle, strengthening the expression of pattern recognition receptors (PRRs) or cytokines[11,49]. It is worth noting that lncRNAs possess relatively low sequence conservation across species. Several lncRNAs are exclusively transcribed in rodents instead of humans, among which there are lnc-lsm3b and lnczc3h7a that regulate the RIG-I-mediated antiviral responses[50,51]; additionally, another batch of human-specific immune gene-priming lncRNAs (IPLs) have been recently identified, which could facilitate the H3K4me3 epigenetic priming of chemokine genes[52].

In this study, we demonstrate that disparate macrophage responses determine the outcome of HTNV infection in humans or rodents. The M1-triggered cytokine storm contributes to HFRS pathogenesis in human patients, whereas the late-phase inactivation of M1 curbs inflammation in rodents. Furthermore, we find that the murine macrophage phenotype is reprogrammed by HTNV through the Notch-lncRNA-p65 axis. HTNV infection promotes NICD release, which assist the p65-mediated M1 polarization. Whereafter, a cluster of Notch-downstream murine-specific lncRNAs, including lncRNA 30740.1 (termed lnc-ip65), could efficiently restrain M1 activation. Loss- and gain-of-function assays show that lnc-ip65 targets p65 and inhibits its phosphorylation at S276, S529 and S536. Ablation of lnc-ip65 led to lethal HTNV infection in rodents. Taken together, our study sheds light on how HTNV elicits discriminative immune responses between rodents and humans, offering potential therapeutic strategies against HFRS and other inflammatory diseases.

## Results

### Hyperactivation of inflammatory monocytes contributes to cytokine storm syndrome in HFRS patients

The monocyte subsets displayed a unique alteration pattern in HFRS patients. Individuals with HTNV infection possessed a higher proportion of M1-like monocytes (CD14++ CD16+) but a relatively lower M2-like percent (CD14+ CD16++) than those with Japanese encephalitis virus (JEV), hepatitis B or C virus (HBV or HCV) infection (Fig. 1a, Supplementary Fig. 1a). The monocytes were rapidly mobilized at the acute stage of HFRS (Fig. 1b, c, Supplementary Fig. 1b). Stratification analysis showed that it was not the M1-like subsets across the whole clinical stages, but the proportion of them at the acute stage, that positively correlated with the disease severity (Fig. 1d, e). No relationship between M2-like monocytes with patient conditions was found (Supplementary Fig. 1c). Additionally, the percentage of regulatory T (Treg) cells, but not Th1, Th2, or Th17, correlated with disease severity (Supplementary Fig. 1d, e), which coincided with previous studies[10,53]. Based on the Bio-Plex system, we identified twenty upregulated cytokines in the severe/critical patients compared with those of mild/moderate or healthy individuals, among which there were TNFα, IL-8 and IL-10 (Fig. 1f). Then we wondered whether the monocyte subsets were associated with the patient immunophenotype. Intriguingly, the M1-like monocytes in HFRS patients were characterized by high expression of TNFα, IL-8, IL-10 and HLA-DR (Supplementary Fig. 2a), suggesting that they might lead to a compounded inflammatory response. Moreover, it was the TNFα+ monocytes that correlated with HFRS severity (Fig. 1g, Supplementary Fig. 2b, c). The TNFα+ IL-10- monocytes, as well as TNFα+ IL-10+ and TNFα- IL-10+ monocytes, showed persistent activation in the severe/critical group (Fig. 1h). These data indicated that excessive inflammation in HFRS might be incriminated with the dysregulated monocyte patterns.

### Late-phase inactivation of inflammatory macrophages occurs in rodents rather than in humans

Considering undue inflammatory responses drive HFRS progression, we hypothesized whether this pathological process is discrepant in rodents. *A. agrarius* mice were captured in the Weihe Plain of China, where HTNV was prevailing (Supplementary Table 1). HTNV maintained a high replication level in rodent lungs (Supplementary Fig. 3a), which was consistent with previous studies[54,55]. Based on the viral RNA

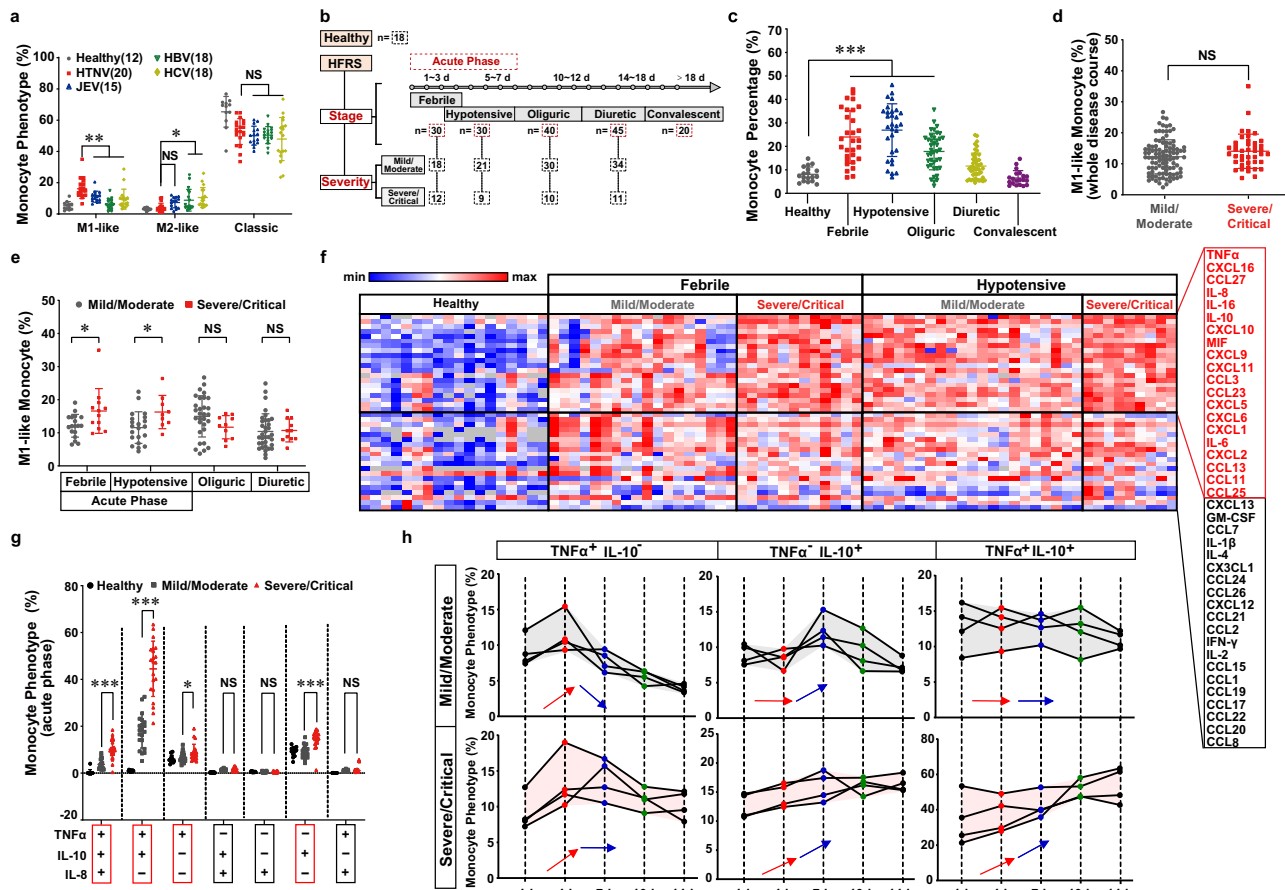

**Fig. 1 | Inflammatory monocytes trigger TNFα-centered cytokine storm in HFRS patients. a** The monocyte phenotype measured by flow cytometry from PBMCs of patients with different virus infections (acute phase for HTNV infection/ $n = 20$, febrile phase for JEV infection/ $n = 15$, and chronic phase for HBV/ $n = 18$, or HCV infection/ $n = 18$) or healthy individuals ($n = 12$). HTNV group as control (vs. JEV, HBV or HCV), for M1-like monocytes, $p = 0.0156/< 0.0001/= 0.0006$; for M2-like monocytes, $p = 0.1917/0.0213/0.001$; for classic monocytes, $p = 0.7341/0.7878/ 0.2571$. Gating strategy is shown in Supplementary Fig. 1a. **b** Overview of disease stages and severity classification for HFRS. The patient sample number of each disease course with different severity for (**c**–**g**) has been listed. **c** The monocyte (CD11b⁺CD11c⁺) percentage measured by flow cytometry in HFRS patients. Healthy group as control, $p < 0.001$ (vs. febrile stage)/$< 0.001$ (vs. hypotensive stage)/ $= 0.0002$ (vs. oliguric stage). Gating strategy is shown in Supplementary Fig. 1b. The relationship between the proportion of M1-like monocytes (measured by flow cytometry) and HFRS severity (Mild/Moderate vs. Severe/Critical) is analyzed either across the whole disease stage (**d**, $p = 0.0787$) or at distinct disease phases

(**e**, $p = 0.0237$, at febrile stage/0.0218, at hypotensive stage/0.1251 at oliguric stage/ 0.8627 at diuretic stage). Gating strategy is shown in Supplementary Fig. 1a. **f** Heatmap of the 40-multiplex array. Upregulated cytokines are marked red ($p < 0.05$). The exact $p$ value of each group is shown in the Source Data File. **g** The secretion of TNFα, IL-10 and IL-8 in monocytes measured by flow cytometry at the acute stage of HFRS. Mild/Moderate vs. Severe/Critical, $p < 0.0001$ (TNFα⁺ IL-10⁺ IL-8⁺)/<0.0001 (TNFα⁺ IL-10⁻ IL-8⁻)/= 0.0402 (TNFα⁺ IL-10⁻ IL-8⁻)/= 0.1711 (TNFα⁻ IL-10⁺ IL-8⁺)/= 0.8487 (TNFα⁻ IL-10⁻ IL-8⁻)/< 0.0001 (TNFα⁻ IL-10⁺ IL-8⁺)/= 0.6775 (TNFα⁺ IL-10⁻ IL-8⁺) of different monocyte phenotypes. Gating strategy is shown in Supplementary Fig. 2b, c. **h** Dynamic alteration of HFRS monocyte subsets detected by flow cytometry along with disease progression. Gating strategy is shown in Supplementary Fig. 2b, c. Data are shown as the mean ± SD. Analysis is performed using one-way ANOVA (**a**, **c**, Dunnett's multiple comparisons test), or two-sided unpaired Student's $t$ test (**d**–**g**). *$p < 0.05$, **$p < 0.01$, ***$p < 0.001$; NS no significance. Source data are provided as a Source Data file.

level and host antibody production, the rodent disease phases were classified as HTNV infection negative stage (HINS), early stage (HIES), progressive stage (HIPS), and clearance stage (HICS) (Fig. 2a). Six inflammatory cytokines, especially TNFα and IP-10, elevated at the HIES and then declined at the HIPS (Fig. 2b), indicating that the murine immune system was transiently activated but timely controlled post HTNV infection. To note, murine alveolar macrophages (AMs), originally scattered in the lung tissue at the HINS, were recruited to alveolar capillaries and distributed surrounding the HTNV-infected endothelial cells (Supplementary Fig. 3b). To evaluate the macrophage polarization state, NF-κB and JAK/STAT pathways in AMs were detected. Intriguingly, although the total expression level of p65 increased, its phosphorylation level peaked at the HIES but collapsed overtly at the HIPS (Fig. 2c). Contemporaneously, the phosphorylation of Stat1 was maintained at a relatively high level at both the HIES and HIPS (Fig. 2c). These findings implied that HTNV might dynamically

manipulate murine macrophage reprogramming via NF-κB signaling in vivo.

Although the M1 activation in *A. agrarius* mice was elaborately controlled, it remained uncertain whether this process was beneficial for hosts against hantaviral infection. To answer this question, clodronate liposomes (clophosome) were applied to eliminate monocytes and macrophages in vivo[56]. In terms of the lethal infection model of neonatal mice by HTNV[57], treatment with clophosome or the TNFα neutralizing antibody at the early stage (namely 1 day post infection/ dpi), but not the progressive stage (namely 5 dpi), could effectively improve the disease outcome (Supplementary Fig. 4a, b). In terms of the asymptomatic infection model of adult mice by HTNV[11], we found that pre-treatment of clophosome promoted the onset of disease (Supplementary Fig. 4c), in which mice showed weakened TNFα production with high viral loads (Supplementary Fig. 4d). Similar results were also observed in the RIG-I⁻/⁻ mouse model (Supplementary

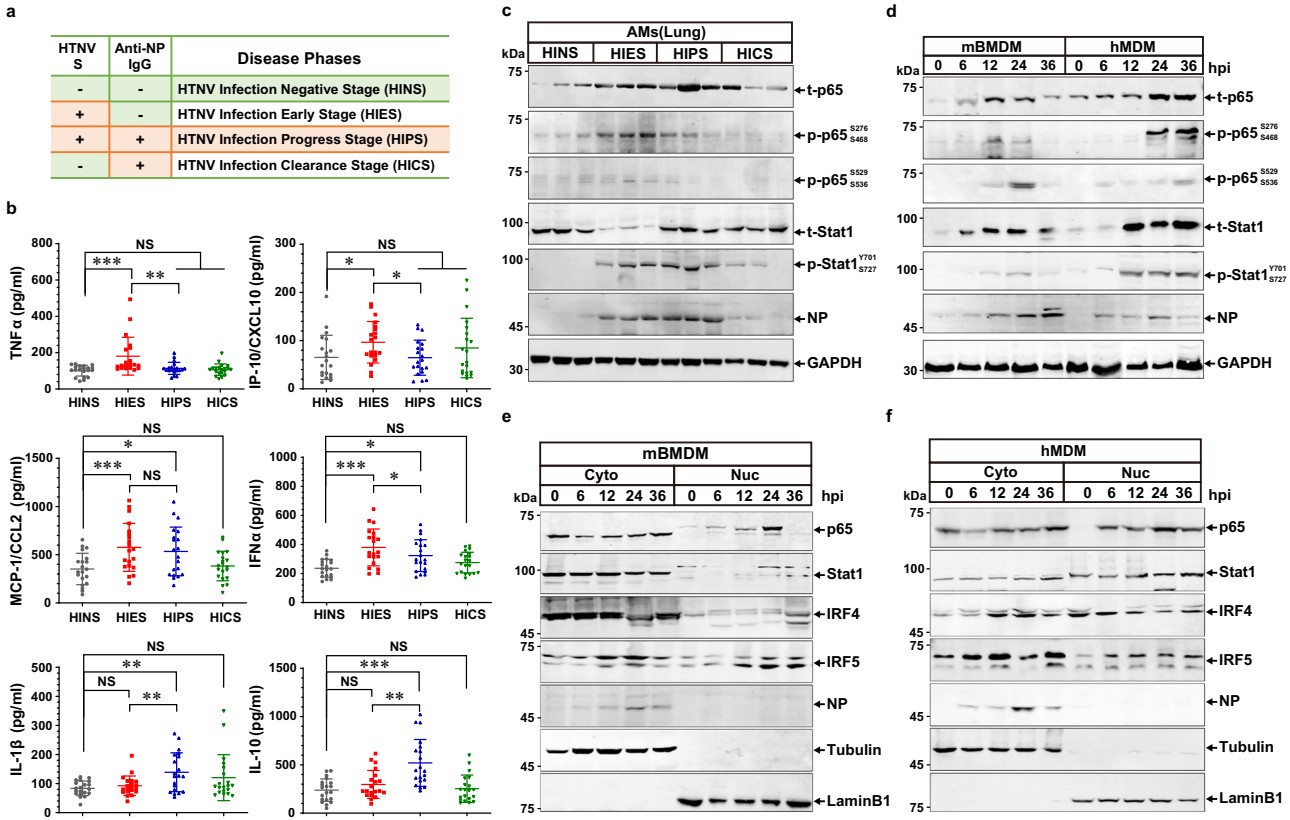

**Fig. 2 | Differential immune status in rodents versus humans might be determined by distinct macrophage responses against HTNV Infection. a** Natural infection phases of HTNV in *A. agrarius* mice defined based on the assessment of HTNV-S and anti-NP IgG in the lung tissue. **b** Cytokine production measured by ELISA in lung tissues ($n = 20$ in each group). HIES as control (vs. HIPS), $p = 0.0092$ (TNFα)/0.003 (IP-10)/0.5973 (MCP-1)/0.0275 (IFNα)/0.0084 (IL-1β)/0.0011 (IL-10). HINS as control (vs. HIES, HIPS or HICS), $p = 0.0002/0.8384/0.9844$ (TNFα); $p = 0.0409/> 0.9999/= 0.4072$ (IP-10); $p = 0.0032/ 0.0203/0.937$ (MCP-1); $p < 0.0001/= 0.0145/= 0.4273$ (IFNα); $p = 0.9233/0.0069/0.1034$ (IL-1β); $p = 0.5576/< 0.0001/= 0.9844$ (IL-10). **c** Representative immunoblot analysis of three independent experiments for p65 and Stat1 in AMs from field mice with

various disease stages. **d** Representative immunoblot analysis of three independent experiments for p65 and Stat1 in mBMDM or hMDM from 0 to 36 hpi with an MOI of 1. (**e** and **f**) Representative immunoblot analysis of three independent experiments for TFs located in the cytoplasm or nucleus from mBMDM (E) or hMDM (F) at an MOI of 1 for HTNV infection. Data are shown as the mean ± SD for animal samples. Data are representative of three independent experiments. Analysis of different groups is performed with two-sided unpaired Student's *t* test, or one-way ANOVA (Dunnett's multiple comparisons test). *$p < 0.05$, **$p < 0.01$, ***$p < 0.001$; NS no significance. Molecular weight markers are shown to the left of the blots in kDa, and antibodies used are indicated to the right. Source data are provided as a Source Data file.

---

Fig. 4c, d). In brief, monocytes/macrophages might act as destroyers in the high-lethal neonatal mouse model but as defenders in the non- or low-lethal adult mouse model against HTNV infection.

To address the specific role of MPS, primary monocytes or macrophages from mice or humans were extracted and subjected to HTNV infection in vitro. TNFα generated from the murine bone marrow-derived macrophages (mBMDM) and peritoneal macrophages (mPMφ) increased from 0 to 24 h post infection (hpi) and then decreased, the production of which showed continuous elevation in the human monocytes (hMo) or monocyte-derived macrophages (hMDM) (Supplementary Fig. 4e), revealing a discrepant proinflammatory identity of MPS in different species. On the other hand, murine and human macrophages exhibited analogous antiviral functions (Supplementary Fig. 4f). As for the inflammatory pathway alteration, phosphorylated p65 (S276, S468, S529, and S536) increased from 0 to 24 hpi and then decreased in mBMDM, the activation pattern of which was different from that in hMDM (Fig. 2d). Consistently, the amount of p65 in the nucleus (Fig. 2e, f), as well as its DNA binding capacity (Supplementary Fig. 4g), also displayed a fluctuating trend in mBMDM rather than hMDM. Nevertheless, the activation schema of some other pivotal TFs, which mediated the polarization of M1 (e.g., Stat1 and IRF5) or M2 (e.g., IRF4), did not differ between mBMDM and hMDM (Fig. 2e, f).

To confirm the distinct reprogramming process by HTNV, further experiments were performed based on macrophage cell lines. The TNFα or IFNα production pattern in murine RAW264.7 and MH-S cells, or human THP-1-derived cells, was identical to the primary cells (Supplementary Fig. 5a). The p65 transcription activity during HTNV infection, detected by the dual-luciferase reporter assays (Supplementary Fig. 5b, c) or the NF-κB-DNA binding assays (Supplementary Fig. 5d), also displayed a late-phase inactivation model in RAW264.7 rather than THP-1-derived cells. To directly assess the p65 activation status, the phosphorylation and subcellular localization of p65 were assessed. Phosphorylated p65 increased from 0 to 24 hpi and then decreased in RAW264.7 cells, which persistently accrued in THP-1-derived cells (Supplementary Fig. 5e). Based on the live cell imaging system, we found that the amount of p65 in the nucleus was significantly reduced in RAW264.7 from 24 to 35 hpi, differing from that in the THP-1-derived cells (Supplementary Fig. 5f). All these data verified the late-phase inactivation of p65 by HTNV in murine rather than human macrophages.

### Macrophage reprogramming by HTNV confers rodents with resistance against secondary sepsis

A significant association of secondary bacterial sepsis with hantavirus disease severity has been reported recently[58–61]. Considering the late-

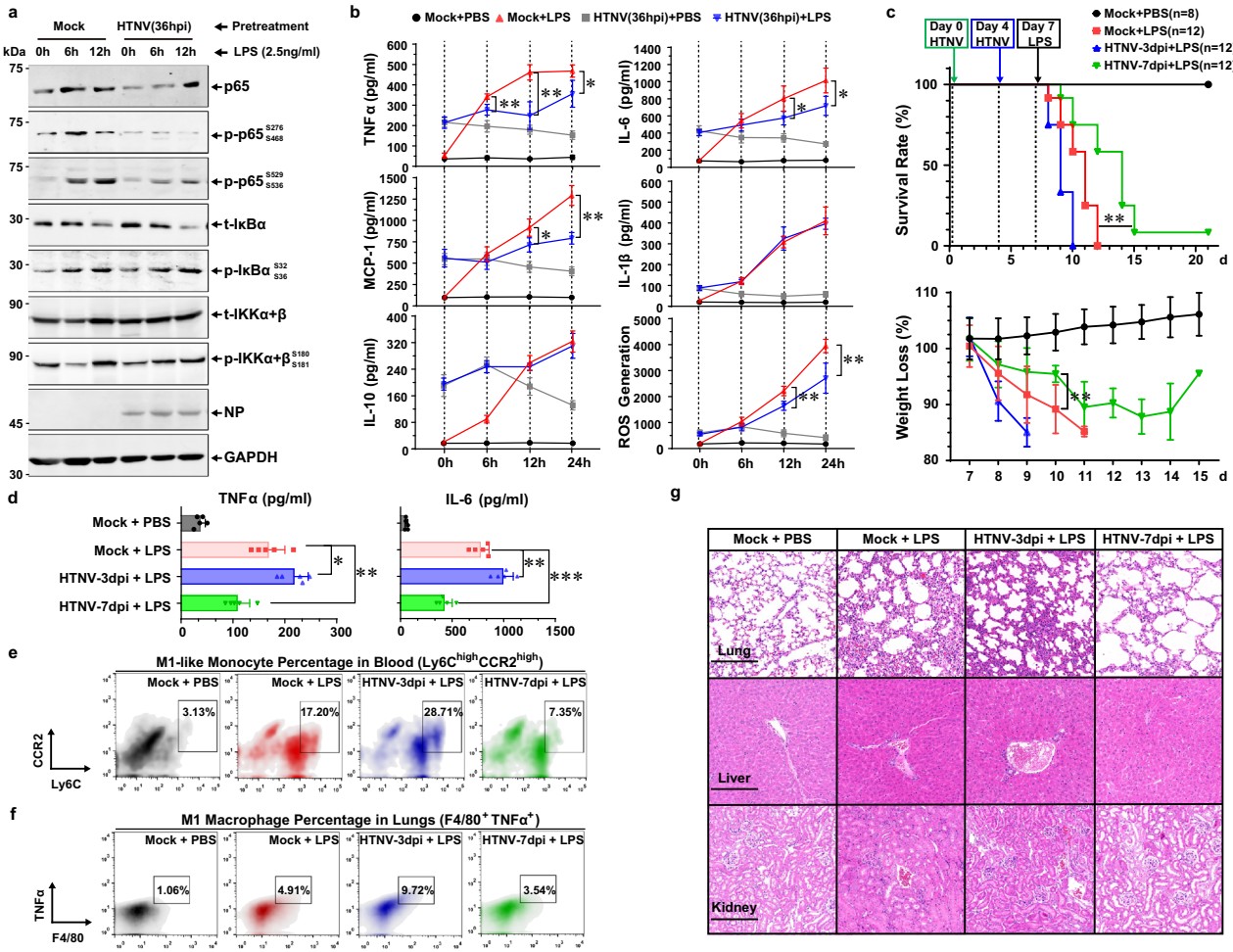

**Fig. 3 | Late-phase Inactivation of Rodent M1 by HTNV Alleviates Immunopathogenesis Caused by Secondary Bacterial Sepsis. a, b** The mBMDM are pre-infected with HTNV (MOI = 1) for 36 h and then stimulated with LPS (2.5 ng/ml). Cell lysates or supernatants at 6 h or 12 h post LPS stimulation are collected for p65 pathway (**a**, representative immunoblot image) or cytokine (**b**, $n = 4$, ELISA) detection, respectively. Mock+LPS vs. HTNV (36pi) + LPS in (**b**), $p = 0.005$ (6 h)/ 0.0015 (12 h)/0.0217(24 h) for TNFα; $p = 0.0364$ (12 h)/0.0176 (24 h) for IL-6; $p = 0.0213$ (12 h)/0.0003 (24 h) for MCP-1; $p = 0.0039$ (12 h)/0.008 (24 h) for ROS. **c** Survival and weight loss data of mice with initial HTNV infection (intramuscular injection/i.m., $8 \times 10^5$ $TCID_{50}$/g) and subsequent LPS challenge (intraperitoneal injection/i.p., 5 mg/kg). For PBS challenge, $n = 8$; for LPS challenge, $n = 12$ in each group. Mock + LPS vs. HTNV-7dpi + LPS, $p = 0.003$ for survival data, $p = 0.0011$ for weigh loss data at 10 d. **d** Serum TNFα or IL-6 concentration measured by ELISA from (**c**) at one day post LPS challenge ($n = 5$). Mock + LPS as control, $p = 0.0281$ (vs.

HTNV-3dpi+LPS)/0.0091 (vs. HTNV-7dpi+LPS) for TNFα, $p = 0.0047$ (vs. HTNV-3dpi +LPS)/0.0001 (vs. HTNV-7dpi+LPS) for IL-6. Representative flow cytometric images for M1-like monocytes (Ly6C⁺ CCR2⁺) (**e**) and M1 AMs (F4/80⁺ TNFα⁺) (**f**) from (**c**) at one day post-LPS challenge. Related statistical analysis is shown in Supplementary Fig. 6d, e, respectively. **g** Representative haematoxylin and eosin (H&E) staining images of mouse tissues from (**c**) at one day post-LPS challenge. Scale bars, 200 μm. Data are shown as the mean ± SEM. Data are representative of three independent experiments. Analysis is performed using two-sided unpaired Student's *t* test (**b** and weight loss comparison in **c**), one-way ANOVA (**c**, Dunnett's multiple comparisons test), or survival curve comparison (survival comparison in **c**, log-rank [Mantel−Cox] test). *$p < 0.05$, **$p < 0.01$, ***$p < 0.001$; NS no significance. Molecular weight markers are shown to the left of the blots in kDa, and antibodies used are indicated to the right. Source data are provided as a Source Data file.

phase inactivation of M1 in rodents, we wondered whether this process could protect hosts against bacterial sepsis. To address this question, a sequential challenge model was established both in vitro and in vivo. Pre-infected with HTNV for 36 h could hinder the LPS-induced p65 phosphorylation, during which the activation of IKBα and IKKα/β were not affected (Fig. 3a). The LPS-induced production of proinflammatory cytokines (e.g., TNFα and IL-6), chemokines (e.g., MCP-1), and antimicrobial ROS, but not the IL-1β and IL-10, was suppressed in the HTNV-36 hpi group (Fig. 3b). However, the p65 activation and cytokine generation post LPS stimulation were strengthened in the HTNV-12 hpi group (Supplementary Fig. 6a, b). These findings suggested that the late-phase inactivation of M1 by HTNV could prohibit the LPS-triggered inflammation, possibly through a p65-dependent manner.

Next, we investigated whether HTNV could affect LPS-induced Gram-negative sepsis in vivo. To define the HTNV infection phase in vivo, the dynamics of viral proteins and cytokines were measured. We found that HTNV NP and TNFα production maintained at comparatively high levels at 3 dpi and then declined remarkably at 7 dpi (Supplementary Fig. 6c). The mice at the late-infection phase (7 dpi), but not the early phase (3 dpi), were protected from the subsequent LPS challenge (Fig. 3c). The LPS-stimulated serum TNFα and IL-6 production (Fig. 3d), as well as the hyperactivation of M1-like monocytes in peripheral blood (Fig. 3e, Supplementary Fig. 6d), M1-type AMs in lungs (Fig. 3f, Supplementary Fig. 6e), and tissue immunopathology (Fig. 3g), was attenuated in the HTNV-7 dpi group but aggravated in the HTNV-3 dpi group. These results indicated that HTNV infection might

alter mouse susceptibility to LPS-induced sepsis. Furthermore, the cecal slurry (CS)-induced polymicrobial sepsis model was established post HTNV infection. We found that late-phase infection (7 dpi), or the clophosome treatment, could defend mice against lethal CS challenge (Supplementary Fig. 7a). Compared with the mock group, the pathological injury of lung tissues (Supplementary Fig. 7b), and the generation of proinflammatory cytokines in mouse serum (Supplementary Fig. 7c) were distinctly improved either in the HTNV-7dpi or clophosome group. No synergic protective effects of HTNV infection and clophosome application were found against secondary polymicrobial sepsis (Supplementary Fig. 7a–c). Additionally, pre-infected HTNV suppressed CS-induced M1 activation by inhibiting the expression of TNFα, IL-6, IL-1β, and Nos2 (Supplementary Fig. 7d), and enhanced the expression of M2-related genes such as Arg-1, Chil3, and Retnla (Supplementary Fig. 7e). These findings signified that late-phase inactivation of inflammatory macrophages by HTNV might defend rodents against lethal polymicrobial toxicity.

### Notch signaling blocks p65 activation and rewires the murine macrophage phenotype at the late HTNV infection stage

The RNA-seq data confirmed the late-phase inactivation of inflammatory macrophages (M1) and the reactivation of the pro-resolution phenotype (M2) (Fig. 4a–i). GO and KEGG analysis indicated that the PRR and Notch signaling changed significantly and displayed a nonlinear alteration pattern with infection (Supplementary Fig. 8a, b). Further RNA interfering and knockout experiments revealed that it was the Notch pathway, but not the TLR, RIG-I, or type I IFN signaling, that controlled the late-phase downregulation of TNFα (Supplementary Fig. 8c-i, ii). To note, RBP-J knockout in macrophages contributed to continuous TNFα production, while NICD overexpression could not affect TNFα release (Supplementary Fig. 8c-iii). Then there was one possibility that RBP-J and downstream genes probably launched negative feedback against NICD-mediated M1 polarization. To verify our hypothesis, the dominant negative form of RBP-J (R218H)[62] and the γ-secretase inhibitor (DAPT/GSI-IX)[30] were used. The overexpression of R218H, which competitively bonded with NICD and blocked endogenous RBP-J activation, remarkably reinforced TNFα production (Supplementary Fig. 8c-iii). The application of DAPT before infection, but not at the late infection stage, suppressed TNFα production (Supplementary Fig. 8c-iv). These results suggested that murine Notch signaling might dynamically rewire macrophage phenotype.

To decipher the specific Notch activation pattern, the production and subcellular localization of Notch proteins were measured. During the natural HTNV infection process, Notch-related genes in mBMDM were upregulated along with infection, while at the late infection stage (from 24 to 48 hpi), the M1- but not M2-related genes showed a descending trend (Supplementary Fig. 9a). Of note, though the total expression of Notch proteins increased (Supplementary Fig. 9b), the NICD firstly accumulated in the cytoplasm (0 to 24 hpi) and then translocated into the nucleus (24 to 48 hpi) (Supplementary Fig. 9c, d). This activation pattern was also validated in vivo, which meant that NICD was stockpiled in the cytoplasm of AMs, Kupffer cells (KCs), kidney or spleen macrophages at 3 dpi and shifted to the nucleus at 7 dpi (Supplementary Fig. 10). These data indicated that HTNV might trigger early incomplete (without target gene expression) but later complete (with target gene activation) Notch signaling in murine macrophages.

The next question was how murine Notch signaling modulates the late-phase passivation of M1. RNA-seq results showed that most M1-related genes were upregulated at the late infection stage in RBP-J[CKO] mBMDM (Fig. 4a-ii), which was further confirmed by qRT-PCR (Supplementary Fig. 11a). RBP-J[CKO] mBMDM showed robust proinflammatory and antigen-presenting function at the late HTNV infection phase, for that the production of TNFα, IL-6 and IL-12, as

well as the expression of CD80 and CD86, was remarkably strengthened at 36 hpi (Fig. 4b, Supplementary Fig. 11b). RBP-J[CKO] mBMDM also displayed enhanced phagocytosis, chemotaxis and anti-microbial function at 36 hpi (Supplementary Fig. 11c-e). Moreover, RBP-J[CKO] mBMDM maintained an increased extracellular acidification rate (ECAR, an indicator for M1-related glycolysis[63]) but a decreased oxygen consumption rate (OCR, an indicator for M2-related mitochondrial respiration[64]) than the WT group at the late phase (Supplementary Fig. 11f). Previous studies showed that the Notch pathway affected the mitochondrial function[65], and here, we also found that RBP-J[CKO] mBMDM possessed a large number of damaged mitochondria (Supplementary Fig. 11g), which might partially explain why Notch could manipulate the metabolic reprogramming of macrophages. These data suggested that the Notch pathway determined the late-phase immunophenotype switch of macrophages.

Considering that the NF-κB pathway contributed to the mitochondria quality control[66] and the late-phase inactivation of M1 (Fig. 2), we wondered whether Notch-mediated macrophage reprogramming was p65-dependent. The increased phosphorylation of p65, but not Stat1, was found in the RBP-J[CKO] mBMDM from 36 to 48 hpi (Fig. 4c). Consistently, the real-time live-cell imaging system confirmed sustained nucleus aggregation of p65 in the RBP-J[CKO] mBMDM from 24 to 36 hpi, during which p65 was shifted to the cytoplasm in the WT group (Fig. 4d). These results substantiated that the murine Notch pathway might inhibit M1 polarization at the late infection stage by turning off NF-κB signaling in vivo. To evaluate whether this process was beneficial in vivo, neonatal and adult mouse models were utilized. RBP-J[CKO] suckling mice showed an early onset of disease than the WT mice (Fig. 4e), which was associated with more severe inflammatory responses but not viral loads (Supplementary Fig. 11h). Intriguingly, RBP-J[CKO] adult mice were susceptible to high-dose HTNV challenge (Fig. 4f, g), and they also showed aggravated TNFα responses and uncontrolled hantaviral replication at 7 dpi (Supplementary Fig. 11i). Additionally, worsen pathological changes in the RBP-J[CKO] spleens at 7 dpi, which were accompanied by hyperactivation of p65 (Fig. 4h, i). The reinforced activation of signaling triggered via p65, c-Jun N-terminal kinase (JNK), c-Jun N-terminal kinase (ERK) or IRF5 was also found in the RBP-J[CKO] spleens at 7 dpi (Fig. 4j). These in vivo models suggested that the RBP-J-mediated late-phase inactivation of murine inflammatory macrophages played a protective role against HTNV infection.

### Notch activation pattern differs in human macrophages and promotes M1-mediated inflammation

Another noteworthy question was whether there existed a discrepant Notch activation pattern in human macrophages. Most Notch-related genes and proteins increased post HTNV infection in hMDM from 0 to 36 hpi (Fig. 5a, b), during which the NICD was continuously generated and translocated into the nucleus (Fig. 5c, d). These results indicated that the Notch pathway was completely activated in human macrophages throughout the infection stage. Hindering NICD generation with DAPT could constrain the phosphorylation of p65 rather than p-JNK or p-ERK, which would also facilitate HTNV replication at the late infection stage (36–48 hpi) (Fig. 5e), suggesting that Notch signaling might consolidate the human M1 polarization process. Consistently, DAPT restrained the secretion of various proinflammatory cytokines at 48 hpi (Fig. 5f), during which the expression of manifold M1-related genes was downregulated while M2-related genes were strengthened (Fig. 5g, h). Furthermore, we found that the activation level of Notch signaling in monocytes was associated with disease severity (Supplementary Fig. 12). These data collectively demonstrated that Notch signaling showed a distinct activation pattern in humans versus mice, which promoted an M1-mediated cytokine storm in HFRS patients.

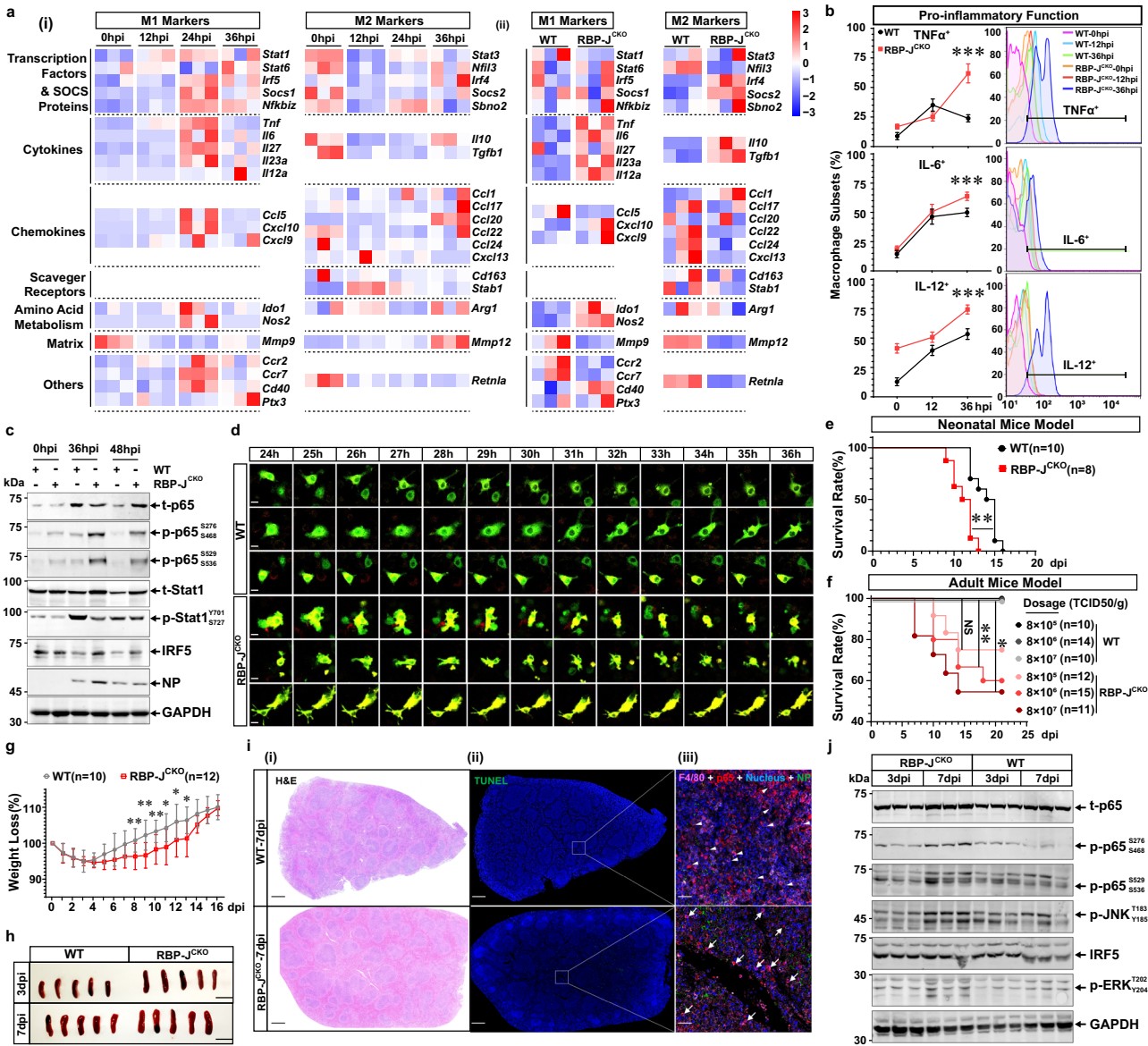

**Fig. 4 | Murine Notch Signaling Contributes to the Late-phase Inactivation of M1 by Inhibiting p65 Activation. a** Heatmap showing major genes associated with mBMDM polarization based on the RNA-seq data (MOI = 1, $n = 3$). Various time points post HTNV infection (i); RBP-J$^{CKO}$ vs. WT mBMDM at 36 hpi (ii). **b** Proinflammatory cytokine secretion detected by flow cytometry at 36 hpi (MOI = 1, $n = 4$). Statistical analysis is shown on the left, and gating strategy is shown on the right. WT vs. RBP-J$^{CKO}$ at 36 hpi, $p = 0.0003$ (TNFα)/ < 0.0001 (IL-6)/ < 0.0001 (IL-12). **c** Representative immunoblot analysis of three independent experiments evaluating the activation of M1-related TFs at the late infection stage (MOI = 1). **d** Representative live cell imaging of three independent experiments depicting the translocation of GFP-p65 in the cytoplasm or nucleus from the WT and RBP-J$^{CKO}$ mBMDM ($n = 3$). Scale bars, 10 μm. **e** Survival data for HTNV-challenged 4-day neonatal mice (i.p., $8 \times 10^5$ TCID$_{50}$/g) ($n = 10$ for WT, $n = 8$ for RBP-J$^{CKO}$). $P = 0.0012$. **f** Survival data for HTNV-challenged 8-week-old adult mice doses (i.p., $8 \times 10^5$ to $8 \times 10^7$ TCID$_{50}$/g). The mice number of each group is shown in the figure symbols. WT vs. RBP-J$^{CKO}$, $p = 0.0979$ ($8 \times 10^5$ TCID$_{50}$/g)/0.009 ($8 \times 10^6$ TCID$_{50}$/g)/0.0172

($8 \times 10^7$ TCID$_{50}$/g). **g** Weight loss analysis of (**f**) (i.p., $8 \times 10^5$ TCID$_{50}$/g). WT ($n = 10$) vs. RBP-J$^{CKO}$ ($n = 12$), $p = 0.0071/0.0028/0.0057/0.0123/0.0286/0.0205$ for 8–13 dpi, respectively. **h** Morphological alteration of spleens at 3 dpi or 7 dpi (i.p., $8 \times 10^5$ TCID$_{50}$/g, $n = 5$). **i** Representative H&E (scale bars, 500 μm) (-i), TUNEL (scale bars, 500 μm) (-ii) and immunofluorescent staining (scale bars, 20 μm) (-iii) of spleen tissues from (**h**). Triangles mark the cytoplasmic location of p65, and arrows show the nuclear location of p65 in F4/80$^+$ macrophages. **j** Representative immunoblot analysis of three independent experiments evaluating the activation of p65, JNK, ERK and IRF5 from (**h**) ($n = 3$). Animal data are shown as the mean ± SD, and the cell data as the mean ± SEM. Results are representative of three independent experiments. Analysis is performed using two-sided unpaired Student's $t$ test (**b**, **g**), or survival curve comparison (**e**, **f**, log-rank [Mantel–Cox] test). *$p < 0.05$, **$p < 0.01$, ***$p < 0.001$; NS no significance. Molecular weight markers are shown to the left of the blots in kDa, and antibodies used are indicated to the right. Source data are provided as a Source Data file.

## Murine-specific LncRNAs downstream of notch signaling retrain M1 polarization

It is ambiguous why Notch signaling regulates macrophage polarization differently in mice versus humans. Considering that this pathway is highly conserved, we wondered whether there existed some other transcripts controlled by Notch. The RNA-seq analysis identified ninety-seven murine-specific lncRNAs in HTNV-infected RBP-J$^{CKO}$ mBMDM (Fig. 6a and Supplementary Data 1), most of which maintained potential protein binding capacity according to the RBPDB database[67] (Supplementary Data 2). Thirty-one lncRNAs were confirmed through qRT-PCR, among which eight lncRNAs (namely, 22387.1, 30740.1, 30928.1, 60100.1, 59654.1, 57001.1 and 11443.1)

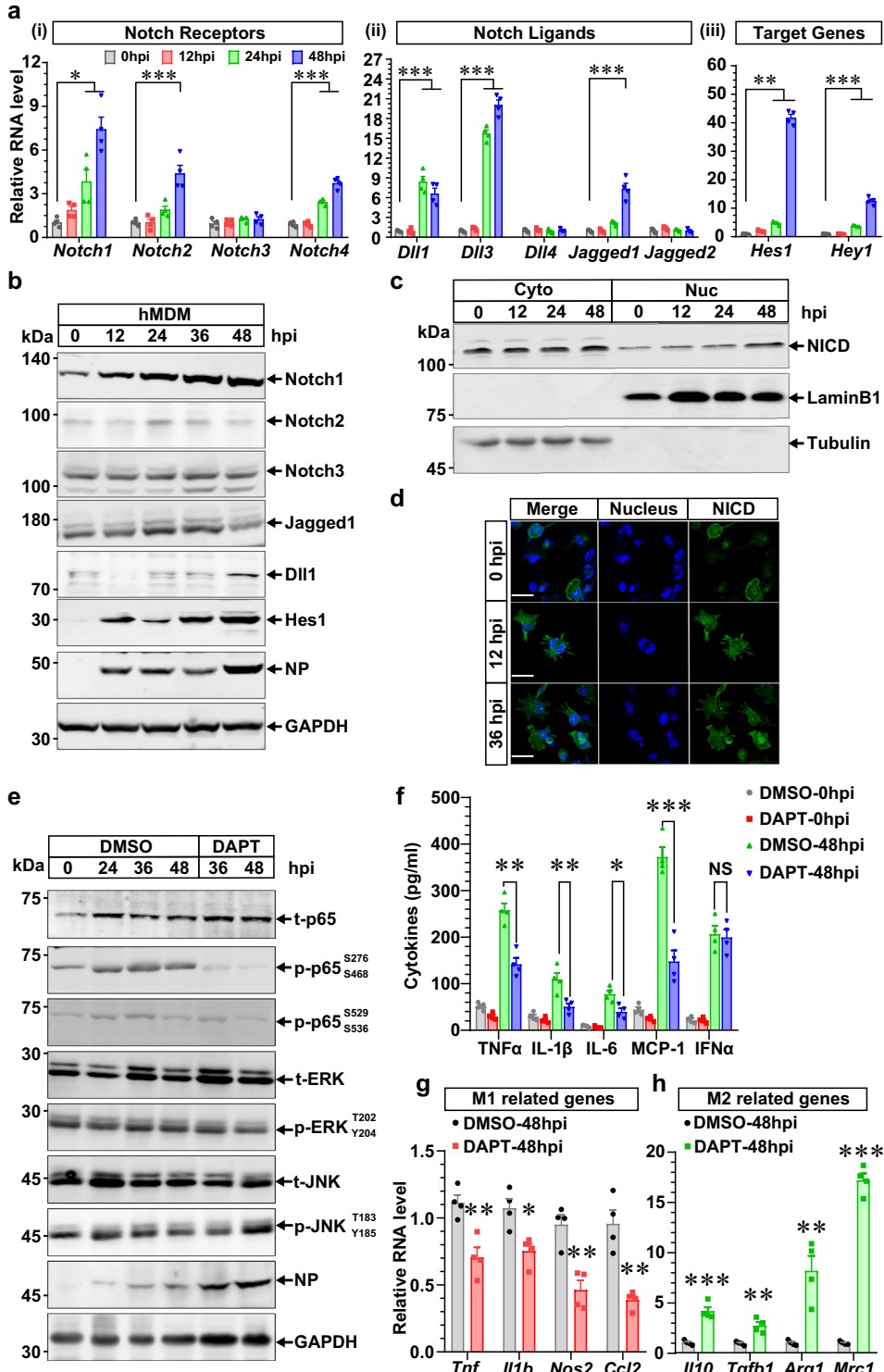

showed a fold change of more than two at the late phase (36 to 72 hpi) of HTNV or Dengue virus 2 (DENV2) infection (Fig. 6b, Supplementary Fig. 13a). Silencing 22387.1, 30740.1 and 30928.1 conspicuously consolidated TNFα production (Fig. 6c-(i), Supplementary Fig. 13b, c), possibly by enhancing the NF-κB pathway activation (Fig. 6c-(ii), Supplementary Fig. 13d). These data suggested that these three lncRNAs might act as negative feedback for M1 polarization.

Then, the biological features of these lncRNAs were analyzed. Sequence-based bioinformatic analysis[68] showed that they had low coding capability (Supplementary Fig. 14a), and conservation analysis

based on the UCSC Genome Browser database[69] indicated that they were murine-specific (Supplementary Fig. 14b). The tissue expression evaluation suggested that they were transcribed endogenously in diversified tissues (Supplementary Fig. 14c). The fluorescence in situ hybridization assay (FISH) showed that 30740.1 and 30928.1.1 were mostly distributed in the cytoplasm, while 22387.1 was located both in the cytoplasm and nucleus (Supplementary Fig. 14d). There existed RBP-J-binding DNA sequences among the upstream region of their transcription site (Supplementary Fig. 15a). Inhibiting the Notch pathway via DAPT would hinder these lncRNAs' expression, while

**Fig. 5 | Notch pathway activates differently in human macrophages and promotes M1-mediated HFRS pathogenesis. a** qRT-PCR analysis of Notch pathway-related genes in HTNV-infected hMDM (MOI = 1, n = 4). *Notch1*, p = 0.0166 (0 vs. 24 hpi)/< 0.001(0 vs. 48 hpi); *Notch2*, p < 0.0001 (0 vs. 48 hpi); *Notch4*, p < 0.0001 (0 vs. 24 or 48 hpi); *Dll1*, p < 0.0001 (0 vs. 24 or 48 hpi); *Dll3*, p < 0.001 (0 vs. 24 or 48 hpi); *Jagged1*, p < 0.0001 (0 vs. 48 hpi); *Hes1*, p = 0.0035(0 vs. 24 hpi)/< 0.0001 (0 vs. 48 hpi); *Hey1*, p = 0.0004 (0 vs. 24 hpi)/< 0.001 (0 vs. 24 hpi).
**b** Representative immunoblot analysis of three independent experiments for the Notch pathway in HTNV-infected hMDM (MOI = 1). **c** Representative immunoblot analysis of three independent experiments for the NICD in the cytoplasm or nucleus in HTNV-infected hMDM (MOI = 1). **d** Representative immunofluorescent analysis of three independent experiments for NICD localization in HTNV-infected hMDM (MOI = 1). **e** Representative immunoblot analysis of three independent

experiments for phosphorylated p65, ERK or JNK in HTNV-infected hMDM that are pretreated with DMSO or DAPT (50 μmol/L) for 12 h. **f** Supernatant cytokine concentration detected by ELISA from (**e**). DMSO vs. DAPT (48 hpi), p = 0.0012 (TNFα)/0.0099 (IL-1β)/0.0117 (IL-6)/0.0004 (MCP-1)/0.7738 (IFNα). qRT-PCR analysis of M1- (**g**) or M2-related (**h**) gene expression in hMDM from (**e**). In (**g**), p = 0.0048 (*Tnf*)/ 0.0112 (*Il1b*)/0.0028 (*Nos2*)/0.0019 (*Ccl2*); In (**h**), p = 0.0003 (*Il10*)/0.0028 (*Tgfb*)/0.003 (*Agr1*)/< 0.0001 (*Mrc1*). Data are shown as the mean ± SEM, and are representative of three independent experiments. Analysis is performed using one-way ANOVA (**a**, Dunnett's multiple comparisons test), or two-sided unpaired Student's t test (**f–h**). *p < 0.05, **p < 0.01, ***p < 0.001; NS no significance. Molecular weight markers are shown to the left of the blots in kDa, and antibodies used are indicated to the right. Source data are provided as a Source Data file.

motivating Notch pathway by adding mDll1 would enhance their transcription (Supplementary Fig. 15b). RBP-J knockout blocked lncRNA expression, while replenishing RBP-J, instead of R218H, rescued this process (Fig. 6d-(i)). These data confirmed that the three lncRNAs were regulated by the Notch pathway. Moreover, silencing TLR3 and TLR4, but not RIG-I and MDA, inhibited these lncRNAs transcription at 36 hpi (Fig.6d-(ii)). Other RNA viruses such as Sendai virus (SeV), vesicular stomatitis virus (VSV) and enterovirus 71 (EV71), but not DNA viruses such as herpes simplex virus type 2 (HSV-2), could propel the expression of these lncRNAs (Fig. 6e). The lncRNA expression was also correlated with viral MOIs (Supplementary Fig. 15c), and could be activated by TLR3 (polyIC) or TLR4 agonist (LPS) (Supplementary Fig. 15d). Taken together, these murine lncRNAs might be involved in multiple RNA virus infection processes.

To fully investigate the role of these lncRNAs, locked nucleic acids (LNAs) were applied to intervene in their expression (Supplementary Fig. 15e). In the LNA-NC (negative control) group, the macrophage phenotype showed a switch from the pro-inflammation to the pro-resolution pattern at 36 hpi, while silencing 22387.1, 30740.1 or 30928.1 could significantly hinder such transition (Fig. 6f). The ablation of lncRNAs promoted the proinflammatory and antiviral capacity by upregulating the CCR7⁺ IL-6⁺ (Fig. 6g) and reducing the CCR2⁺ CX3CR1⁺ macrophage proportion (Fig. 6h). Notably, silencing these lncRNAs impaired the chemotactic ability, but improved the phagocytosis and antigen-presenting function of macrophages (Supplementary Fig. 15f–h). Furthermore, the expression of CD206 (M2 marker) was decreased in the knockdown group (Supplementary Fig. 15i), and the metabolic process was also converted to the M1-related glycolysis type in the LNA-interfering group (Supplementary Fig. 15j). Similar results were acquired using siRNAs (Supplementary Fig. 15k–m). On the other hand, compensating these lncRNAs might partially offset the pro-M1 effects (Supplementary Fig. 15n), verifying the negative feedback launched by these lncRNAs. In brief, the cluster of RBP-J-targeted lncRNAs facilitated macrophage transformation from a pro-inflammatory to a pro-resolutory phenotype at the late HTNV infection phase.

## Lnc-ip65 obstructs M1 polarization by interacting with and inhibiting p65 phosphorylation

To investigate how murine-specific lncRNAs restrain M1 activation, the gain of and loss of function experiments were performed. Reinforced expression of these lncRNAs could separately or simultaneously repress p65 or Stat1 phosphorylation at 24 hpi, while the ablation of them enhanced p65 and Stat1 activation at 36 hpi (Supplementary Fig. 16a). Notably, intervening in lncRNA expression could not affect the phosphorylation of IKKα/β and IκBα (Supplementary Fig. 16a), indicating that they might directly regulate p65 activity. Based on RNA-binding protein immunoprecipitation (RIP) experiments, we found that lncRNA 30740.7 and 30928.1 could interact with p65 under either overexpressing or natural infection conditions (Fig. 7a, b). Considering that the fold change of 30740.1 is more substantial than 22387.1 or

30928.1 against viral infection (Fig. 6e), we mainly focused on the function of 30740.1(termed lnc-ip65).

RNAScope experiments suggested that lnc-ip65 colocalized with p65 at the resting status (0 hpi) or late HTNV infection stage (36 hpi) in mBMDM, during which nearly no p65 in the nucleus could be detected; however, at the early phase (24 hpi), the lnc-ip65 expression was decreased, during which p65 was accumulated in the nucleus (Supplementary Fig. 16b). The interaction between lnc-ip65 and p65 has also been confirmed in RAW264.7 cells post HTNV/DENV infection (Fig. 7c) or polyIC/LPS stimulation, with different FISH probes targeting lnc-i65 (Supplementary Fig. 16c). Ablation of lnc-ip65 promoted the p65 phosphorylation at 36 hpi, principally at S276, S529 and S536 (but not S468) (Fig. 7d). Similar results were detected with the live-cell imaging system (Fig. 7e). To investigate the exact interaction region of p65 with lnc-ip65, different mutants of p65 were constructed according to the potential RNA-binding domain (Supplementary Fig. 16d). The 1–549, 1–300, 401–549 and 401–500 amino acid (aa) segments of p65, but not the 1–260 and 301–400 aa segments, could interact with lnc-ip65 as measured by RIP (Fig. 7f), and the interaction relationship was further verified by RNAScope experiments (Fig. 7g). The results suggested that lnc-ip65 was possibly absorbed to the region adjacent to phosphorylation points (S276, S529 and S536), which would interfere with their phosphorylation process through conformational hindrance. To validate whether this steric effect makes an impact, competitive experiments were implemented by exogenously expressing p65 (401–500 aa). As expected, p65 (401–500 aa) could recruit and remove the negative effects of lnc-ip65, strengthening endogenous p65 phosphorylation and its translocation into the nucleus (Supplementary Figs. 16e, 17f).

To unearth the functional region of lnc-ip65, the secondary structure and relative thermodynamic free energy were analyzed with RNAfold[70], and different truncated segments were designed and constructed based on the structural stability (Supplementary Fig. 17a). We found that the middle part of lnc-ip65 (1001–2000 nucleotides/nt) could notably hinder p65 phosphorylation at S529 and S536, and the head region of lnc-ip65 (1–1000 nt) seemed to exert better inhibitory effects on S276 phosphorylation, both of which could not affect the T254 and S311 phosphorylation of p65 or the activation of IκBα at 24 hpi (Fig. 7h). RNAScope proved that it was the head or middle region of lnc-ip65 that interacted with p65 and restrained its translocation into the nucleus in HTNV-infected macrophages (Fig. 7i). Functionally, exogenous expression of the head or middle region of lnc-ip65 weakened TNFα but strengthened IL-10 mRNA transcription (Fig. 7j, Supplementary Fig. 17b). The lncRNA-protein interaction propensity was computed with catRAPID omics[71], and the results predicted that the head and middle part of lnc-ip65 might bind to p65 (Supplementary Fig. 17c). RIP results confirmed that the head and middle region of lnc-ip65 bound to the p65 (1–300 aa) and p65 (401-500 aa), respectively (Fig. 7k), proving the hypothesis that lnc-ip65 was attached to the nearby serine area and exerted steric effects.

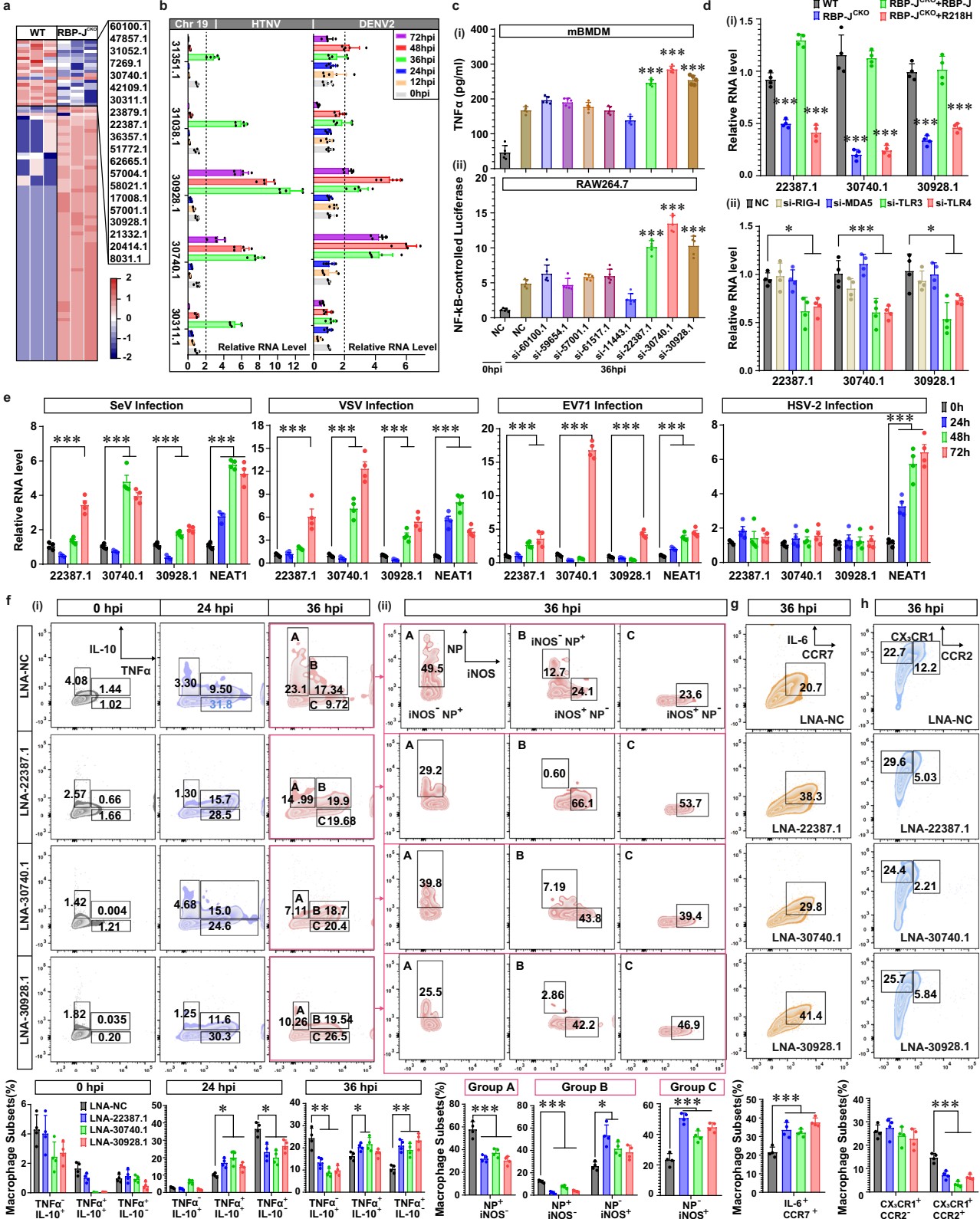

## Lnc-ip65 deficiency aggravates systemic inflammation and sensitizes mice to HTNV infection

To further elucidate the protective role of lnc-ip65 against HTNV infection, lnc-ip65$^{-/-}$ mice were generated with the CRISPR/Cas9 technology (Supplementary Fig. 18a). There were no evident physiological or behavioral differences for neonatal or adult lnc-ip65$^{-/-}$ mice compared with their WT littermates, while the transgenic mice showed a shortened lifespan (Supplementary Fig. 18b). For the neonatal challenge model, the disease course in the lnc-ip65$^{-/-}$ group was characterized by early onset and prompt death (Supplementary Fig. 18c). For the adult challenge model, lnc-ip65$^{-/-}$ mice were susceptible to HTNV infection, as they showed high fatality and severe weight loss than WT mice, which could be partially rescued through anti-TNFα antibody treatment (Fig. 8a). Continuously high concentrations of serum TNFα and IL-6 at the early

**Fig. 6 | Murine notch pathway prevents M1 hyperactivation by inducing inhibitory LncRNAs. a** Heatmap of murine-specific lncRNAs that were differentially expressed in RBP-J$^{CKO}$ compared with WT mBMDM at 36 hpi ($n = 3$). False discovery rate (FDR) < 0.01. **b** qRT-PCR analysis of the indicated lncRNAs on chromosomal 19 (Chr19) in HTNV- or DENV2-infected mBMDM from 0 to 72 hpi (MOI = 1, $n = 5$). **c** ELISA analysis of TNFα secretion from mBMDM (-i) and luciferase detection of NF-κB activity in RAW264.7 cells (-ii) at 36 hpi (MOI = 1, $n = 5$). In (-i) or (-ii), NC-36 hpi as control, $p < 0.0001$ (vs. si-22387.1, si-30740.1 or si-30928.1). **d** qRT-PCR analysis of indicated lncRNAs in HTNV-infected mBMDM (MOI = 1, at 36 hpi) either overexpressing RBP-J/R218H (-i), or underwent RNAi experiments for 24 h (-ii). In (-i), WT as control, $p < 0.0001$ (vs. RBP-J$^{CKO}$ or RBP-J$^{CKO}$ + 218H) for lncRNA 22387.1, 30740.1 or 30928.1; In (-ii), NC as control, $p < 0.0001$ (vs. si-TLR3)/ = 0.0101 (vs. si-TLR4) for lncRNA 22387.1, $p = 0.0006$ (vs. si-TLR3)/0.0005 (vs. si-TLR4) for lncRNA 30740.1, $p = 0.0003$ (vs. si-TLR3)/ = 0.0194 (vs. si-TLR4) for lncRNA 30928.1. **e** qRT-PCR analysis of the indicated lncRNAs in mBMDM post various virus infections (MOI = 0.1). SeV infection, $p < 0.0001$ (0 vs. 72 h) for lncRNA 22387.1, $p < 0.0001$ (0 vs. 48 or 72 h) for lncRNA 30740.1, 30928.1 or NEAT1; VSV infection, $p < 0.0001$ (0 vs. 72 h) for lncRNA 22387.1, $p < 0.0001$ (0 vs. 48 or 72 h) for lncRNA 30740.1, $p = 0.0003$ (0 vs. 48 h)/ < 0.0001 (0 vs. 72 h) for lncRNA 30928.1, $p < 0.0001$

(0 vs. 24 or 48 h)/ = 0.0003 (0 vs. 72 h) for lncRNA NEAT1; EV71 infection, $p = 0.0054$ (0 vs. 48 h)/0.0002 (0 vs. 72 h) for lncRNA 22387.1, $p < 0.0001$ (0 vs. 72 h) for lncRNA 30740.1 or 30928.1, $p < 0.0001$ (0 vs. 48 or 72 h) for lncRNA NEAT1; HSV-2 infection, $p = 0.0003$ (0 vs. 24 h)/ < 0.0001(0 vs. 48 or 72 h) for lncRNA NEAT1. **(f to h)** Flow cytometry analysis of the indicated markers in NICD$^{STOP-floxed}$ mBMDM that underwent RNAi experiments and subsequent HTNV infection (MOI = 1, $n = 4$). Statistical analysis is shown at the bottom line. LNA-NC as control vs. LNA-22387.1, 30740.1 or 300.0117 for TNFα$^+$ IL-10$^-$; in the 36 hpi group of **(f)**-(i), $p = 0.0005/<0.0001/<0.0001$ for TNFα$^-$ IL-10$^+$, $p = 0.0484/ 0.0115/ 0.6998$ for TNFα$^+$ IL-10$^+$, $p = 0.0004/ 0.002/ 0.0117/ < 0.0001$ for TNFα$^+$ IL-10$^-$; in group A of **(f)**-(ii), $p < 0.0001$ for different group comparison; in group B of **(f)**-(ii), $p < 0.0001/ = 0.0003/<0.0001$ for NP$^+$ iNOS$^-$, $p = 0.0003/0.0172/0.0581$ for NP$^-$ iNOS$^+$; in group C of **f**-(ii), $p < 0.0001$ for different group comparison; in **(g)**, $p = 0.0002/ 0.0006/ < 0.0001$; in **(h)**, $p < 0.0001$ for different group comparison of CX$_3$CR1$^+$ CCR2$^+$ group. Data are shown as the mean ± SEM, and are representative of three independent experiments. Analysis is performed using one-way ANOVA (Dunnett's multiple comparisons test). *$p < 0.05$, **$p < 0.01$, ***$p < 0.001$. Molecular weight markers are shown to the left of the blots in kDa, and antibodies used are indicated to the right. Source data are provided as a Source Data file.

---

infection course, as well as low IL-10, were detected in lnc-ip65$^{-/-}$ mice (Fig. 8b), suggesting that excessive inflammatory responses occurred and might contribute to the lethal pathogenesis.

Then, the host systemic inflammatory injuries of transgenic mice post HTNV infection were evaluated. Proinflammatory cytokine production was significantly consolidated in lung tissues from the lnc-ip65$^{-/-}$ mice (Supplementary Fig. 18d), and massive immunocyte infiltration and deteriorated apoptosis were detected (Fig. 8c-(i), c-(ii)). The lnc-ip65$^{-/-}$ AMs showed more NICD production, iNOS expression, p65 and Stat1 in the nucleus by immunofluorescent assays (Fig. 8c-(iii), c-(iv)). The phosphorylation of p65 and Stat1 was reinforced in lnc-ip65-deficient AMs by immunoblot analysis (Fig. 8d). Intriguingly, although the AMs of lnc-ip65$^{-/-}$ mice displayed enhanced inflammatory and antiviral phenotypes, HTNV replication was not limited in alveolar epithelial and interstitial cells (Fig. 8c-(v)). Similar results were detected in lnc-ip65$^{-/-}$ livers (Fig. 8c, d, Supplementary Fig. 18d). In the lnc-ip65$^{-/-}$ spleens, proinflammatory cytokine production was slightly increased at 6 dpi (Supplementary Fig. 18d), while the pathological section indicated a prominent white pulp reduction and tissue apoptosis (Supplementary Fig. 19a-(i), a-(ii)). Likewise, knocking out lnc-ip65 forced M1 macrophage polarization and restricted viral replication in spleens, in which M2 macrophage activation was largely blocked (Fig. 8e, Supplementary Fig. 19a, b). Augmented inflammatory responses were also found in murine kidneys (Supplementary Figs. 18d, 19a), while the alterations in hearts or brains seemed to be insubstantial (Supplementary Figs. 18d, 20). The overall inflammation score evaluation in various organs suggested that there were more serious immunopathological alterations for alterations in the lung, liver and spleen in lnc-ip65$^{-/-}$ mice than in WT mice at 6 dpi (Supplementary Fig. 19c).

Additionally, murine heat and mechanical hypersensitivity, highly related to the TNFα-induced inflammation, were measured at different time points post HTNV challenge. We found that the responsive latency or threshold was decreased in the lnc-ip65$^{-/-}$ mice (Fig. 8f), which hinted that lnc-ip65 knockout might aggravate host inflammation. Finally, classical sepsis models were established, and we found that lnc-ip65$^{-/-}$ mice were susceptible to LPS or CS challenge (Fig. 8g), which indicated that deteriorated inflammation occurred in lnc-ip65$^{-/-}$ mice. Taken together, the in vivo data indicated that lnc-ip65 played a critically protective role in maintaining host immune homeostasis against HTNV infection.

### NICD activates NF-κB signaling by recruiting IKKβ and p65, which is blocked by Lnc-ip65

Since murine Notch signaling was initially activated upon HTNV infection in both murine and human macrophages, we were curious

about the role of NICD in macrophage polarization. Previous studies have shown complicated crosstalk between the Notch and NF-κB pathways[72] (Supplementary Fig. 21a), and here, we found that NICD directly bound to p65 and IKKβ, but not IκBα through coimmunoprecipitation experiments (Supplementary Fig. 21b). This interaction process could also be detected during the HTNV infection process (Supplementary Fig. 21c). To determine whether NICD participated in HTNV-triggered activation of the NF-κB pathway at the early infection phase, NICD was exogenously expressed in RBP-J$^{CKO}$ mBMDM, in which the negative regulation caused by Notch downstream lncRNAs was blocked. Overexpressing NICD in RBP-J$^{CKO}$ mBMDM promoted the phosphorylation and degradation of IκBα even at a low challenge dose of HTNV (Supplementary Fig. 21d-(i)), in which the DNA-binding activity of NF-κB and the production of TNFα were enhanced (Supplementary Fig. 21e). Alternatively, suppressing NICD generation with DAPT weakened p65 phosphorylation against HTNV infection (Supplementary Fig. 21d-(ii)), in which NF-κB activity and TNFα expression were also downregulated (Supplementary Fig. 21f). These data suggested that NICD might facilitate the interaction between IKKβ and p65.

To make clear the exact interaction domains, truncated NICD, p65 or IKKβ was constructed based on the intrinsic domains (Supplementary Fig. 22a). In terms of the interaction between NICD and p65, we found that NICD could interact with truncated p65 containing (aa 1−300) or (aa 401-549) (Supplementary Fig. 22b), and p65 bond to the ankyrin (ANK) repeat domain of NICD (Supplementary Fig. 22c). Regarding the interaction of NICD with IKKβ, we found that NICD could pull down IKKβ mutants containing serine/threonine protein kinase catalytic (STKc) domains (Supplementary Fig. 22d), and IKKβ collaborated with the RAM or ANK domain of NICD (Supplementary Fig. 22e). Considering that lnc-ip65 binds to the (aa 1−300) and (aa 401-549) of p65, where there exists the interaction between p65 and NICD, we wondered whether lnc-ip65 negatively influenced the NICD-p65 interaction. Expectedly, the full length of lnc-ip65, as well as the head or middle region of lnc-ip65, could significantly detach p65 from NICD without affecting the NICD-IKKβ interaction (Supplementary Fig. 22f). We also found that HTNV-induced Notch signaling was also crucial for early-phase activation of inflammatory macrophages in humans (Supplementary Fig. 22g). Replenishing murine-specific lnc-ip65 could conspicuously prohibit p65 and Stat1 phosphorylation, and consolidate the activation of Stat3 and IRF4 in hMDM (Supplementary Fig. 22h). The release of proinflammatory cytokines, especially TNFα, IL-6 and IL-8, was prominently decreased in human macrophages once lnc-ip65 was exogenously expressed, in which IL-10 production was enhanced but IFNα generation remained unchanged (Supplementary

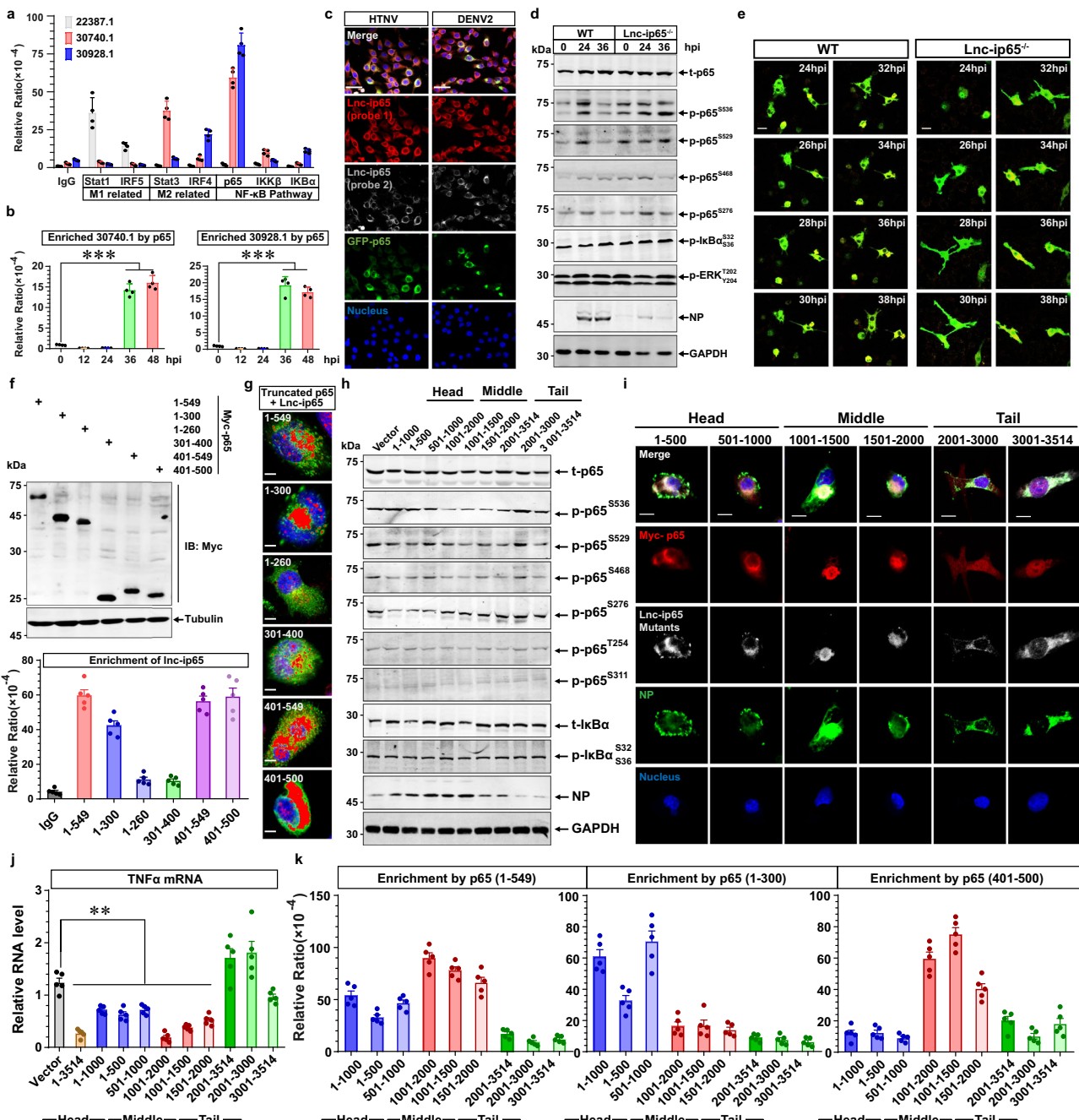

**Fig. 7 | Head and middle region of Lnc-ip65 binds to p65 hinders its phosphorylation. a, b** RIP experiments assessing the interaction between indicated lncRNAs and crucial TFs. Samples are acquired from the mBMDM exogenously expressing the indicated proteins and lncRNAs (**a**, $n = 4$) or in HTNV-infected mBMDM from 0 hpi to 48 hpi (**b**, $n = 4$). **c** Representative immunofluorescent analysis of three independent experiments for lnc-ip65 with different probes (FISH, probe 1 targeted to the 1–1000 nt, and probe 2 to 1001-2000 nt) and GFP-65 in RAW264.7 cells at 36 hpi (MOI = 1). Scale bars, 50 μm. **d** Representative immunoblot analysis of three independent experiments for protein phosphorylation in WT and lnc-ip65$^{-/-}$ mBMDM (MOI = 5). **e** Representative live cell imaging of three independent experiments depicting the translocation of GFP-p65 in WT and lnc-ip65$^{-/-}$ mBMDM (MOI = 1). Live cell imaging is recorded from 24 to 36 hpi. Scale bars, 10 μm. **f** Immunoblot confirmation for truncated p65 expression (upper) and RIP analysis for the interaction of truncated p65 with lnc-ip65 in RAW264.7 cells (bottom). **g** Representative immunofluorescent analysis of three independent experiments for RAW264.7 cells overexpressing p65 mutants and lnc-ip65. The co-

localization of truncated p65 and lnc-ip65 is shown with overexposure. Scale bars, 10 μm. **h** Representative immunoblot analysis of three independent experiments for p65 and IκBα in RAW264.7 cells that were exogenously expressed with different lnc-ip65 mutants at 24 hpi with an MOI of 5. **i** Representative immunofluorescent analysis of three independent experiments for the subcellular localization of myc-p65 and lnc-ip65 mutants in HTNV-infected RAW264.7 cells (MOI = 5, 24 hpi). Scale bars, 10 μm. **j** qRT-PCR analysis for TNFα mRNA expression from (**h**). Vector as control (vs. 1–3514, 1–1000, 1–500, 501–1000, 1001–2000, 1001–1500 or 1501–2000), $p < 0.0001/ = 0.0024/ = 0.0004/ = 0.0028/<0.0001/ < 0.0001/ <0.0001$. **k** RIP analysis for the interaction of truncated p65 with lnc-ip65 mutants in RAW264.7 cells. Data are shown as the mean ± SEM, and are representative of three independent experiments. Analysis is performed using one-way ANOVA (Dunnett's multiple comparisons test). $*p < 0.05$, $**p < 0.01$, $***p < 0.001$. Molecular weight markers are shown to the left of the blots in kDa, and antibodies used are indicated to the right. Source data are provided as a Source Data file.

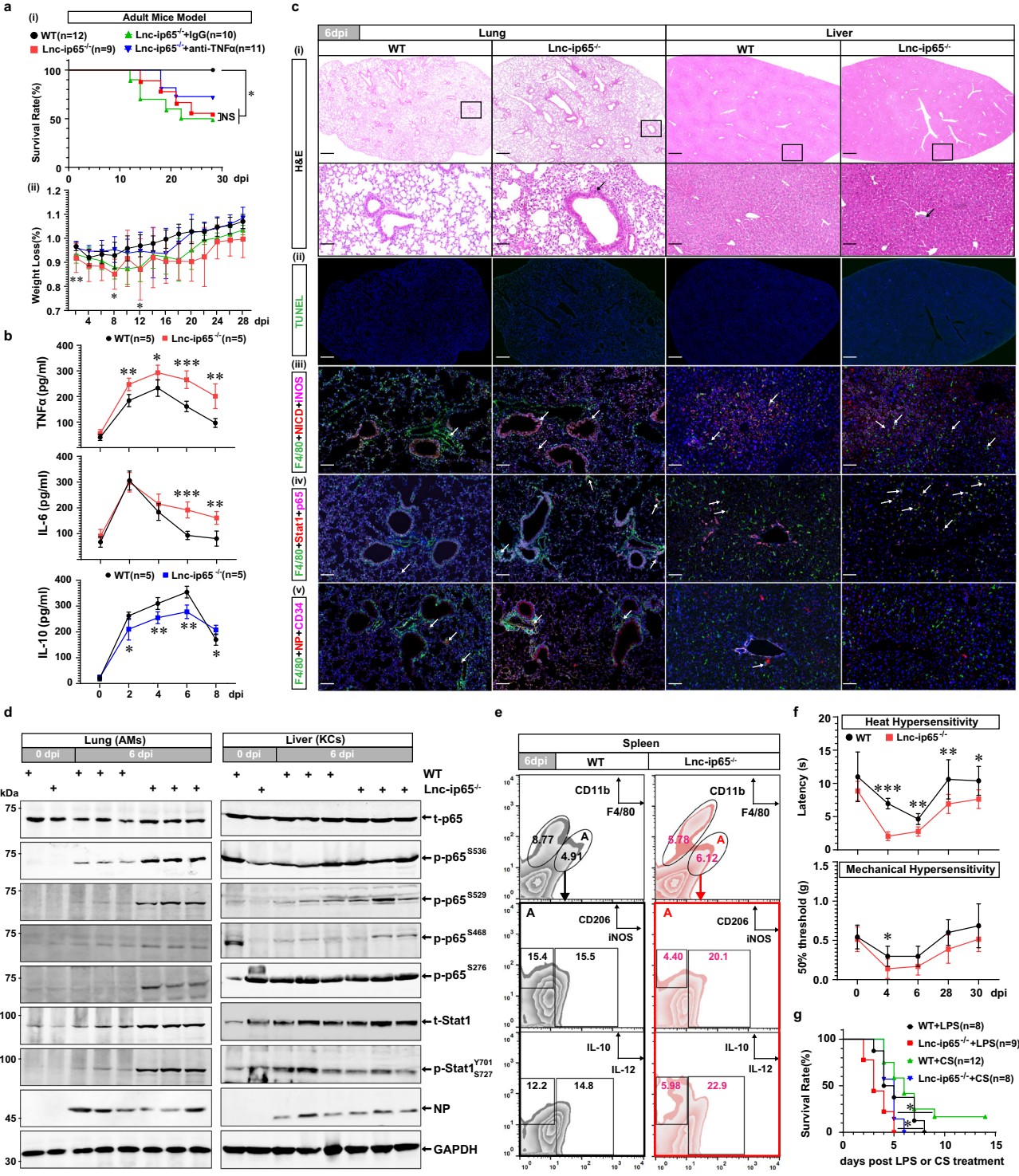

Fig. 23a, b). These results indicated that compensating for lnc-ip65, which possibly rewired the macrophage phenotype from M1 to M2, might be a potential anti-inflammatory therapeutic strategy in HFRS patients.

## HTNV NP facilitates Notch-mediated M1 activation and exacerbates disease progression

It was unclear how HTNV infection activates Notch signaling. Bioinformatic analysis by P-HIPSTer[73] indicated that the viral proteins of HTNV could not interact with the Notch components (Supplementary Fig. 23c). Nevertheless, we found that direct NP stimulation, but neither exogenously expressing viral RNAs nor treatment with virus-like particles (VLPs) as we previously constructed[11,74], could promote the expression Hes1 and the generation of NICD (Fig. 9a-(i), a-(ii)). DAPT inhibited NP-induced inflammatory gene expression (Fig. 9a-(iii)), suggesting that NP might trigger M1 activation via the Notch pathway. As no similar domains were found between HTNV NP and Notch ligands, NP might indirectly motivate Notch signaling, possibly through the TLR pathway, as previously reported[29].

To evaluate the relationship between NP and host pathogenesis, serum NP was detected in HFRS patients. NP production was positively associated with disease severity and the percentage of M1-like monocytes (Fig. 9b), suggesting that HTNV NP contributed to HFRS pathogenesis. A series of non-neutralizing antibodies against NP were

**Fig. 8 | Exacerbated inflammation and tissue injury correlate with disease progression in HTNV-infected Lnc-ip65$^{-/-}$ mice. a** Survival (-i) and weight loss (-ii) data for 8-week-old WT or lnc-ip65$^{-/-}$ mice with a high HTNV challenge dose (i.m., $8 \times 10^7$ TCID$_{50}$/g). The lnc-ip65$^{-/-}$ mice are treated with IgG or TNFα neutralizing antibody (5 μg/g) every two days from 3 dpi. The mice number of each group is shown in the figure symbols. In (**a**)-(i), WT as control, $p = 0.0108$ (vs. Lnc-ip65$^{-/-}$)/0.0057 (vs. Lnc-ip65$^{-/-}$+IgG); Lnc-ip65$^{-/-}$ as control, $p = 0.4372$ (vs. Lnc-ip65$^{-/-}$+anti-TNFα). In (**a**)-(ii), WT vs. Lnc-ip65$^{-/-}$, $p = 0.0302$, 0.013, 0.0241 of 2, 8, 12 dpi, respectively. **b** Serum cytokine concentration measured by ELISA from 0 to 8 dpi ($n = 5$). For TNFα, $p = 0.0031$, 0.0157, 0.0004 or 0.0018 from 2 to 8 dpi, respectively; for IL-6, $p < 0.0001$, = 0.0031 of 6 and 8 dpi, respectively; for TNFα, $p = 0.0306$, 0.0051, 0.0011 or 0.0177 from 2 to 8 dpi, respectively. **c** Representative H&E (scale bars, 500 μm for the upper and 50 μm for the bottom) (-i), TUNEL (scale bars, 500 μm) (-ii) and immunofluorescent staining (scale bars, 50 μm) (-iii to -iv) of three independent experiments for the lung (left) and liver (right) tissues at 6 dpi. Arrows show the infiltrated lymphocytes (-i), the F4/80$^+$ macrophages marked by iNOS$^+$ NICD$^+$(-iii), p-p65$^+$(-iv),

or NP$^+$(-v). **d** Representative immunoblot analysis of three independent experiments for the phosphorylation of p65 and Stat1 in AMs (left) or KCs (right). **e** Representative flow cytometry analysis of macrophage polarization in the spleens. Related statistical analysis is shown in Supplementary Fig. 19b ($n = 4$). **f** Heat and mechanical hypersensitivity of WT and lnc-ip65$^{-/-}$ mice ($n = 7$). WT vs. Lnc-ip65$^{-/-}$, for heat data, $p = 0.0006$, 0.0012, 0.007 or 0.0262 of 4, 6, 28 or 30 dpi, respectively; for mechanical data, $p = 0.0437$ of 4 dpi. **g** Survival data for WT or lnc-ip65$^{-/-}$ mice treated with LPS (i.p., 5 mg/kg) or CS (i.p., 0.6 mg/g). The mice number of each group is shown in the figure symbols. WT + LPS vs. Lnc-ip65$^{-/-}$+LPS, $p = 0.0334$; WT + CS vs. Lnc-ip65$^{-/-}$+CS, $p = 0.0387$. Data are shown as the mean ± SD, and are representative of two independent experiments. Analysis is performed mainly with the survival curve comparison (**a**-(i), **g**, log-rank [Mantel–Cox] test), two-sided unpaired Student's $t$ test (**a**-(ii), **b**), or Mann–Whitney $U$ test (**f**). *$p < 0.05$, **$p < 0.01$, ***$p < 0.001$. Molecular weight markers are shown to the left of the blots in kDa, and antibodies used are indicated to the right. Source data are provided as a Source Data file.

screened as we previously reported[75], and we found that 1A8 could efficiently reverse NP-mediated M1 activation by restraining TNFα and iNOS production (Fig. 9c). To ensure the functional epitope, different truncated NP proteins were applied. The 0.3NP (0-100 aa) could mimic the pro-M1 effects of 1.3NP (full length), which could be blocked by 1A8 or DAPT (Fig. 9d). Notably, 1A8 treatment improved the survival curve of the lethal neonatal mouse model (Fig. 9e). For 1A8-treated mice, the activation of M1-like monocytes in blood and inflammatory macrophages in spleens were impeded (Supplementary Fig. 23d). The therapeutical effects of the neutralizing antibody 3D8 would be impaired if it was applied later than 5 dpi; however, it was noteworthy that combined application of 1A8 with 3D8 at 5 dpi could protect neonatal mice from lethal HTNV challenge (Supplementary Fig. 23e). The data suggested NP might promote Notch-mediated M1 activation and exacerbate disease progression, while the non-neutralizing antibodies against NP might ameliorate HTNV-induced immunopathogenesis.

## Discussion

Numerous negative modulators of the NF-κB pathway, such as the deubiquitinase TNFAIP3/A20[76], ubiquitin ligase SOCS-1[77], a group of miRNAs[78] and a few lncRNAs[79,80], etc., have been identified as potent anti-inflammatory molecules. However, it is unknown whether distinctive regulatory mechanisms exist for NF-κB signaling between different species. In this study, we reported that several murine-specific lncRNAs controlled by the Notch pathway, particularly lnc-ip65, formed a negative feedback loop to prohibit sustained or excessive activation of the NF-κB pathway in macrophages (Supplementary Fig. 24).

Hantaviruses have drawn worldwide attention as emerging zoonotic viruses. Although it is universally acknowledged that the pathogenesis of HFRS or HPS caused by hantaviruses is highly involved in immoderate immune responses[81,82], the key regulator that governs the initiation and conversion of host inflammation remains unclear. Previous researchers have observed massive NK-cell expansion and activation[83], as well as uncontrolled virus-specific T-cell responses[84], in hantavirus disease progression, which might directly execute tissue-destructive effects but not manipulate the inflammatory status. Moreover, the relationship between Treg cells and hantaviral immunopathogenesis is still under debate[85,86]. Herein, we found that activated inflammatory monocytes or macrophages, but not some other T-cell subsets, showed a correlation with HFRS disease severity and proved that their hyperactivation triggers a TNF-α-centered cytokine storm and leads to the turbulence of the T-cell response. Another intriguing question is why hantaviruses do not cause lethal infection in rodent reservoirs[12,87,88]. Previous studies have shown that hantavirus might interrupt host IFN production by various strategies[89,90], resist TRAIL-mediated cell death[91], disturb virus-specific CTL-associated pathogen clearance[92] and promote Treg-associated immune

suppression[93,94], thus resulting in viral persistence in rodents. Little is known about why excessive inflammation is prevented in hantavirus-infected mice. We identified the differential macrophage phenotype rewired by HTNV, which was consistent with previous studies[43,95], and further demonstrated that the Notch-lncRNA-p65 pathway constrains the magnitude of inflammatory responses in mice versus humans, adding special insights into the immunological mechanisms and identifying new possible targets for intervention.

Recent evidence suggests that Notch signaling is an important modulator of macrophage-mediated immune responses[28,30,65], while the downstream molecular mechanisms, particularly during acute viral infection, largely remain elusive. JEV induces the expression of the miRNA let-7a/b, which activates the Notch-TLR7 pathway and enhances microglia-mediated neuroinflammation[96]. DENV upregulates the expression of Notch ligands through IFN signaling in monocytes and macrophages, which further modulates host Th1/Th2 differentiation during the adaptive immune response but does not affect viral replication[32]. IAV challenge elicits Notch ligand Dll1 expression on macrophages through RIG-I but not the TLR3-TRIF pathway, the blockage of which with GSI would result in higher mortality caused by excessive inflammation and impaired production of IFN-γ in lungs post IAV infection[33]. These data show that Notch signaling might exert either pro- or anti-inflammatory effects by rewiring macrophages during viral diseases, while it was unclear how Notch played a dual role and which factor determined the ultimately deleterious effect. We report that the murine Notch pathway is dynamically activated by HTNV, which will rewire the macrophage phenotype at different infection phases. At the early infection stage, NICD accumulates in the cytoplasm and facilitates p65 phosphorylation by interacting with both p65 and IKKβ, thus promoting M1 polarization. At the late infection stage, NICD translocates into the nucleus and induces various murine-specific lncRNAs, among which lnc-ip65 binds to and suppresses p65 phosphorylation, reprogramming macrophages from the M1 to the anti-M1 state.

Cytoplasmic lncRNAs have previously been reported to be vital immune regulators that affect mRNA stability and translation or influence protein function[97,98]. LncRNA Sros1 stabilized Stat1 mRNA in macrophages by blocking the interaction of Stat1 mRNA with RBP CAPRIN1, promoting IFN-γ-STAT1-mediated M1 polarization[99]. Nuclear Malat1 suppressed IFN production by inhibiting the cleavage of TDP43 to TDP35, which stabilized the Rbck1 pre-mRNA and promoted the proteasomal degradation of IRF3 upon viral infection[100]. LncRNA-GM promoted macrophage antiviral responses by binding to and relieving the suppression of GSTM1 on TBK1 activity[101]. It is unknown whether lncRNA expression is specifically induced by certain stimuli or controlled by classic signaling pathways. Here, a number of lncRNAs were found to be downstream of the Notch pathway, which negatively affected NICD-mediated NF-κB activation, thus reprogramming

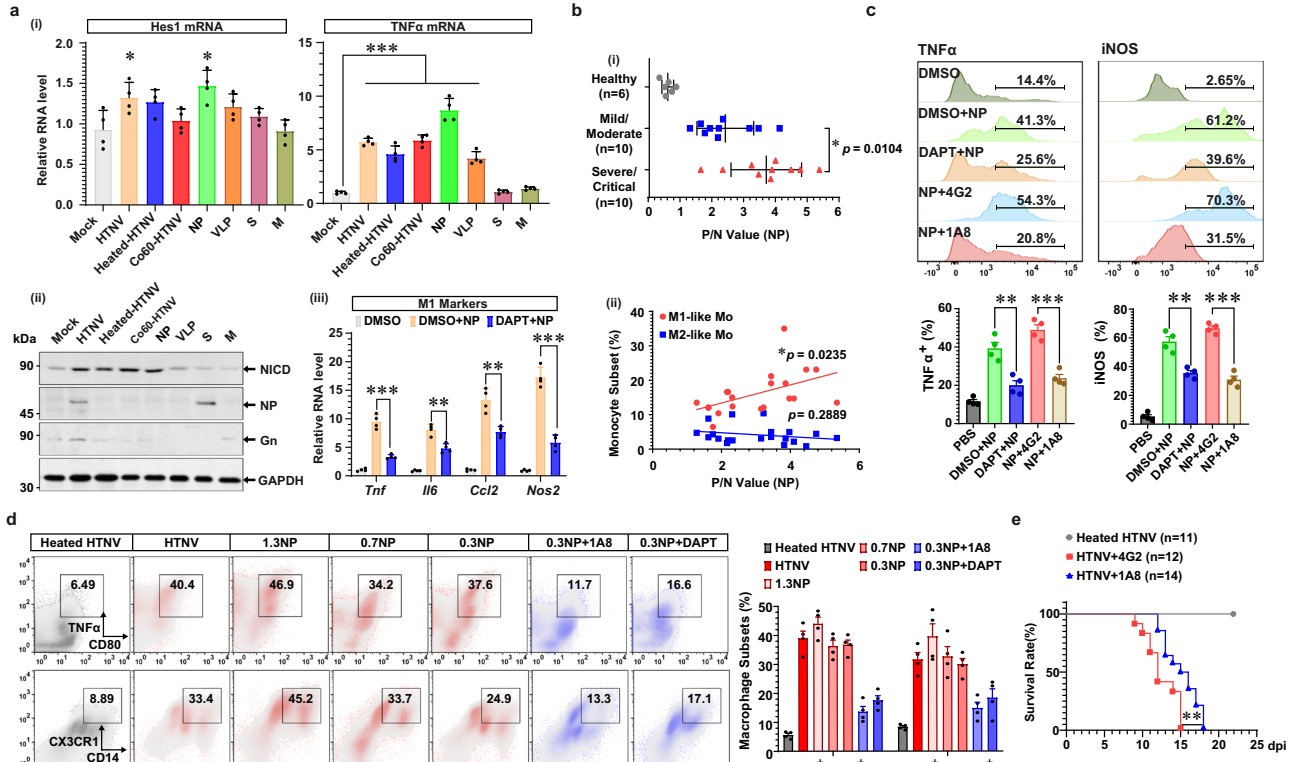

**Fig. 9 | HTNV NP promotes M1 polarization by activating the notch pathway and accelerates disease procession. a** qRT-PCR analysis of Hes1 (-i) or TNFα (-ii), and immunoblot analysis for NICD (-iii). The mBMDM are infected with HTNV (MOI = 1), treated with heat/Co[60]-devitalized HTNV (MOI = 5), transfected with viral RNAs or stimulated with viral proteins (NP, 1 μg/ml; VLP, MOI = 5). qRT-PCR analysis of inflammatory genes in NP-stimulated mBMDM with DPAT pre-treatment (-iv). Mock as control, in the left part of (**a**)-(i), $p = 0.0167$ (vs. HTNV) or 0.0008 (vs. NP); in the right part of a-(ii), $p < 0.0001$ (vs. HTNV, Heated HTNV, Co60-HTNV, NP or VLP). DMSO + NP vs. DAPT + NP, in (**a**)-(iii), $p < 0.0001$ (*Tnf*)/ = 0.002 (*Il6*)/ = 0.0025(*Ccl2*)/ < 0.0001 (*Nos2*). **b** (-i) Serum NP production in HFRS patients with distinct disease severity detected by ELISA. The sample number of each group is shown in the figure. (-ii) The correlation of HFRS patient serum NP with the M1- or M2-like monocyte percentage (*n* = 20). The exact *P* values are shown in the figure. **c** Flow cytometry analysis of TNFα+ or iNOS+ mBMDM at 24 h post-NP stimulation (5 μg/ml) (*n* = 4). The mBMDM are pretreated with DAPT for 12 h and then subjected to NP stimulation. The NP (5 μg/ml) is co-incubated with control antibody 4G2 (the flavivirus group

antibody, 50 ng/ml) or anti-NP antibody 1A8 (50 ng/ml) for 2 h at room temperature and then added to stimulate mBMDM. DMSO + NP vs. DAPT + NP, $p = 0.0026$ (TNFα)/0.0014 (iNOS); NP + 4G2 vs. NP + 1A8, $p = 0.0002$ (TNFα)/< 0.0001 (iNOS). **d** Flow cytometry analysis of CD80+ TNFα+ or CD14+ CX3CR1+ mBMDM at 36 h post-HTNV infection or NP stimulation. The truncated NP peptides are constructed and purified with the baculovirus system. **e** Survival data for 4-day-old neonatal mice challenged with HTNV (i.p., $8 \times 10^5$ TCID$_{50}$/g) and then treated with 4G2 or 1A8 (0.25 μg/g) from 1 dpi to death (every two days). HTNV + 4G2 vs. HTNV + 1A8, $p = 0.0061$. Data are shown as the mean ± SEM, and are representative of two independent experiments. Analysis is performed mainly with one-way ANOVA (**a**-(i), Dunnett's multiple comparisons test), two-sided unpaired Student's *t* test (**a**-(iii), **c**), or the survival curve comparison (**e**, log-rank [Mantel−Cox] test). *$p < 0.05$, **$p < 0.01$, ***$p < 0.001$. Molecular weight markers are shown to the left of the blots in kDa, and antibodies used are indicated to the right. Source data are provided as a Source Data file.

macrophage polarization. Mechanistically, lnc-ip65 was directly bound to the protein domains of p65 that were adjacent to its phosphorylation sites, whose conformational hindrance might disturb the NICD-bridged interaction of p65 with IKKβ and block the S276, S529 and S536 phosphorylation of p65. This finding shed light on a new mechanism of lncRNAs in immunoregulation.

Moreover, two potential intervention tactics for HFRS were proposed in this study. On the one hand, murine-specific lncRNAs were found to hinder immoderate inflammation both in mice and human macrophages, which suggested that applying negative regulons from other species might be a potential therapy choice for patients. On the other hand, the non-neutralizing antibody against NP could incompletely improve host conditions in HTNV-infected neonatal models, possibly by attenuating macrophage-mediated inflammation, which would also prolong the effective therapeutic window of neutralizing antibodies. This indicated that the combination of antibodies against different viral proteins, not only neutralizing antibodies, might achieve better clinical efficacy.

Collectively, we demonstrated the differential macrophage responses against HTNV infection in mice versus humans, and the late-

phase inactivation of inflammatory macrophages in mice prohibited the cytokine storm and protected them from secondary endotoxin sepsis. Murine Notch signaling dynamically rewired the macrophage phenotype by producing NICD and lncRNAs, of which lnc-ip65 could inhibit the NF-κB pathway and impel an anti-M1 status. Blocking Notch activation to prevent M1 activation at the early stage, or applying lnc-ip65 to restrain hyperactivation of M1 at the late stage, might be effective for the control of inflammation and NF-κB-associated autoimmune diseases.

## Methods

The research methods applied in this study followed the guidelines of the World Medical Association's Declaration of Helsinki and subsequent revisions, and the study was reviewed and approved by the ethics committee of the Fourth Military Medical University, and the ethics committee of Tangdu Hospital.

### Human samples and murine experiments
**Study participants.** This study was approved by the Institutional Review Board of Tangdu Hospital (TDLL-2016323). Peripheral blood

samples and related medical records were collected from two hundred thirty-six hospitalized patients aged from 18 to 35 years old at the Department of Infectious Disease, Tangdu Hospital, from October 2016 to March 2018 (HFRS patients, $n = 185$; Japanese encephalitis patients, the febrile phase, $n = 15$; hepatitis B patients, confirmed chronic infection for more than one-year, inactive phase without liver cirrhosis and antiviral therapy, $n = 18$; hepatitis C patients, confirmed chronic infection for more than one-year, inactive phase without liver cirrhosis and antiviral therapy, $n = 18$). All patients were Han Chinese, and the proportion of males to females nearly equalled 1:1. The diagnosis of HFRS or Japanese encephalitis was made based on typical symptoms and signs as well as IgM and IgG antibody positivity against HTNV or JEV in the serum as assessed by ELISA by the Department of Clinical Laboratory, Tangdu Hospital. The diagnosis of chronic HBV or HCV infection was confirmed by viral RNA detection with qRT–PCR. The definition of HFRS phases, classification of disease severity and exclusion criteria were previously described[41,102]. In brief, based on the classically defined 5 stages of HFRS, we classified the HFRS patients in this study into acute phase (including febrile and hypotensive), oliguric, diuretic and convalescent phase (Fig. 1b). In general, samples were collected at 3–6 days for the febrile or hypotension stage, 7–12 days for the oliguric stage, 13–18 days for the diuretic stage and after 18 days for the convalescent stage. The phase within 8 days from the fever onset to the early oliguric stage was typically defined as the acute or early phase of the disease. HFRS disease severity was classified on the basis of clinical and laboratory parameters used in the diagnostic criteria for HFRS in China as (1) mild,mild renal failure without an obvious oliguric stage; (2) moderate, obvious symptoms of uremia, effusion (bulbar conjunctiva), hemorrhage (skin and mucous membrane), and renal failure with a typical oliguric stage; (3) severe, severe uremia, effusion (bulbar conjunctiva and either peritoneum or pleura), hemorrhage (skin and mucous membrane), and renal failure with oliguria (urine output, 50–500 mL/day) for ≤5 days or anuria (urine output, <50 mL/day) for ≤2 days; and (4) critical, for those with ≥1 of the following symptoms during severe disease: refractory shock, visceral hemorrhage, heart failure, pulmonary edema, brain edema, severe secondary infection, and severe renal failure with either oliguria (urine output, 50–500 mL/day) for >5 days, anuria (urine output, <50 mL/day) for >2 days, or a blood urea nitrogen level of >42.84 mmol/L. For the purpose of this study, mild and moderate were considered to be one group, severe and critical were considered to be another group. Additionally, to control for potential confounders, we excluded HFRS patients with autoimmune diseases, viral hepatitis, hematological diseases, diabetes, cardiovascular diseases, and other kidney or liver diseases.

The clinical blood samples of healthy individuals between the ages of 20 and 35 years were obtained from the Blood Transfusion Department of Tangdu Hospital ($n = 55$) in agreement with institutional ethics regulations. To obtain human *monocyte-derived macrophages (hMDM)*, peripheral blood mononuclear cells (PBMCs) were first enriched by Ficoll (TBDscience) from peripheral blood density gradient centrifugation. Then, human monocytes were magnetically purified from PBMCs with negative selection beads (EasySep™ Human Monocyte Isolation Kit, StemCell). Finally, monocytes were primed with recombinant human macrophage colony-stimulating factor (M-CSF) (15 ng/ml, PeproTech) with medium exchange every other day for a week to generate hMDM. Alternatively, the PBMCs were laid into Petri dishes for 4 h, and the supernatant cells were collected to acquire the *monocyte-removed PBMCs*.

**Animal models.** All procedures involving mice were reviewed and approved by the ethics committee of the Fourth Military Medical University, and the ethics committee of Tangdu Hospital. The epidemiological data of the field mice were collected from the Shaanxi Provincial Notifiable Disease Surveillance System in collaboration with

the research team of Pengbo Yu, which was authorized by the government[13]. For the capture, traps were placed outdoors (set as four parallel lines of 25 traps each and spaced intervals of 5 meters). *A. agrarius* mice were removed from the traps once captured for further investigation. According to the standard measurement for HTNV infection in rodents[103–105], the lung tissues were acquired for analysis. Mouse samples weighing 22–28 g and without apparent trauma and skin infection were included for analysis. Only male adult *A. agrarius* mice were used, and the rest were euthanized by $CO_2$ inhalation. To better illuminate the natural infection process of hantaviruses in field mice, the disease phases were classified as HINS (HTNV S/viral RNA: negative, anti-NP IgG/antibodies against viral protein: negative), HIES (HTNV S: positive, anti-NP IgG: negative), HIPS (HTNV S: positive, anti-NP IgG: positive), and HICS (HTNV S: negative, anti-NP IgG: positive) according to the assessment results of viral RNA and host anti-hantaviral antibody in lungs (Fig. 2a). The lung tissues of captured *A. agrarius* mice were acquired for viral RNA or anti-NP IgG detection by qRT-PCR or ELISA, respectively (details were shown in the following). The viral RNA assessment was evaluated as positive if the Cq (quantification cycle) value of HTNV S segments was lower than 30, as we described previously[102]. In brief, virus RNA was extracted using Pure-linkTM Viral RNA/DNA Kits (Invitrogen Life Technologies), according to the manufacturer's instructions. The SuperScript III Platinum One-Step Quantitative RT-PCR System kit (Invitrogen Life Technologies) was used for the real-time RT-PCR assay. The primers and probe were designed on the basis of the sequence alignment of the S segment of the HTNV standard strain 76-118 (NC_005218) and two hantaan virus strains obtained from Shaanxi Province: A16(AF288646.1) and 84FLi(AY017064). The anti-NP IgG assessment was evaluated as positive if the P/N (positive versus negative) value was higher than 2.1, as we described previously[106]. In brief, anti-NP IgG was coated on microplates in 0.1 M sodium carbonate bicarbonate buffer (pH 9.0) at 4 °C overnight. Samples were incubated on the microplates at 37 °C for 2 h. HRP-conjugated 1A8 was used as the detection antibody. The absorbance of the color reaction developed using TMB was measured at 450 nm. An absorbance was required and positive/negative (P/N) > 2.1 was considered significant.

Except for the *A. agrarius* captured from the field (also termed as the field mice), the strain of mice in this study, including the wild type or transgenic mice, is C57BL/6J. The C57BL/6J mice (6- to 8-week-old male adult mice weighing 20–22 g or four-day-old neonatal mice) were provided by the Experimental Animal Center of Fourth Military Medical University. All animals were housed in standard cages in a temperature- and humidity-controlled environment on a 12-h light/dark cycle (temperature: $23 \pm 1$ °C; relative humidity: 50–60%) with free access to water. Mice were euthanized by $CO_2$ inhalation at the appropriate time during the study and tissue samples were removed for further experiments. The numbers of mice are indicated in the figure or figure legends for each experiment.

WT and transgenic mice were bred under specific pathogen-free conditions in the animal facilities belonging to the School of Basic Medical Sciences and housed in groups of up to four mice. The lnc-ip65-deficient mice were generated using the CRISPR/Cas9 system in the C57BL/6J background, the sgRNA targeting sequences of which are shown in Supplementary Fig. 18a. The lnc-ip65 targeting vector was electroporated into C57BL/6J mouse embryonic stem (ES) cells, followed by double drug selection. Positive ES cell clones were expanded and injected into C57BL/6J blastocytes to generate chimeric offspring. The offspring mice were examined by genotyping PCR using the primers shown in Supplementary Fig. 18a.

All animals received care according to institutional guidelines and were randomly assigned to the control or treatment group. For HTNV infection, mice were intramuscularly injected with HTNV ($8 \times 10^5$ $TCID_{50}$/g, $8 \times 10^6$ $TCID_{50}$/g, or $8 \times 10^6$ $TCID_{50}$/g). The HTNV titer was measured by In-cell Western assays as we previously described[106]. In

brief, for In-cell Western assays, cells grown in specific 96-well plates (transparent bottom) until they reached 60–70% confluency were treated as indicated and then fluxed with 4% paraformaldehyde (PFA) at selected time intervals post-infection for the ICW assay. Then, the cells were permeabilized with 0.25% Triton X-100 for 15 min at room temperature (RT) and blocked with LI-COR Odyssey Blocking Solution (LI-COR Biosciences) for 30 min. Cells were incubated at 4 °C overnight with indicated mouse monoclonal antibodies against NP or Gn which were pre-mixed with a rabbit IgG antibody against Tubulin. After five washes with DPBS, the cells were stained with a goat anti-mouse IgG IRDye TM 800 antibody and a goat anti-rabbit IgG IRDye TM 680 antibody at room temperature for 2 h. The microplates, which could be preserved from light at 4 °C for at least 6 months, were scanned with the Odyssey CLx Infrared Imaging System (LI-COR Biosciences), and the integrated fluorescence intensities representing the protein expression levels were acquired using the software provided with the imager station (Odyssey Software Version 3.0, LI-COR Biosciences). The relative amount of NP protein was obtained by normalizing to endogenous Tubulin in the experiments. For HTNV titer measurement, A549 cells grown in microplates until they reached 60–70% confluency were incubated with gradient diluted HTNV, which was propagated in a BALB/c mouse brain and then in Vero E6 cells, at 37 °C for 90 min. Then, the A549 cells were washed with DPBS (HyClone) and cultured in DMEM with 10% FBS. At 48 h post-infection, the ICW assay were performed to detect the amount of HTNV NP; positive/negative (P/N) responses >2.1 were considered significant. The viral titer was calculated as the $TCID_{50}$ using the Reed and Muench formula. For monocyte and macrophage depletion, mice were intraperitoneally injected with clophosome (10 µl/g). For antibody treatment, mice were intraperitoneally injected with 1A8, 3D8, or 4G2 (0.25 µg/g). For the bacterial sepsis challenge, mice were intraperitoneally injected with LPS (5 mg/kg) or CS (0.6 mg/g).

## In vitro experiments

**Cell culture.** THP-1, Vero E6, bEnd.3, NIH/3T3, RAW264.7, and MH-S cells were cultured in DMEM (HyClone) supplemented with 10% (v/v) fetal bovine serum (FBS, Gibco). Cell source is shown in Supplementary Table 4. The suspension THP-1 cells were stimulated with PMA (25 ng/ml, Sigma-Aldrich) for 24 h to differentiate into adherent macrophages. RAW264.7 and THP-1 cells stably expressing GFP-p65 and RFP-IκBα were constructed with a lentivirus system and screened with puromycin and neomycin sequentially.

**Primary macrophage acquisition.** To generate murine *bone marrow-derived macrophages (mBMDM)*, the femur and tibia were removed from sacrificed adult mice. The bones were first rinsed with sterile phosphate-buffered saline (PBS) containing 0.1% (v/v) penicillin–streptomycin (P/S) solution. Subsequently, the bone marrow was flushed with Roswell Park Memorial Institute 1640 (RPMI 1640, HyClone) containing 10% FBS and 0.1% P/S and filtered with a cell strainer (70 mm). Cells were resuspended in RPMI 1640 after centrifugation and then primed with M-CSF (20 ng/ml, PeproTech) with medium exchange every other day for four days to generate mBMDM.

Mouse *peritoneal macrophages (mPMφ)* were isolated from the peritoneal cavities of mice 3 d after injection with thioglycolate medium and were cultured in DMEM supplemented with 10% FBS. After 2 h, nonadherent cells were removed by thorough washing, and adherent cells (mPMφ) were infected.

To harvest mouse *alveolar macrophages (AMs)*, bronchoalveolar lavage was performed. The vein catheter (27G) was installed into the trachea through a small incision after sacrificing the adult mice, and then PBS with EDTA (2 mM) was administered to unfold the lung tissue and retrieve the cells in suspension. Cells were centrifuged and seeded into cell culture dishes and stimulated with GM-CSF (20 ng/ml,

PeproTech) for 24 h, and finally, the adherent cells (AMs) were collected for further experiments.

*Kupffer cells (KCs)* were extracted as described previously[107,108]. In brief, adult mice were sacrificed and underwent liver perfusion with Hank's balanced salt solution (HBSS, HyClone) (from 3 ml/min to 7 ml/min). The excised liver tissues were digested with RPMI 1640 containing 0.1% (v/v) type IV collagenase (Sigma-Aldrich) and bathe-watered. Following digestion, the liver homogenate was filtered and centrifuged to acquire the cell suspension. KCs were further separated from hepatocytes and other sinusoidal cells by gradient centrifugation (300 g, 50 g, and $300 \times g$ for 5 min at 4 °C) and then purified from satellite cells with the method of selective adherence to plastic.

**Transfection.** The indicated plasmids were transfected into NIH/3T3 cells using JetPEI reagents (Polyplus). siRNA transfection was performed with Lipofectamine 2000 (Invitrogen) at 24 h before infection, the sequences of which are shown in Supplementary Table 2. For LNA-mediated RNAi, the LNAs were directly added to the medium of mBMDM (50 nM, the short oligonucleotides would be taken up naturally by cells). The exogenous expression of plasmids in murine macrophages relied on electrotransfection with Neon™ transfection system instruments (Invitrogen, Cat# MPK5000). The virus strains used in this paper were preserved in our lab and propagated in Vero E6 cells.

**Viral infection.** HTNV strain 76–118 (24th generation), as well as DENV2 (31th generation), SeV (12th generation), VSV (17th generation), EV71 (21th generation) and HSV-2 (9th generation), was propagated in mice brain and then in Vero E6 cells. The infection of HTNV, DENV2, SeV, VSV, EV71 and HSV-2 was performed as previously described[11,109–113]. In brief, Cultured cells were infected with the indicated multiplicities of infection (MOI) of HTNV or other viruses. After 2 h, the virus-containing medium was discarded, and the cells were washed thoroughly with the sterile medium and replaced with culture medium. As a control, cells were incubated with culture supernatant from uninfected Vero E6 cells, which were referred to as mock-infected cells.

**Live cell imaging.** RAW264.7 or THP-1 (primed by PMA) cells stably expressing GFP-p65 and RFP-IκB, as well as the WT, RBP-J$^{CKO}$ or lnc-ip65$^{-/-}$ mBMDM transiently expressing these proteins through electro-transfection, were seeded into 35 mm µ dishes (ibidi, Cat# 81156) and infected with HTNV (MOI = 1). At the late infection stage, the µ-dish was transferred to the climate chamber (37 °C, 5% CO$_2$), which was connected to the Live Cell Station (A1R-HD25, Nikon). Fluorescent (GFP and RFP filter) images were chosen randomly and acquired with a 40× objective every 10 min from 24 hpi to 36 hpi. Single images were then merged, and movies were recorded with the Imaging Software NIS-Elements F Ver4.60.00 (Nikon). Representative visual fields were selected and m for each group, and the representative view is shown in figures or videos.

## Macrophage function

**Immunophenotype.** To evaluate the in vitro *phagocytosis* capacity of mBMDM, FAM-labeled RNAs (22 bp, GenePharma) were added to WT or RBP-J$^{CKO}$ mBMDM at 36 h, or the LNA-pretreated WT mBMDM at 36 h (MOI = 1, 3 µg RNAs/$2.5 \times 10^5$ cells). The FAM$^+$ macrophages were under a fluorescence microscope at 24 h post-treatment. For assessment of the *chemotaxis* ability of macrophages, the mBMDM and bEnd.3 cells were added to the middle and bottom layers of the transwell plate (6.5 mm Transwell® with 5.0 µm pore polycarbonate membrane, Corning), and HTNV was added to the intervals between them at an MOI of 1. The number of migrating macrophages on the back of the middle layer (the region towards the bottom) was counted through crystal violet staining at 24 hpi. The *antigen-presenting* ability was measured by the expression of CD80 and CD86 through flow

cytometry. The *immunoregulatory* function was detected by the production of cytokines or chemokines with ELISA or qRT-PCR. To assess the *antimicrobial* ability, cellular ROS production was detected with DCFDA/H2DCFDA. In brief, mBMDM with the indicated treatments were harvested and seeded into a dark, clear bottom 96-well microplate and stained by incubating with DCFDA Solution (100 μl/well) for 45 min at 37 °C in the dark. The plate was measured immediately on a fluorescence plate reader at Ex/Em = 485/535 nm in endpoint mode.

**Metabolic phenotype.** Mitochondrial respiration (oxygen consumption rate, OCR) and glycolysis (extracellular acidification rate, ECAR) were measured in mBMDM. WT and RBP-J$^{CKO}$ mBMDM or LNA-pretreated mBMDM were seeded into a Seahorse XFe96 culture plate (Agilent Technologies) and analyzed at 36 hpi on a Seahorse XFe96 Analyser (Agilent Technologies). To assess OCR, oligomycin (1 μM), FCCP (0.75 μM), antimycin A (1 μM) and rotenone (2 μM) were added at the indicated time points. To measure ECAR, glucose (10 mM), oligomycin (1 μM), and 2-DG (50 mM) were added at the indicated time points. The assay protocols were designed, and the data were analyzed using Seahorse Wave desktop software (Version: 2.6, Agilent).

**Mitochondria pathophysiology.** The number and morphological changes in mitochondria were analyzed with transmission electron microscopy (TEM). The HTNV-infected WT or RBP-J$^{CKO}$ mBMDM at the indicated time points were harvested and fixed with 2.5% glutaraldehyde on ice for 2 h, which was followed by fixation in 2% osmium tetroxide. Then, the cells were dehydrated with sequential washes in 50%, 70%, 90%, 95%, and 100% ethanol. Areas containing cells were mounted and thinly sliced. Sections were photographed using a Hitachi HT7700 transmission electron microscope (Hitachi), and the images were processed with a Hitachi TEM system.

### RNA-seq, transcriptomic, and LncRNA data analysis

**Library construction and sequencing.** Total RNA was extracted from WT mBMDM at 0, 12, 24, or 36 hpi, as well as WT and RBP-JCKO mBMDM at 36 hpi, using TRIzol (Invitrogen). Ribosomal RNA was removed using the Ribo-Zero™ kit (Epicentre Biotechnologies). Fragmented RNA (the average length was approximately 200 bp) was subjected to first-strand and second-strand cDNA synthesis followed by adaptor ligation and enrichment with a low cycle according to the instructions of the NEBNext® Ultra™ RNA Library Prep Kit for Illumina (NEB). The purified library products were evaluated using the Agilent 2200 TapeStation and Qubit®2.0 (Life Technologies). The libraries were paired-end sequenced (PE150, sequencing reads were 150 bp) at Guangzhou Ribo Biotechnology (Guangzhou, China) using the Illumina HiSeq 3000 platform.

**Preprocessing of sequencing reads and quality control.** To remove trailing sequences below a Phred quality score of 20 and achieve uniform sequence lengths for downstream clustering processes, raw fastq sequences were treated with Trimmomatic tools (v 0.36) using the following options: TRAIL-ING: 20, MINLEN: 235 and CROP: 235. Sequencing read quality was inspected using FastQC software. Adapter removal and read trimming were performed using Trimmomatic. Sequencing reads were trimmed from the end (base quality less than Q20) and filtered by length (less than 25).

**Quantification of gene expression level.** Paired-end reads were aligned to the mouse reference genome mm10 with HISAT2. HTSeq v0.6.0 was used to count the read numbers mapped to each gene. The whole sample expression levels were presented as the expected number of reads per kilobase of transcript sequence per million base pairs sequenced (RPKM), which is the recommended and most common method to estimate the level of gene expression.

**Identification of new LncRNAs.** The raw data were first filtered to remove low-quality reads, and then the clean data that passed repeated testing was assembled using StringTie based on the reads mapped to the reference genome. The assembled transcripts were annotated using the gffcompare program. The unknown transcripts were used to screen for putative lncRNAs. Putative protein-coding RNAs were filtered out using a minimum length and exon number threshold. Transcripts with lengths greater than 200 nt with predicted ORFs shorter than 300 nt were selected as lncRNA candidates. They were subjected to further screening using CPC/CNCI/Pfam to distinguish the protein-coding genes from the noncoding genes.

**Differential expression analysis.** The statistically significant differentially expressed genes were obtained by an adjusted *p* value threshold of <0.05 and |log2(fold change) |>1 using DEGseq software. Finally, a hierarchical clustering analysis was performed using the R language package gplots according to the RPKM values of differential genes in different groups. Colors represent different clustering information, such as similar expression patterns in the same group, including similar functions or participation in the same biological process.

**GO terms and KEGG pathway enrichment analysis.** All differentially expressed mRNAs were selected for GO and KEGG pathway analyses. GO was performed with KOBAS 3.0 software. GO provides label classification of gene function and gene product attributes (http://www.geneontology.org). GO analysis covers three domains: cellular component (CC), molecular function (MF) and biological process (BP). The differentially expressed mRNAs and the enrichment of different pathways were mapped using the KEGG pathways with KOBAS 3.0 software.

### Molecular analyses

**Antibodies.** For flow cytometry assays, Alexa Fluor® 488 Rat Anti-Mouse IL-6 (MP5-20F3) (BD Biosciences; Cat# 561363; RRID: AB_10694253); Alexa Fluor® 647 Rat Anti-Mouse CD14 (rmC5-3) (BD Biosciences; Cat# 565743; RRID: AB_2739340); Alexa Fluor® 647 Rat Anti-Mouse CD206 (MR5D3) (BD Biosciences; Cat# 565250; RRID: AB_2739133); APC-Cy™7 Mouse Anti-Human CD16 (3G8) (BD Biosciences; Cat# 557758; RRID: AB_396864); APC-Cy™7 Mouse Anti-Human CD3 (SK7) (BD Biosciences; Cat# 557832; RRID: AB_396890); APC-Cy™7 Rat Anti-CD11b (M1/70) (BD Biosciences; Cat# 557657; RRID: AB_396772); APC-R700 Mouse Anti-Human IL-17A (N49-653) (BD Biosciences; Cat# 565163; RRID: AB_2739087); BB515 Mouse Anti-Human CD4 (RPA-T4) (BD Biosciences; Cat# 564419; RRID: AB_2744419); BB700 Hamster Anti-Mouse CD11C (HL3) (BD Biosciences; Cat# 566505; RRID: AB_2869773); BB700 Rat Anti-Mouse CD197 (CCR7) (4B12) (BD Biosciences; Cat# 566462; RRID: AB_2744307); BB700 Rat Anti-Mouse TNF (MP6-XT22) (BD Biosciences; Cat# 566511; RRID: AB_2869775); BUV661 Mouse Anti-Human HLA-DR (G46-6) (BD Biosciences; Cat# 612980); BV421 Mouse Anti-Human RORγt (Q21-559) (BD Biosciences; Cat# 563282; RRID: AB_2738114); BV421 Rat Anti-Human and Viral IL-10 (JES3-9D7) (BD Biosciences; Cat# 564053; RRID: AB_2738566); BV421 Rat Anti-Mouse CX3CR1 (Z8-50) (BD Biosciences; Cat# 567531); BV480 Rat Anti-Mouse F4/80 (T45-2342) (BD Biosciences; Cat# 565635; RRID: AB_2739313); BV510 Mouse Anti-Human CD14 (MφP9) (BD Biosciences; Cat# 563079; RRID: AB_2737993); BV510 Mouse Anti-Human IFN-γ (B27) (BD Biosciences; Cat# 563287; RRID: AB_2738118); BV605 Mouse Anti-Human CD206 (19.2) (BD Biosciences; Cat# 740417; RRID: AB_2740147); BV605 Mouse Anti-Human CD25 (2A3) (BD Biosciences; Cat# 562660; RRID: AB_2744343); BV605 Rat Anti-Mouse CD192 (CCR2) (475301) (BD Biosciences; Cat# 747969; RRID: AB_2872430); BV650 Mouse Anti-Human CD11c (B-ly6) (BD Biosciences; Cat# 563404; RRID: AB_2732048); FITC Mouse Anti-HTNV NP (1A8)

(Prepared by our Lab); FITC Rat Anti-Mouse IL-10 (JES5-16E3) (BD Biosciences; Cat# 554466; RRID: AB_395411); FITC Rat Anti-Mouse Ly-6C (AL-21) (BD Biosciences; Cat# 561085; RRID: AB_394628); FITC Rat Anti-Mouse TNF (MP6-XT22) (BD Biosciences; Cat# 561064; RRID: AB_395379); FITC Mouse Anti-Human CD11b (ICRF44) (BD Biosciences; Cat# 562793; RRID: AB_1645544); PE Hamster Anti-Mouse CD80 (16-10A1) (BD Biosciences; Cat# 561955; RRID: AB_395039); PE Mouse anti-Human FoxP3 (236A/E7) (BD Biosciences; Cat# 560852; RRID: AB_10563418); PE Mouse Anti-Human IL-8 (G265-8) (BD Biosciences; Cat# 554720; RRID: AB_395529); PE Rat Anti-Mouse CD86 (GL1) (BD Biosciences; Cat# 561963; RRID: AB_10896971); PE Rat Anti-Mouse F4/80 (T45-2342) (BD Biosciences; Cat# 565410; RRID: AB_2687527); PE Rat Anti-Mouse IL-12 (p40/p70) (C15.6) (BD Biosciences; Cat# 554479; RRID: AB_395420); PE Rat Anti-mouse iNOS (CXNFT) (Thermo Fisher; Cat# 12-5920-82; RRID: AB_2572642); PE-Cy™7 Mouse Anti-GATA3 (L50-823) (BD Biosciences; Cat# 560405; RRID: AB_1645544); PerCP-Cy™5.5 Mouse Anti-Human TNF (MAb11) (BD Biosciences; Cat# 560679; RRID: AB_1727579); PerCP-Cy™5.5 Mouse Anti-T-bet (O4-46) (BD Biosciences; Cat# 561316; RRID: AB_10611726). For flow cytometry, antibodies are added 1 µl per $10^6$ cells. Respective antibodies are shown in the Supplementary Table 4.

For immunoblot and immunofluorescent measurements, Anti-NF-κB p65 Antibody (Abcam; Cat# ab16502; RRID: AB_443394); Anti-activated Notch1 Antibody (NICD) (Abcam; Cat# ab8925; RRID: AB_306863); Anti-CD34 Antibody [EP373Y] (Abcam; Cat# ab81289; RRID: AB_1640331); Anti-DDDDK Tag (Binds to FLAG® tag sequence) Antibody [F-tag-01] (Abcam; Cat# ab18230; RRID: AB_444336); Anti-ERK1 + ERK2 (phospho T202 + Y204) Antibody [ERK12T202Y204-A11] (Abcam; Cat# ab278538); Anti-ERK1 + ERK2 Antibody [EPR17526] (Abcam; Cat# ab184699; RRID: AB_2802136); Anti-F4/80 Antibody [CI:A3-1] (Abcam; Cat# ab6640; RRID: AB_1140040); Anti-GAPDH Antibody [6C5] (Abcam; Cat# ab8245; RRID: AB_2107448); Anti-GFP Antibody (Abcam; Cat# ab290; RRID: AB_303395); Anti-HA Tag Antibody (Abcam; Cat# ab9110; RRID: AB_307019); Anti-IKKα + IKKβ (phospho S180 + S181) Antibody (Abcam; Cat# ab55341; RRID: AB_883038); Anti-IKKα + IKKβ Antibody [EPR16628] (Abcam; Cat# ab178870; RRID: AB_2801301); Anti-iNOS Antibody [EPR16635] (Abcam; Cat# ab210823; RRID: AB_2861417); Anti-IRF4 Antibody (Santa Cruz Biotechnology; Cat# sc-48338; RRID: AB_627828); Anti-IRF5 Antibody [EPR17067] (Abcam; Cat# ab181553; RRID: AB_2801301); Anti-IκB α (phosphoS36) Antibody [EPR6235(2)] (Abcam; Cat# ab133462; RRID: AB_2801653); Anti-IκBα (phospho S32) Antibody [EPR3148] (Abcam; Cat# ab92700; RRID: AB_10562951); Anti-IκBα Antibody [E130] (Abcam; Cat# ab32518; RRID: AB_733068); Anti-Jagged1 Antibody (Abcam; Cat# ab7771; RRID: AB_2280547); Anti-Jagged2 Antibody [EPR3646] (Abcam; Cat# ab226814); Anti-JNK1 (phospho T183/Y185) Antibody [EPR20763] (Abcam; Cat# ab215208); Anti-JNK1 Antibody [EPR17557] (Abcam; Cat# ab199380); Anti-Lamin B1 Antibody (Abcam; Cat# ab16048; RRID: AB_443298); Anti-Myc Tag Antibody [9E10] (Abcam; Cat# ab32; RRID: AB_303599); Anti-NF-κB p65 (phospho S276) Antibody (Abcam; Cat# ab194726); Anti-NF-κB p65 (phospho S468) Antibody (Abcam; Cat# ab31473; RRID: AB_881299); Anti-NF-κB p65 (phospho S529) Antibody (Abcam; Cat# ab97726; RRID: AB_10681170); Anti-NF-κB p65 (phospho S536) Antibody (Abcam; Cat# ab86299; RRID: AB_1925243); Anti-Notch1 Antibody [EP1238Y] (Abcam; Cat# ab52627; RRID: AB_881725); Anti-Notch2 Antibody (Abcam; Cat# ab137665); Anti-Notch3 Antibody (Abcam; Cat# ab23426; RRID: AB_776841); Anti-STAT1 (phospho S727) Antibody [EPR3146] (Abcam; Cat# ab109461; RRID: AB_10863745); Anti-STAT1 (phospho Y701) Antibody (Abcam; Cat# ab30645; RRID: AB_779082); Anti-STAT1 Antibody (Abcam; Cat# ab47425; RRID: AB_882708); Anti-STAT3 (phospho S727) Antibody [E121-31] (Abcam; Cat# ab32143; RRID: AB_2286742); Anti-STAT3 (phospho Y705) Antibody [EPR23968-52] (Abcam; Cat# ab267373; RRID: AB_2889877); Anti-STAT3 Antibody [EPR787Y] (Abcam; Cat# ab68153; RRID: AB_2889877); Anti-Tubulin Antibody (Abcam; Cat# ab6046; RRID: AB_2210370); Donkey Anti-Goat IgG H&L (Cy3 ®) (Abcam; Cat# ab6949; RRID: AB_955018); FITC Anti-NF-κB p65 (phospho S536) Antibody [NFKBp65S536-B7] (Abcam; Cat# ab278631); Goat Anti-Mouse IgG H&L (Cy3 ®) (Abcam; Cat# ab97035; RRID: AB_10680176); Goat Anti-Mouse IgG H&L (Cy5 ®) (Abcam; Cat# ab6563; RRID: AB_955068); Goat Anti-Rabbit IgG H&L (Cy3 ®) (Abcam; Cat# ab6939; RRID: AB_955021); Goat Anti-Rabbit IgG H&L (Cy5 ®) (Abcam; Cat# ab6564; RRID: AB_955061); Human/Mouse/Rat RelA/NF-κB p65 Antibody (R&D Systems; Cat# AF5078; RRID: AB_2179033); IRDye® 680RD Goat Anti-Mouse IgG (H + L) (LI-COR; Cat# 925-68070; RRID: AB_2651128); IRDye® 800CW Goat Anti-Rabbit IgG (H + L) (LI-COR; Cat #926-32211; RRID: AB_621843); Mouse monoclonal Anti-HTNV Gn (Gn-1) (Prepared by our Lab); Mouse monoclonal Anti-HTNV NP (1A8) (Prepared by our Lab); Mouse/Rat Notch1 Antibody (R&D Systems; Cat# AF1057; RRID: AB_2153372); Phospho-NF-κB p65/RelA-S276 Rabbit pAb (ABclonal; Cat# AP0123; RRID: AB_2771505); Phospho-NF-κB p65/RelA-S468 Rabbit pAb (ABclonal; Cat# AP0446; RRID: AB_2771508); Phospho-NF-κB p65/RelA-S529 Rabbit pAb (ABclonal; Cat# AP0944; RRID: AB_2863855); Phospho-NF-κB p65/RelA-S536 Rabbit pAb (ABclonal; Cat# AP0475; RRID: AB_2771511); Rabbit Anti-Rat IgG H&L (FITC) (Abcam; Cat# ab6730; RRID: AB_955327). For immunoblot, antibodies are diluted as 1:1000; for immunostaining, antibodies are diluted as 1:200. Respective antibodies are shown in Supplementary Table 4.

**Flow cytometry (FCM).** The monocytes or T cells from HFRS patients underwent intracellular cytokine staining immediately, while for the in vitro experiments, the secretion inhibitors are used to precisely measure cytokine production. Generally, FcγII/III receptors of monocytes and macrophages were blocked with anti-CD16/32 antibody (BD Bioscience) before staining the cell surface markers, and brilliant stain buffer (BD Bioscience) was applied prior to staining intracellular cytokines. For the TFs (FoxP3, RORγt, GATA3, and T-bet) in T cells from healthy or patient PBMCs, the BD Pharmingen™ TF Buffer Set was applied. The cells were manipulated in FCM buffer during the FCM assays, which included PBS containing 2% FBS (Gibco) and 2 mM EDTA. For in vitro assays, cells were enzymatically detached with Trypsin-EDTA solution (Solarbio) and subsequently washed and processed with FCM buffer. For the FCM detection of macrophages in spleens, the single-cell suspension of the spleen tissue was acquired through gentle grinding and filtration with a 70 µm cell strainer, and the erythrocytes were lysed with RBC lysis buffer (Gibco). The main procedure was as follows: Cell acquisition→ Fc receptor block→ Surface marker staining→ Permeabilization and fixation→ brilliant stain buffer treatment→ intracellular iNOS or cytokine staining→ Compensation adjustment with beads→ Samples were analyzed with a BD FACSCalibur™ 3-laser flow cytometer or BD FACSCanto 10-laser flow cytometer (cell number = 10,0000/group). Finally, the data were processed with FlowJo v10 (TreeStar). Respective antibodies are shown in the Supplementary Table 4.

**Bio-plex multiplex immunoassay.** The serum samples were centrifuged at 16,000 × $g$ for 15 min at 4 °C and then diluted (for serum, sample diluent HB, 1:4; for cell supernatants, culture media, 1:5). After preparing standards, controls and samples, the Bio-plex multiplex immunoassay was conducted as the workflow showed: prewetting wells→ Adding the magnetic beads containing the antibodies against various cytokines and chemokines (50 µl, forty kinds of cytokines and chemokines)→ Adding the sample/standard/control (incubation on a shaker at 120 × $g$ for 1 h at RT)→ Adding biotinylated detection antibodies containing the phycoerythrin fluorescent reporters (25 µl, incubation on shaker at 120 × $g$ for 30 min at RT)→ Adding streptavidin-PE (50 µl, incubation on shaker at 120 × $g$ for 10 min at RT)→ Resuspending in assay buffer (125 µl, shaking at 120 × $g$ for 30 s)→ Acquiring data on Bio-Plex system (Bio–Rad).

**Enzyme-linked immunosorbent assay (ELISA).** Sandwich ELISA was applied to detect *HTNV NP* as we previously described[106]. Briefly, the ani-NP mouse monoclonal antibody 1A8 was coated on microplates in 0.1 M sodium carbonate bicarbonate buffer (pH 9.0) at 4 °C overnight. Patient serum or mouse tissue lysates with RIPA buffer (Sigma–Aldrich) were collected after centrifugation and then incubated on microplates at 37 °C for 2 h. HRP-conjugated 1A8 was used as the detection antibody. Color reaction developed with the tetramethylbenzidine (TMB, Abcam) and stop solution (2 M $H_2SO_4$), the absorbance of which was measured at 450 nm. An absorbance was acquired and positive/negative (P/N) > 2.1 was considered significant. The results are presented as ratios of the sample value versus that of the negative control (P/N value).

Indirect ELISA was conducted to assess *mouse IgG against HTNV NP* based on the manufacturer's information (WAITAI BioPharm). In brief, the recombinant NP was coated on microplates and incubated with mouse lung tissue lysates (dilution with 1:30 by PBS). HRP-conjugated anti-mouse IgG antibody was added, and the absorbance was assessed after TMB treatment at 450 nm. The results are shown as anti-NP IgG positive or negative to analyze the disease phase stage for the HTNV-infected field mice.

The concentrations of multiple *cytokines and chemokines* from cell supernatants or mouse tissues were evaluated with ELISA kits (Abcam or R&D Systems) according to the manufacturer's instructions. In short, standard samples were prepared with gradient dilution to build the standard curve. The samples were diluted with special buffer, added to a plate precoated with anti-cytokine or chemokine antibodies and then reacted with HRP-conjugated detection antibodies. TMB and stop solution were added sequentially to measure the OD450. The cytokine or chemokine concentration was calculated with the standard curve.

**Protein preparation.** To detect M1/M2-related signaling in macrophages or confirm the overexpression efficacy in co-IP experiments, *whole-cell lysates* (WCLs) were collected with RIPA lysis buffer (Sigma-Aldrich) supplemented with protease and phosphatase inhibitors (Sigma-Aldrich) for further immunoblot analysis. To assess the activation of the NF-κB, JAK/STAT or IRF pathway post HTNV infection in murine versus human macrophages, the translocation of key TFs, such as p65, Stat1, IRF4, and IRF5, was determined with *nuclear and cytoplasmic extraction* reagents (Thermo Fisher). In brief, human or murine macrophages were harvested at the indicated time points with trypsin-EDTA (Solarbio). The cell pellets were acquired through washing and centrifuging. Then, CER I (100 µl) was added to the packed cells (10 µl) with vigorous vortexing for 15 s to fully suspend the cell pellet and incubated on ice for 10 min, destroying the cell membrane but not the karyolemma. Next, ice-cold CER II (5.5 µl) was added to the sample with vigorous vortexing for 5 s, incubation on ice for 1 min, and repeated vigorous vortexing for 5 s. Supernatants containing the cytoplasmic extracts were collected after centrifugation at 16,000 × *g* for 10 min. The insoluble pellets were suspended in ice-cold NER (50 µl) with intermittent vigorous vortexing for 15 s and incubation on ice for 10 min, and this procedure was repeated four times (total of 40 min). Finally, the supernatant fraction containing nuclear extracts was obtained after centrifugation at 16,000 × *g* for 15 min.

**Immunoblot assay.** The protein concentration was first determined based on the bicinchoninic acid (BCA) method using the CompatAble™ BCA Protein Assay Kit (Thermo Fisher). Equal amounts of protein were boiled at 95 °C for 10 min, separated by SDS-PAGE at different concentrations, and then electrophoretically transferred onto polyvinylidene fluoride membranes (PVDF). After blocking with 5% nonfat milk in TBS, the membrane was incubated with the primary antibodies, followed by secondary antibodies labeled with infrared dyes. For the assessment of protein phosphorylation, the antibody targeted at various phosphorylated points was applied separately or combined for the first scanning, and then the PVDF membrane was stripped with the restoration buffer (Thermo Fisher) and subjected to secondary antibody incubation with the total proteins, as well as the infrared dye-labeled antibodies. The signals on the PVDF membrane were visualized using the Odyssey Infrared Imaging System (LI-COR Biosciences).

**Coimmunoprecipitation (Co-IP) assay.** Cells transfected with the appropriate plasmids were harvested and lysed with IP lysis buffer (50 mM Tris-HCl [pH 7.4], 150 mM NaCl, 1% [w/v] Triton X-100, 1 mM EDTA [pH 8.0], 0.1% [v/v] SDS, and protease inhibitor cocktail) for 30 min. The supernatants were collected via centrifugation at 28,000 × *g* for 25 min at 4 °C. The protein extract was incubated with equilibrated magnetic beads (for assessing the protein interaction with the exogenous expression system; beads of Bimake) or protein G Sepharose (for detecting the endogenous interaction; Sepharose of Proteintech) that were coincubated with the desired primary antibodies overnight at 4 °C. Beads or Sepharose were collected and washed three times with washing buffer (5% [w/v] sucrose, 5 mM Tris-HCl [pH 7.4], 5 mM EDTA [pH 8.0], 500 mM NaCl, 1% [v/v] Triton X-100). Then, the beads were boiled at 100 °C for 5 min in 5× SDS protein loading buffer and analyzed by immunoblotting.

**Immunofluorescence assay (IFA).** Cells with the indicated treatment were fixed with ice-cold 4% (w/v) paraformaldehyde (PFA, Sigma-Aldrich) for 15 min and then permeabilized with 0.1% Triton X-100 (Sigma-Aldrich) for 20 min at RT. After blocking with 3% bovine serum albumin (BSA, Sigma-Aldrich) for 30 min, specific primary antibodies (1:50 to 1:200 dilution) were added and incubated overnight at 4 °C. After five washes with DPBS, the secondary antibodies, namely, FITC-, Cy3- or Cy5-conjugated goat anti-rabbit or goat anti-mouse IgG (Abcam), were used for detection (incubation at 37 °C for 1 h). Cell nuclei were stained with DAPI (Thermo Fisher) for 5 min at RT. After sealing with the ProLong™ Gold Antifade Mountant (Thermo Fisher), the samples were observed using a fluorescence microscope (A1R-HD25, Nikon). To observe the localization relationship between lncRNAs and p65, IFA was performed after the FISH or RNAScope experiments.

**RNA extraction, quantitative real-time PCR (qRT-PCR) analysis, and northern blot.** Total cellular RNA was extracted with TRIzol reagent (Invitrogen) and the Total RNA Extraction Kit (TIANGEN Biotech), the concentration of which was measured with a spectrophotometer. Quantitative real-time PCR (qRT-PCR) was performed with PrimeScript RT Master Mix (TaKaRa) according to the manufacturer's protocol. Each cDNA sample was denatured at 95 °C for 5 min and amplified for 35 cycles of conditions including 15 s at 98 °C, 30 s at 58 °C, and 30 s at 72 °C with a LightCycler 96 (Roche). The mRNA expression level of each target gene was normalized to the respective GAPDH and analyzed using LightCycler® 96 Application Software (Roche). The qRT-PCR primers are listed in Supplementary Table 3. Notably, five pairs of qRT-PCR primers for the newly identified lncRNAs by RNA-seq were designed and applied. The suitable primers were screened with stable results from three independent experiments and are listed in Supplementary Table 3. Northern blotting was performed using a NorthernMax Kit (Ambion) with biotin-labeled probes, the sequences of which are shown in Supplementary Table 3. The viral load of HTNV was quantified based on the qRT-PCR and absolute quantitative PCR.

**Fluorescence in situ hybridization (FISH) and RNAScope assays.** *FISH* was performed with a FISH kit (Ribo Biotechnology) according to the manufacturer's instructions. In brief, cells were fixed with 4% PFA (Sigma-Aldrich) for 10 min at RT and permeabilized with 0.5% Triton X-100 (Sigma-Aldrich) for 15 min at RT. Prehybridization was

performed with lncRNA FISH probe mix at 37 °C for 30 min, and then hybridization was performed by adding lncRNA FISH probe mix and incubating the mixture at 37 °C overnight. After washing with 4×, 2×, and 1× SSC (1×SSC is 0.15 M NaCl, 0.015 M Na-citrate), the cell nuclei were stained with DAPI (Thermo Fisher).

*RNAScope* was performed with an RNAscope Fluorescent Multiplex Reagent Kit (ACD Bio) based on the manufacturer's protocols. In short, cells were first placed on slides and fixed in 4% PFA (Sigma-Aldrich) for 30 min, followed by antigen repair with RNAscope® hydrogen peroxide (ACD Bio) for 10 min at RT and digestion with RNAscope® protease III (ACD Bio) for another 10 min at RT in a humidifying box. Next, the cells were incubated at 40 °C with the following solutions: (1) RNAScope probes of target RNAs, namely, lnc-ip65-C3 and HTNV-S-C2 (v/v, 1:1), in hybridization buffer A (6 × SSC, 25% formamide, 0.2% lithium dodecyl sulfate, blocking reagents) for 2 h; (2) preamplifier (AMP1, 2 nM) in hybridization buffer B (20% formamide, 5×SSC, 0.3% lithium dodecyl sulfate, 10% dextran sulfate, blocking reagents) for 30 min; (3) amplifier (AMP2, 2 nM) in hybridization buffer B at 40 °C for 30 min; and (4) label probe (AMP3, 2 nM) in hybridization buffer C (5×SSC, 0.3% lithium dodecyl sulfate, blocking reagents) for 15 min. After each hybridization step, the slides were washed with wash buffer (0.1×SSC, 0.03% lithium dodecyl sulfate) three times at RT. Then, the probe signaling was further recognized and amplified by HRP-C2 (ACD Bio) (for 15 min at 40 °C), followed by chromogenic detection with TSA® Plus Cy3 (Akoya Biosciences) (for 30 min at 40 °C) for detecting HTNV-S. After treatment with HRP-C2-blocker (ACD Bio), the aforementioned steps were repeated with HRP-C3 (ACD Bio) and TSA® Plus Cy5 (Akoya Biosciences) to assess lnc-ip65. Finally, after DAPI staining and Prolong Gold Antifade Mountant (Thermo Fisher) treatment, the samples underwent IFA for p65 detection or were directly observed with a confocal microscope (Nikon).

**RNA immunoprecipitation (RIP).** RNA immunoprecipitation was performed using a Magna RIP™ RNA-Binding Protein Immunoprecipitation Kit (Millipore) according to the manufacturer's instructions in an RNase-free environment. Briefly, mBMDM or RAW264.7 cells that were electrotransfected with the indicated proteins and lncRNAs for 48 h or mBMDM at different time points post HTNV infection were collected and treated with RIP lysis buffer. The anti-myc antibody conjugated magnetic beads (targeting myc-p65 or related mutants) or primary antibodies enriched by protein A + G magnetic beads (targeting M1- or M2-related TFs or NF-κB pathway components) were incubated with cell lysates on a shaker overnight at 4 °C. For the positive control, the anti-SNRNP70 antibody that could pull down the U1 snRNA was applied. The supernatants were discarded after washing on the magnetic frame, and then, proteinase K was added to the sediments with gentle shaking for 30 min at 55 °C. Finally, the supernatants were collected on the magnetic frame, from which the total target protein-attached RNAs were extracted as mentioned above. The enriched lncRNAs were detected by qRT−PCR and normalized to the positive control (U1 snRNA).

**Dual-luciferase reporter assay.** RAW264.7 or THP-1 cells were cotransfected with pNF-κB-luc, pRL-TK, and the indicated plasmids. Cells in 24-well plates were infected with HTNV (MOI = 1) 36 h after electrotransfection and then harvested and lysed. Luciferase activity was measured using the Dual-Glo Luciferase Assay System (Promega) according to the manufacturer's instructions. Luciferase activity was normalized to Renilla luciferase activity.

## Tissue analyses

**Histological analyses.** Paraffin embedded tissue samples were sectioned and stained with haematoxylin and eosin for histomorphological analysis. First, the slides were deparaffinized and hydrated with water, and then they were processed as follows sequentially: Xylene I for 20 min, Xylene II for 20 min; 100% alcohol I for 5 min; 100% alcohol II for 5 min; 75% alcohol for 5 min; and then rinsed in water. Second, the slides were stained in haematoxylin solution, immersed in haematoxylin solution for 3 to 5 min, and rinsed in water. Then, the sections were differentiated with acid alcohol and rinsed again. Blue up sections with ammonia solution, and then the sections were washed slowly in running tap water. Third, the slides were sequentially processed for eosin staining: 85% alcohol I for 5 min, 95% alcohol II for 5 min and eosin for 5 min. Finally, the samples were dehydrated and mounted as follows sequentially: 100% alcohol I for 5 min, 100% alcohol II for 5 min, 100% alcohol III for 5 min, Xylene I for 5 min, and Xylene II for 5 min and mounted with resin. Slides were scanned with the Panoramic MIDI (3DHISTECH). Immunostaining of paraffin sections was preceded by different antigen unmasking methods. Immunohistochemical staining was performed on paraffin-embedded tissue sections using anti-HTNV NP antibodies (1A8 prepared by our lab) and related secondary antibodies, followed by chromogenic detection with DAB.

**Tissue TUNEL and IFA.** Tissue TUNEL assays were performed with the TUNEL Assay Kit (Enhanced FITC) (Elabscience) based on the manufacturer's instructions. In short, the freezing section samples of different mouse tissues were fixed with 4% PFA, followed by incubation with Terminal Deoxynucleotidyl Transferase (TdT) Equilibration working buffer at RT for 30 min and TdT Enzyme working solution at 37 °C for 30 min in a wet bow. Then, nuclei were stained with DAPI, and the slides were sealed with a mounting medium. Tissue IFA was based on frozen sections, the procedure of which was similar to cellular IFA. The imaging data were acquired with Panoramic MIDI (3DHISTECH).

## Animal experiments

**Heat and mechanical sensitivity.** For heat sensitivity, the latency time of paw withdrawal in response to noxious heat (plantar light irradiation) was recorded as previously reported[114]. An analgesia meter (Model 336TG, IITC Life Science Inc.) was used as the heat source. In brief, each mouse was placed in a box containing a smooth, temperature-controlled glass floor (30 °C) and allowed to habituate for 20 min. The heat source was focused on a portion of the hind paw, which was flush against the glass, and a radiant thermal stimulus was delivered to that site. The stimulus shut off when the hind paw moved (or after 6 s to prevent tissue damage). The intensity of the heat stimulus was maintained constant throughout all experiments. The elicited paw movement occurred at latency between 2.5 and 4 s in control animals. Thermal stimuli were delivered four times to each hind paw at 5 to 6 min intervals. Mechanical allodynia was determined by measuring the incidence of the foot withdrawal in response to mechanical indentation of the plantar surface of each hind paw with a sharp, cylindrical probe with a uniform tip diameter of ~0.2 mm provided by a set of Von Frey filaments (0.04–2 g; North Coast medical Inc.). In brief, the mouse was placed on a metal mesh floor and covered with a transparent plastic dome (10 × 15 × 15 cm). The animal rested quietly in this situation after a few (~15) min of exploration. The filament was applied from underneath the metal mesh floor to the plantar surface of the foot. The duration of each stimulus was 3 s, in the absence of withdrawal, and the inter-stimulus interval was 10−15 s. The incidence of foot withdrawal Response frequency was calculated out of 10 applications of the respective filament at 30 s intervals. All measurements were done in awake, unrestrained and age-matched mice.

## Statistics and reproducibility

Statistical analysis is performed with GraphPad Prism (GraphPad Software, Version 8). For comparison of two groups, two-tailed unpaired Student's *t* test is applied. For multiple comparisons, one way ANOVA is performed, followed by Dunnett's multiple comparisons test. Survival analysis is performed with the log-rank [Mantel−Cox]

test. Differences are considered statistically significant when the *p* values were <0.05 (*), <0.01 (**) and <0.001 (***). Statistically non-significant data (*p* value > 0.05) are indicated as NS. Data are presented as the mean ± SEM if not stated otherwise in the figure legends. The exact *p* values are shown in the figure or figure legends. The n number for all experiments, including animal experiments and in vitro experiments are listed in the figure legends. The sample size is chosen based on our prior studies[11,115] and other previous papers with similar experiments[50,79,116], which showed sufficient statistical power for in vitro experiments and animal experiments.

All animal experiments are repeated at least twice and in vitro experiments are repeated at least three times. All results are reproducible and representative data are shown in the figures or supplementary files. The repeated times of independent experiments are shown in the figure legends. No samples or animals are excluded from analyses. Animals are allocated to their respective group at birth by a blinded investigator. For other experiments, including cell experiments, before performing the corresponding treatment, samples are randomly assigned to control and experimental groups by an investigator blinded to subsequent experimental information using the random number table. The standard laboratory procedures are strictly followed to keeping the experimental environment and facilities consistent and performed under the same conditions. Investigators are blinded to group allocation during data collection, image quantification and data analysis.

**Reporting summary**

Further information on research design is available in the Nature Portfolio Reporting Summary linked to this article.

## Data availability

Source data are provided with this paper. Further information and requests for resources or reagents should be directed to and will be fulfilled by Fanglin Zhang (flzhang@fmmu.edu.cn). RNA-seq data are available online from the ArrayExpress database with the accession number E-MTAB-11353 (for Fig. 4a) and E-MTAB-11926 (for Fig. 6a). Source data are provided with this paper.

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

## Acknowledgements

The authors thank Hongyan Qin and Hua Han for providing the RBP-J^CKO and NICD^STOP-floxed mice and guidance for flow cytometry assays, as well as Jing Ye for technical and analytical support for detecting mouse tissue pathogenic injuries. We further thank Zhansheng Jia, Jianqi Lian and Wen Yin for assisting with the clinical sample and medical record collection, as well as Pengbo Yu for *A. agrarius* mice capture. The authors acknowledge the support from the National Natural Science Foundation of China (82172272, 81671994, 82202367, and 31970148) and the Key Research and Development Program of Shaanxi Province (2021ZDLSF01-02). The graphical abstract was created with BioRender.com (YB2689PDFA).

## Author contributions

F.Z., Y.L., and H.M. conceptualized the study. X.Z. and Z.X. supervised the research and provided excellent scientific discussion when this study encountered problems. H.M. and Y.L. designed the methodology. H.M., Y.Y., and T.N. performed the experiments. R.Y. and Y.S. identified and bred the transgenic mice. J.W. and M.L. took charge of the field mice capture. H.L. and W.Y. collected the clinical samples and medical records of patients. H.Z. and X.L. constructed the protein and lncRNA mutants, as well as other vectors and the VLP of HTNV. L.C. and L.Z. contributed the reagents and analytical tools. L.L., Z.X., and X.Z. conducted the pathology analysis. F.Z., H.M., and X.L. acquired the funding. H.M., Y.Y., and T.N. analyzed the data and wrote the manuscript with input from all the authors.

## Competing interests

The authors declare no competing interests.
