## [Peer Review File · Nature Communications]

Disparate Macrophage Responses Are Linked to Infection Outcome of Hantaan Virus in Humans or RodentsREVIEWER COMMENTS

Reviewer #1 (Remarks to the Author):

This manuscript is based on the differential observation that HTV infection in mice is less pathogenic than in humans and the authors provide evidence that the differential activation of monocytes/macrophages in humans is responsible. In doing so they identify differential regulation of lncRNAs in mice (which may be murine specific) that underpin this phenotype.

The first thing to say is that the manuscript is an exceptional amount of data. That said -the density of the data makes the manuscript somewhat hard to read at times. I would recommend distilling out the key data from each figure to make the paper easier to read.

Overall, I have very little to say about the approach - the story builds and utilises powerful KO mouse models to demonstrate pathogenicity can be restored in the mouse model.

From a virological point of view I would prefer some clarity on the role of HTV infection. Is it just HTV infected monocytes that demonstrate this phenotype or is it initiated by HTV infected monocytes which is then propagated by cytokine secretions to form a positive feedback loop? I don't know the frequency of infected macrophages in the systems and so direct versus bystander effects are not clear.

Indeed, I am not convinced this is HTV specific. The authors suggest other RNA viruses trigger the same response in macrophages and so to my mind HTV has revealed the differential regulation of macrophages in mouse and human. This is not a criticism per se but focuses the pathogenesis on the host response rather than the nature of the pathogen itself? This could be discussed

Experimentally, I have no additional expts to suggest - it as I say above pretty exhaustive. I think better curation of the data and presentation would greatly enhance what I think is a really interesting story. (A lot of the descriptive timecourse data could be put into supplementary to allow the main figs to be focused on the key data)

Reviewer #2 (Remarks to the Author):

The authors seek to understand why humans develop severe disease following HTNV infection while reservoir (and laboratory) mice do not. This is an exciting area of research and the authors have done an incredible amount of work in this one manuscript. They hypothesize that inflammatory mediators released by monocytes/macrophages drive disease in humans but that this is suppressed in mice. They use in vitro hMDM and mBMDM to show that there is a downregulation of p65 later in mBMDM. They identify a transcriptomic profile of Notch signaling in association with this phenotype. Using mice with specific mutations in Notch pathways, they show that Notch signaling does indeed dampen inflammation during the late (but not early) phase of HTNV infection. They go on to identify lnc RNAs that are differentially expressed in mice but not human mBMDM and show one that is induced during HTNV infection and responsible for downregulation of inflammation via association with p65. They even generate a KO mice of the specific lnc RNA and show that this makes HTNV go from benign to about 50% lethal in mice.

I have spent the better part of two days reviewing this manuscript as it is exceptionally

complex and there is a tremendous amount of very different kinds of experiments that are performed throughout that are not well explained. Much time was spent going back and forth between the disjointed methods and the figure legends to try to figure out exactly what was done. This was not an easy read and was difficult to follow.

I'm left with the impression of (at least) 4 separate stories that each build upon the other to come to the final conclusion. In many situations these would be separate manuscripts as I've outlined below:

Paper 1: There is a nice human study from figure 1, Supp 1 & Supp 2. Showing an association of inflammation in mononuclear phagocytes with disease severity.

Paper 2: There is an important epidemiological study of the rodent reservoir in Figure 2 and Supp 3. This is a very nice part of the paper as it uses the natural reservoir mice that naturally acquired infection, and they show control of inflammation over the course of infection.

Paper 3: There are in vitro and in vivo studies in MDM and laboratory mouse models demonstrating a late anti-inflammatory state induced by HTNV (in mice but not human MDM) that is associated with loss of p65 phosphorylation and that this can confer a modest protection from LPS/bacterial sepsis.

Paper 4: The remainder of the paper is a unique tale of associating notch signaling with the phenotype, ultimately leading to the identification of a lnc RNA that is responsible for the phenotype and this is proved both in vitro and in vivo.

While the data are interesting and novel, this manuscript is not ready for publication as is.

There are several major issues that need to be addressed:

1. This manuscript requires extensive English editing for grammar.
2. Insufficient methods are provided for multiple experiments.
3. Within each section there are methodological issues with data analysis or presentation and these will be detailed below.
4. Throughout the entire manuscript there are numerous figures that are just too small for the reader to see, there is no way to assess the data if you can't visualize it.
5. It would be super helpful to the reader if you would give just a little more info about how an experiment was done ie in which cell type, model, animal etc as you describe the results. For example: Line 608 if you ended this sentence in "...at various time points in mBMDM" it would save the reader some time searching for it.
6. Passage history of the HTNV used in this study should be noted.

Human data section

1. The gating strategy shown in S1A has a log scale for what is reported as SSC-H this seems unusual as SSC and FSC are typically on a linear axis and indeed the authors have linear scales for SSC and FSC in the other subparts of S1. This needs to be checked and corrected.
2. There is nothing upstream of S1 flow plots to explain the gating strategy used. This needs to be shown
3. S1C why are total lymphocytes selected for Th2 and Th17 gating but CD4+ are selected for Treg and Th1 gating? Typically, total CD4's are selected then they are plotted on the subsequent plots to define subtypes.

4. An explanation needs to be provided for why monocytes are defined differently for each analysis- in one CD14/CD16 are used, in another CD11b/11c++ and in a third only CD11b+.
5. What is the reasoning behind comparing HNTV patients to JEV, HBV and HCV? JEV is at least an acute RNA viral infection so might be relevant, but HBV and HCV are chronic DNA infections so they seem inconsequential. Also, there is no comment on the timing of sample acquisition with respect to symptom onset for any of these viral infections. Since we know immune responses change over time during any infection this is quite pertinent.
6. Define time post sx onset for each cohort (Febrile, Hypotensive, Oliguric, Diuretic, conval) shown in Figure 1. See comment above about the importance of timing.
7. There is no description of how human blood was processed for the various flow cytometric assays. Typically, secretion inhibitors are used to capture ICS events, but this isn't mentioned. Were the cells processed and stained immediately or rested in vitro?
8. It is not clear how the cytokine data presented in Figure 1G were analyzed to cohort them into red/green/blue/black. What threshold was used to put a given cytokine in each category?
9. Line 251 "various clinical symptoms experienced by patients" are attributed to TNF and IL-8 levels- what clinical symptoms? Were they quantitated in some way to determine if they correlated with cytokine levels? Along these same lines, what criteria were used to consider patients as "severe/critical" vs mild/mod"
10. It is mentioned that the lack of detection of IFN γ in HFRS patient is evidence of lack of Th1 activity- this should be considered carefully. MIG (monokine induced by IFN γ levels), CXCL-10 and CXCL-11 were elevated in HFRS patients with severe disease, and they are all markers of IFN γ activity. Given the limited sampling of humans, the IFN peak could have been missed or occurred at a tissue level and not in the bloodstream. All humans studies are limited by the fact that only blood is routinely sampled.
11. A complicated PMBC/HUVEC/293 co-culture system is established and not well explained. Why are 293 cells even needed in this system? Can't you measure the resistance generated by the HUVEC layer in the transwell without having cells below? It also seems that what is being measured is HTNV infection in PMBCs with or without monos and you don't need transwells for this. What is meant by red/green/blue in this context in S2? It is compared to the human data in Fig 1 but red/green/blue had definitions related to disease severity which is clearly not possible in this in vitro co-culture system.
12. Line 298- "negative screening technology" used to purify monocytes- I think you mean negative selection, correct?
13. Correlation does not equal causation. This error is made repeatedly with respect to discussion of human clinical data- which is only able to show correlation/association.

Natural mouse reservoir studies

1. I cannot find a description in the methods regarding collection, manipulation or processing of field mice. Was there institutional or community level approval for this?
2. Fig 2A is too small to be read.
3. Where do the data from Figure 2Ai come from? No reference is provided for this prevalence data.
4. Why was anti-HTNV antibody measured from the lungs and not the blood of mice? What samples were collected from these animals? (see point 1 in this section)
5. Figure S3A- IHC images cannot be visualized at current magnification, a zoomed in inset of each should be included.
6. For studies in field mice, what assay was used to measure HTNV S- IHC? Antigen capture? qRT-PCR?
7. Figure 2B-please state in text what sample you are measuring cytokines in.
8. Nothing can be seen in Figure 2B. This data cannot be interpreted in print. I had to zoom

in to 300% on the PDF to see this.

Laboratory mouse model studies showing in vitro p65 differences between human and mouse and late in vivo anti-inflammatory state in mice.

1. Please state in the text what species of mice are being used in these studies. The method says all mice are BL/6 (clearly not since you use field mice too) the legend for S3 says "WT" or RIG -/- (the latter are not mentioned at all in the methods nor are the various mice used later in the paper).
2. Figure 2E/F- how do HTNV viral loads compare between these groups?
3. S3 E seems to belong on S4 given that it is the same line of experimentation with RAW/THP cells.
4. Figure S4D also not easy to see, needs zoomed in insets.
5. Figure 3 some fonts are so small as to not be legible eg 3B,D legends
6. Figure 3J- can't appreciate this pathology-too small.
7. Regarding figure sizes- Figure S5B is how all of the path should be shown- total tissue with zoom in to see the details.

Notch pathway studies

1. Again, can't see Figure 4D or E without 300% magnification.
2. Figure 4 J and K are pointless if you can't magnify the image sufficiently for the reader to see it.
3. The most important piece of the RBP-Jcko mouse data (the survival curves) and viral loads are buried in S7. Please check doses of HTNV noted in this legend. I can't read what is on the figure, but the legend has the same thing twice for Fii and Fiii. Also please show viral loads for adult mouse models. This is just as important as showing it for the neonatal model, are the adult mice that succumb doing so with high viral loads or not?
4. Again, there is no way a reader can see anything in figure S7Giii. Those spleens are mere spots when printed on a page. Eg. figure 4Q is good. I can actually see the spleen architecture.
5. Wow, human notch signaling data are also buried in S7. These data are more significant in supporting your overarching hypothesis than everything in figure 4J-R.
6. I'm not sure what I'm looking at for Fig 5C. I assume that if there is differential expression between WT and RBPJcko mice at the loci indicated with arrows, we should be able to see a difference? I don't. Maybe b/c it is too small?
7. I understand the inclusion of DENV infected mBMDM for figure 5 but a sentence explaining why it was chosen would be a nice intro to the expt. Also, I don't see it noted in the methods about his virus or its use. Likewise, no info on use of other viruses- VZV EV71, SeV, HSV...
8. I've stopped commenting on each of the super small figures but this continued to be a problem throughout.

Reply to Reviewer#1

Comments: This manuscript is based on the differential observation that HTV infection in mice is less pathogenic than in humans and the authors provide evidence that the differential activation of monocytes/macrophages in humans is responsible. In doing so they identify differential regulation of lncRNAs in mice (which may be murine specific) that underpin this phenotype.

Question-1(Q1): The first thing to say is that the manuscript is an exceptional amount of data. That said -the density of the data makes the manuscript somewhat hard to read at times. I would recommend distilling out the key data from each figure to make the paper easier to read.

Overall, I have very little to say about the approach - the story builds and utilizes powerful KO mouse models to demonstrate pathogenicity can be restored in the mouse model.

Response-1(R1): Thank you for taking time out of your busy schedule to review our manuscript. Here, I represent all the authors to express our genuine appreciation for your positive comments and constructive suggestions.

We have noticed that too much data in the main figures will make the paper somewhat hard to read, and the figure annotations are too small to recognize. To tackle this issue, we have rearranged all the figures and figure legends. The modifications are listed as followings.

(1) The crucial data in the main text have been distilled, and now each figure contains 8 to 10 small pictures.

(2) The figure annotations have been adjusted to a suitable size for reading.

(3) The main results, as well as the supplementary data, have been put in order based on one certain theme, which might be easy to understand the research.

(4) The result part has also been modified to be more concise.

We hope that all these alterations could make our study easy to read.

Q2: Indeed, I am not convinced this is HTV specific. The authors suggest other RNA viruses trigger the same response in macrophages and so to my mind HTV has revealed the differential regulation of macrophages in mouse and human. This is not a criticism per se but focuses the pathogenesis on the host response rather than the nature of the pathogen itself? This could be discussed.

R2: Thanks very much for your comment, which is highly appreciated. This is an important question that could not be neglected. To discuss this issue, we made two changes in the revised paper, which may help the readers to understand this question.

On the one hand, we emphasized that the expression of lncRNAs (22387.1, 30740.1 and 30928.1) could also be induced by other RNA viruses (e.g., SeV, VSV and EV71) but not DNA viruses (e.g., HSV-2) (Figure 5E). Notably, the expression of these lncRNAs was correlated with HTNV MOIs, as shown by qRT-PCR and Northern blot (Figure S16C). Additionally, both TLR3 agonist (polyIC) and TLR4 agonist (LPS) could promote the transcription of these lncRNAs (Figure S16D). This indicated that the alteration of these murine lncRNAs might not be HTNV-specific, but be highly associated with HTNV infection (in the result part of the revised manuscript, from line 559 to 566, shown in the yellow shadow).

On the other hand, although this phenomenon is nonspecific, these lncRNAs, especially lnc-i65, did play an important role in restraining the HTNV-caused hyperactivation of M1 macrophage and preventing the cytokine storm syndrome. Even though, it is still an intriguing question of whether lnc-i65 knockout could affect the pathogenesis of other RNA viruses, such as DENV2, VSV or EV71. And it could also not exclude the possibility that there exist some other factors specific for HTNV and determinative for the nonpathogenic feature in rodents. It is of great importance to investigate the characteristics of the pathogen itself (in the discussion part of the revised manuscript, from line 901 to 907, shown in the yellow shadow).

Q3: Experimentally, I have no additional expts to suggest - it as i say above pretty exhaustive. I think better curation of the data and presentation would greatly enhance what I think is a really interesting story. (A lot of the descriptive time course data could be put into supplementary to allow the main figs to be focused on the key data).

R3: We are grateful for your valuable comments and suggestions. As we replied in question 1, all the figures and results were retuned to be concise and to the point. We hope the revised manuscript could be easily understood.

Reply to Reviewer#2

Comments: The authors seek to understand why humans develop severe disease following HTNV infection while reservoir (and laboratory) mice do not. This is an exciting area of research and the authors have done an incredible amount of work in this one manuscript. They hypothesize that inflammatory mediators released by monocytes/macrophages drive disease in humans but that this is suppressed in mice. They use in vitro hMDM and mBMDM to show that there is a downregulation of p65 later in mBMDM. They identify a transcriptomic profile of Notch signaling in association with this phenotype. Using mice with specific mutations in Notch pathways, they show that Notch signaling does indeed dampen inflammation during the late (but not early) phase of HTNV infection. They go on to identify lnc RNAs that are differentially expressed in mice but not human mBMDM and show one that is induced during HTNV infection and responsible for downregulation of inflammation via association with p65. They even generate KO mice of the specific lnc RNA and show that this makes HTNV go from benign to about 50% lethal in mice.

I have spent the better part of two days reviewing this manuscript as it is exceptionally complex and there is a tremendous amount of very different kinds of experiments that are performed throughout that are not well explained. Much time was spent going back and forth between the disjointed methods and the figure legends to try to figure out exactly what was done. This was not an easy read and was difficult to follow.

I'm left with the impression of (at least) 4 separate stories that each build upon the other to come to the final conclusion. In many situations these would be separate manuscripts as I've outlined below:

Paper 1: There is a nice human study from figure 1, Supp 1 & Supp 2. Showing an association of inflammation in mononuclear phagocytes with disease severity.

Paper 2: There is an important epidemiological study of the rodent reservoir in Figure 2 and Supp 3. This is a very nice part of the paper as it uses the natural reservoir mice that naturally acquired infection, and they show control of inflammation over the course of infection.

Paper 3: There are in vitro and vivo studies in MDM and laboratory mouse models demonstrating a late anti-inflammatory state induced by HTNV (in mice but not human MDM) that is associated with loss of p65 phosphorylation and that this can confer a modest protection from LPS/bacterial sepsis.

Paper 4: The remainder of the paper is a unique tale of associating notch signaling with the phenotype, ultimately leading to the identification of a lnc RNA that is responsible for the phenotype and this is proved both in vitro and in vivo.

While the data are interesting and novel, this manuscript is not ready for publication as is.

General Questions:

There are several major issues that need to be addressed:

1. This manuscript requires extensive English editing for grammar.
2. Insufficient methods are provided for multiple experiments.
3. Within each section there are methodological issues with data analysis or presentation and these will be detailed below.
4. Throughout the entire manuscript there are numerous figures that are just too small for the reader to see, there is no way to assess the data if you can't visualize it.
5. It would be super helpful to the reader if you would give just a little more info about how an experiment was done ie in which cell type, model, animal etc as you describe the results. For example: Line 608 if you ended this sentence in "...at various time points in mBMDM" it would save the reader some time searching for it.
6. Passage history of the HTNV used in this study should be noted.

Responses:

Very thankful for taking your precious time to review our manuscript. Here, I represent all the authors to express our genuine appreciation for your valuable comments.

We have carefully revised the manuscript according to your constructive suggestions and did our best to reply to all the questions point by point. Detailed responses are shown as followings.

Response to Question-1:

We have sent our manuscript to the American Journal of Experts (AJE) for language polishing, and the English grammar and other mistakes have been corrected. The editing certificate was also submitted with our revised paper (the attached pdf file was entitled "editing certificate").

We hope that the revised manuscript could reach the standard in terms of the language.

Response to Question-2:

More details about the methods, including the virus information and mice experiments, were added in the revised manuscript, and we are sure now almost all the crucial experiment protocols were concluded (in the method part of the revised paper, from page 28 to 40). Additionally,

partial experiments were explained in the figure legends to make it clear (e.g., the legend for Figure S8C, from line 1707 to 1720, shown in the yellow shadow, in the revised paper). Moreover, the specific reagents were all listed in the “KEY RESOURCES TABLE”. We hope that these modifications could make the experimental design easy to understand.

Response to Question-3:

The methodological issues were handled one by one based on your suggestions, which will be explained in the following responses.

Response to Question-4:

We have noticed that too much data in the main figures will make the paper somewhat hard to read, and the figure annotations are too small to recognize. To tackle this issue, we have rearranged all the figures and figure legends. The modifications are listed as followings. (1) The crucial data in the main text have been distilled, and now each figure contains 8 to 10 small pictures. (2) The figure annotations have been adjusted to a suitable size for reading. (3) The main results, as well as the supplementary data, have been put in order based on one certain theme, which might be easy to understand the research. (4) The result part has also been modified to be more concise. We hope that all these alterations could make our study easy to read.

Response to Question-5:

Thanks for your constructive suggestion. We have modified related statements throughout the text according to the comment. according to the proposal, which makes the result part concise and consistent. In fact, we define the infection stages as we previously reported (Wang et al., 2019), which are comprised of the early (from 0 to 24 hpi) and late (from 24 to 48 hpi). To make this clear, we reserve the description of “at the early phase (from 0 to 24 hpi)” or “at the late phase (from 0 to 24 hpi)” in case of misinterpretation.

Response to Question-6:

Passage history of the HTNV was explained in the revised manuscript (line 1042 to 1045, “HTNV strain 76-118 (24th generation) …”, in the revised paper).

Specific Questions

Human data section

1. The gating strategy shown in S1A has a log scale for what is reported as SSC-H this seems

unusual as SSC and FSC are typically on a linear axis and indeed the authors have linear scales for SSC and FSC in the other subparts of S1. This needs to be checked and corrected.

Response-1: We are extremely grateful to you for pointing out this problem, and this mistake has been corrected in the revised article (Figure S1A) as the following shown. And we have checked the raw data, and found the “log” scale annotation in the original pictures was caused by the default analysis of the FlowJo V10 when it read the data from the BD FACSCalibur Machine. The SSC-H scale has been adjusted with the linear instead of the log unit.

2. There is nothing upstream of S1 flow plots to explain the gating strategy used. This needs to be shown.

Response-2: Thanks very much for your comment. We are not sure whether we get the point, for that generally the SSC (side scatter measures the subcellular components) and FSC (forward scatter measures the cell size) analysis were the initial process to select the target cells. In terms of the gating strategy for M1- or M2-like monocytes, the sketch map was shown on the left of the flow cytometry results (Figure S1A) as shown in the following.

3. S1C why are total lymphocytes selected for Th2 and Th17 gating but CD4+ are selected for Treg and Th1 gating? Typically, total CD4's are selected then they are plotted on the subsequent plots to define subtypes.

Response-3: Thank you for the suggestion. The layout of the original Figure S1C was not suitable and caused a misunderstanding. This has been modified in the revised Figure S1D as shown in the following. CD4+ T cells are selected for Th1, Th2 and Treg subset gating.

4. An explanation needs to be provided for why monocytes are defined differently for each analysis- in one CD14/CD16 are used, in another CD11b/11c++ and in a third only CD11b+.

Response-4: This is a valuable question to discuss that we did not explain in the original article.

Complement receptors CR3 (CD11b/CD18) and CR4 (CD11c/CD18) belong to the family of beta2 integrins and are expressed mainly in human monocytes (Sándor et al., 2016). CD11b+ is relatively sufficient to target the monocytes in PBMCs according to current research. However, it should be noted the expression of CD11b on monocytes could be altered under viral infection conditions (Mairpady Shambat et al., 2022). To exclude such influences, CD11b+ and CD11c+ are mostly used to identify monocytes in our study. CD14 and CD16 are used to directly analyze the M1- and M2-like monocytes from the PBMCs as previously reported (Cole et al., 2017). This gating strategy was emphasized in the revised paper (line 196 to 199, shown in the yellow shadow, in the revised paper. “Previous studies have shown that M1-like (marked by CD14⁺ CD16⁺) or M2-like (marked by CD14⁺ CD16⁺⁺) monocytes are the major immunological determinants for life-threatening influenza or chronic viral hepatitis, respectively (Cole et al.,

2017; Saha et al., 2016), but whether monocyte activation patterns affect HFRS pathogenesis is unclear.”).

5. What is the reasoning behind comparing HNTV patients to JEV, HBV and HCV? JEV is at least an acute RNA viral infection so might be relevant, but HBV and HCV are chronic DNA infections so they seem inconsequential. Also, there is no comment on the timing of sample acquisition with respect to symptom onset for any of these viral infections. Since we know immune responses change over time during any infection this is quite pertinent.

6. Define time post sx onset for each cohort (Febrile, Hypotensive, Oliguric, Diuretic, conval) shown in Figure 1. See comment above about the importance of timing.

Response-5 and -6: Thank you for your precious comments and advice.

(1) As for the comparison of HTNV infection with other viruses:

Firstly, in terms of the infection features, JEV was used as a control for acute and HBV or HCV was used for chronic infection.

Secondly, in terms of the role of monocytes on the pathogenesis process post infection, previous studies have shown that M1-like (marked by CD14⁺⁺ CD16⁺) or M2-like (marked by CD14⁺ CD16⁺⁺) monocytes are the major immunological determinants for life-threatening influenza or chronic viral hepatitis, respectively (Cole et al., 2017; Saha et al., 2016), but whether monocyte activation patterns affect HFRS pathogenesis is unclear. To narrow this gap, peripheral blood mononuclear cells (PBMCs) from patients with distinct viral infections were collected, and the monocyte subset was examined and analysed.

Thirdly, there also existed the objective limitation, that is only these patients' samples were accessible for our research.

This part has been fully interpreted in our revised paper (line 196 to 201, shown in the yellow shadow, in the revised paper).

(2) As for the timing of sample acquisition:

In Figure 1A and S1A, HFRS patients, the febrile phase, n=12; Japanese encephalitis patients, the febrile phase, n=15; hepatitis B patients, confirmed chronic infection for more than one-year, inactive phase without liver cirrhosis and antiviral therapy, n=18; hepatitis C patients, confirmed chronic infection for more than one-year, inactive phase without liver cirrhosis and antiviral therapy, n=18. This has been explained in the revised manuscript (line 949 to 955, shown in the yellow shadow, in the revised paper).

For the infection stage of HTNV, we have defined the time post six onsets of HFRS for each cohort in the revised text, and to make this clear, a schematic diagram was added to Figure 1 (Figure 1B) as shown in the following. In general, samples were collected at 3–6 days for the febrile or hypotension stage, 7–12 days for the oliguric stage, 13–18 days for the diuretic stage and after 18 days for the convalescent stage. The phase within 8 days from the fever onset to the early oliguric stage was typically defined as acute or early phase of the disease. Details for classification of disease severity and exclusion criteria were previously described (Yi et al., 2013; Zhang et al., 2015). Additionally, to control for potential confounders, we excluded HFRS patients with autoimmune diseases, viral hepatitis, hematological diseases, diabetes, cardiovascular diseases, and other kidney or liver diseases. This has been explained in the revised manuscript (line 960 to 966, shown in the yellow shadow, in the revised paper).

7. There is no description of how human blood was processed for the various flow cytometric assays. Typically, secretion inhibitors are used to capture ICS events, but this isn't mentioned. Were the cells processed and stained immediately or rested *in vitro*?

Response-7: Thank you for this technique question. The monocytes or T cells from HFRS patients underwent intracellular cytokine staining immediately, while for the *in vitro* experiments, the secretion inhibitors are used to precisely measure cytokine production. This has been explained in the revised manuscript (line 1148 to 1149, shown in the yellow shadow, in the revised paper).

8. It is not clear how the cytokine data presented in Figure 1G were analyzed to cohort them into red/green/blue/black. What threshold was used to put a given cytokine in each category?

Response-8: Our deepest gratitude goes to you for your careful work and thoughtful suggestions

Similarly, in Figure S3D, related issues have also been explained as followings:

“Upregulated cytokines post HTNV infection were marked with colourful labelling (red/green/blue), among which the downregulated (labelled with red), upregulated (labelled with green), and unchanged (labelled with blue) cytokines in the monocyte-removed PBMC group compared with the whole PBMC group were labelled distinctively. Black labelling represents the downregulated cytokines post HTNV infection” (in the revised manuscript, figure legends, line 1631 to 1636, shown in the yellow shadow, in the revised paper).

9. Line 251 “various clinical symptoms experienced by patients” are attributed to TNF and IL-8 levels- what clinical symptoms? Were they quantitated in some way to determine if they correlated with cytokine levels? Along these same lines, what criteria were used to consider patients as “severe/critical” vs mild/mod”

Response-9: Very thankful for your question.

Indeed, it was speculated that “various clinical symptoms experienced by patients” are attributed to TNF and IL-8 levels according to previous researches (Angulo et al., 2017; Li et al., 2018; Saksida et al., 2011). Related descriptions have been deleted in the revised manuscript in case of being misunderstood.

As for the criteria, mild HFRS was defined as mild renal failure without an obvious oliguric stage and moderate for obvious symptoms of uremia, effusion (bulbar conjunctiva), hemorrhage (skin and mucous membrane), and renal failure with a typical oliguric stage. While patients with severe uremia, effusion (bulbar conjunctiva and either peritoneum or pleura), hemorrhage (skin and mucous membrane), and renal failure with oliguria (urine output, 50–500 mL/day) for ≤ 5 days or anuria (urine output, < 50 mL/day) for ≤ 2 days were defined as severe HFRS, and critical patients were considered as those with ≥ 1 of the following symptoms: refractory shock, visceral hemorrhage, heart failure, pulmonary edema, brain edema, severe secondary infection, and severe renal failure with either oliguria (urine output, 50–500 mL/day) for > 5 days, anuria (urine output, < 50 mL/day) for > 2 days, or a blood urea nitrogen level of > 42.84 mmol/L. Details for classification of disease severity and exclusion criteria were previously described (Yi et al., 2013; Zhang et al., 2015). This has been explained in the revised manuscript (line 960 to 961, shown in the yellow shadow, in the revised paper).

10. It is mentioned that the lack of detection of IFN γ in HFRS patient is evidence of lack of Th1 activity- this should be considered carefully. MIG (cytokine induced by IFN γ levels),

CXCL-10 and CXCL-11 were elevated in HFRS patients with severe disease, and they are all markers of IFN γ activity. Given the limited sampling of humans, the IFN peak could have been missed or occurred at a tissue level and not in the bloodstream. All humans studies are limited by the fact that only blood is routinely sampled.

Response-10: Very thankful for your question. Related descriptions about “the lack of detection of IFN γ in HFRS patients is evidence of lack of Th1 activity” were deleted in the revised manuscript. Indeed, this might be caused by the limitation of the sample number as you mentioned.

11. A complicated PMBC/HUVEC/293 co-culture system is established and not well explained. Why are 293 cells even needed in this system? Can't you measure the resistance generated by the HUVEC layer in the transwell without having cells below? It also seems that what is being measured is HTNV infection in PMBCs with or without monos and you don't need transwell for this. What is meant by red/green/blue in this context in S2? It is compared to the human data in Fig 1 but red/green/blue had definitions related to disease severity which is clearly not possible in this in vitro co-culture system.

Response-11: Thanks for your question. This coculture system was applied for the crosstalk between HUVECs and HEK293 cells might mimic the *in vivo* infection environment of the kidney tissue, as we used in our recently published article (Yang et al., 2022). Of course, the TER of HUVECs could be detected based on the transwell system without HEK293, but this could not reflect the real infection situation as the cytokines secreted by tissue cells also matter a lot in activating immune responses, which might affect the vascular permeability post HTNV infection. As for the heatmap of cytokine concentration (Figure S3D in the revised paper), this issue has been clearly explained in the **Response-8**.

12. Line 298- “negative screening technology” used to purify monocytes- I think you mean negative selection, correct?

Response-12: Very thankful for your kind help in correcting the description. Yes, it means the “negative selection”, and this has been modified in the revised manuscript (line 251, shown in the yellow shadow). Other parts of this article have been carefully checked, and related description has also been adjusted (e.g., “negative screening beads” was rectified as “negative selection beads” in line 971, shown in the yellow shadow, in the revised paper).

13. Correlation does not equal causation. This error is made repeatedly with respect to

discussion of human clinical data- which is only able to show correlation/association.

Response-13: Very thankful for your kind help in pointing out this error. In the revised paper, all the related descriptions have been rectified with the words “hinting that it might be ……were correlated with……” and “might be involved in host immune disorder”. Neutral narration is more rigorous (line 214 and 234, shown in the yellow shadow, in the revised paper).

Specific Questions

Natural mouse reservoir studies

1. I cannot find a description in the methods regarding collection, manipulation or processing of field mice. Was there institutional or community level approval for this?

Response-1: Very thankful for the reminder. The epidemiological data of the field mice were collected from the Shaanxi Provincial Notifiable Disease Surveillance System in collaboration with the research team of Pengbo Yu, which was authorized by the government (Tian et al., 2017). For the capture, traps were placed outdoors (set as 4 parallel lines of 25 traps each and spaced intervals of 5 meters). Rodents were removed from the traps once captured for further investigation. According to the standard measurement for HTNV infection in rodents (Tian and Stenseth, 2019; Tian et al., 2015; Xiao et al., 2018), the lung tissues were acquired for analysis. This has been explained in the revised manuscript (line 977 to 982, shown in the yellow shadow, in the revised paper). And We thanked the research team of Pengbo Yu in the ACKNOWLEDGMENTS (line 923, shown in the yellow shadow, in the revised paper).

2. Fig 2A is too small to be read.

Response-2: Very thankful for your question. We have rearranged all the figures and figure legends, and this picture has been enlarged in the revised manuscript for clear presentation (for details, see above, the Response to Question-4 in the general question part). We hope this could settle your question. Here are the revised picture of Figure 2A.

3. Where do the data from Figure 2Ai come from? No reference is provided for this prevalence data.

Response-3: Very thankful for the reminder. Figure 2A-i showed a Geographic heatmap depicting HFRS incidence in China from 2016 to 2017, and the epidemiological data of HFRS in China were acquired from the Chinese Center for Disease Control and Prevention (CDCDC) (Available at: <http://www.stats.gov.cn/>). This has been explained in the revised manuscript (line 1377 to 1379, shown in the yellow shadow, in the revised paper). The raw data were shown as followings (The incidence and mortality of HFRS in China from 2016 to 2017).

地区	2016 年统计结果				2017 年统计结果			
	发病数	死亡数	发病率	死亡率	发病数	死亡数	发病率	死亡率/
全 国	8853	48	0.6458	0.0035	11262	64	0.8162	0.0046
北京市	9	0	0.0415		7	0	0.0322	
天津市	31	0	0.2004		32	0	0.2048	
河北省	434	1	0.5845	0.0013	447	0	0.5983	
山西省	8	0	0.0218		26	2	0.0706	0.0054
内蒙古	106	1	0.4221	0.004	93	0	0.369	
辽宁省	863	3	1.9692	0.0068	1078	3	2.4624	0.0069
吉林省	515	2	1.8705	0.0073	574	3	2.1003	0.011
黑龙江	1200	2	3.148	0.0052	1119	2	2.9455	0.0053
上海市	4	0	0.0166		2	0	0.0083	
江苏省	330	8	0.4137	0.01	375	3	0.4688	0.0038
浙江省	345	0	0.6229		351	1	0.6279	0.0018
安徽省	207	0	0.3369		273	2	0.4406	0.0032
福建省	366	0	0.9534		388	1	1.0015	0.0026
江西省	676	5	1.4806	0.011	571	1	1.2434	0.0022
山东省	985	12	1.0003	0.0122	1252	10	1.2587	0.0101
河南省	176	1	0.1857	0.0011	435	11	0.4563	0.0115
湖北省	236	5	0.4033	0.0085	505	6	0.8581	0.0102
湖南省	588	3	0.8678	0.0044	632	4	0.9264	0.0059
广东省	410	3	0.3779	0.0028	443	0	0.4028	
广 西	5	0	0.0104		14	0	0.0289	
海南省	3	0	0.0329		3	0	0.0327	
重庆市	13	0	0.0431		5	0	0.0164	
四川省	110	1	0.1341	0.0012	260	1	0.3147	0.0012
贵州省	59	0	0.1672		58	0	0.1632	
云南省	216	0	0.4555		229	0	0.48	
西 藏	1	0	0.0309		1	0	0.0302	
陕西省	933	1	2.4599	0.0026	2064	14	5.4129	0.0367
甘肃省	19	0	0.0731		20	0	0.0766	
宁 夏	4	0	0.0599		2	0	0.0296	
新疆	1	0	0.0042		3	0	0.0125	
台湾省	4	0	0.0056		0	0		

4. Why was anti-HTNV antibody measured from the lungs and not the blood of mice? What samples were collected from these animals? (see point 1 in this section)

Response-4: Very thankful for your question. On the one hand, as we mentioned above, according the standard measurement for HTNV infection in rodents (Tian and Stenseth, 2019;

Tian et al., 2015; Xiao et al., 2018), the lung tissues were acquired for analysis (line 977 to 982). On the other hand, as we explained in the result part, “indeed, we found that *A. agrarius* mouse lung tissue was more susceptible to HTNV infection than liver or kidney (Figure S4A), which was consistent with previous studies (Kim et al., 2016; No et al., 2019) and ensured the authenticity of our testing system” (line 267 to 269, shown in the yellow shadow, in the revised paper). All these alterations have been updated in the revised manuscript.

5. Figure S3A- IHC images cannot be visualized at current magnification, a zoomed in inset of each should be included.

Response-5: Very thankful for your suggestion. This figure has been adjusted in the revised paper (Figure S4A), which means a zoomed in inset of each is included as the following shown.

6. For studies in field mice, what assay was used to measure HTNV S- IHC? Antigen capture? qRT-PCR?

Response-6: Very thankful for your question. qRT-PCR was used to measure HTNV S in the rodent lung tissue. This has been clearly explained in the revised manuscript:

“Natural infection phases of HTNV in *A. agrarius* mice defined based on the assessment of HTNV-S (qRT-PCR) and anti-NP IgG (ELISA) in mouse lung tissue” (legends of Figure 2A-ii, line 1381 to 1382, shown in the yellow shadow, in the revised paper).

7. Figure 2B-please state in text what sample you are measuring cytokines in.

Response-7: Very thankful for your question. The samples were the lung tissues as we explained above, and this was emphasized in the revised manuscript:

(1) “Elevated production of six inflammatory cytokines from the lung tissue was observed in ……” (the result part, line 270, shown in the yellow shadow, in the revised paper);

(2) “Cytokine production measured by ELISA in *A. agrarius* mouse lung tissue (n=20 in each group)” (the figure legends, line 1383, shown in the yellow shadow, in the revised paper);

8. Nothing can be seen in Figure 2B. This data cannot be interpreted in print. I had to zoom in to 300% on the PDF to see this.

Response-8: Very thankful for your advice. This picture has been zoomed in for clear presentation in the revised article (Figure 2B, as shown in the following). We hope this could settle your question.

Specific Questions

Laboratory mouse model studies showing in vitro p65 differences between human and mouse and late in vivo anti-inflammatory state in mice.

1. Please state in the text what species of mice are being used in these studies. The methods say all mice are BL/6 (clearly not since you use field mice too) the legend for S3 says “WT” or

RIG^{-/-} (the latter are not mentioned at all in the methods nor are the various mice used later in the paper).

Response-1: Thanks for providing the precious suggestion. Except for the *A. agrarius* captured from the field (also termed as the field mice), the strain of mice in this study, including the wild type or transgenic mice, is BL/6J. RIG-I deficient (RIG-I^{-/-}) C57BL/6J mice were constructed and identified at the Experimental Animal Center of Air Force Medical University. These have been emphasized in the revised manuscript (the methods part, line 982 to 984; also listed in the KEY RESOURCES TABLE, shown in the yellow shadow, in the revised paper)

2. Figure 2E/F- how do HTNV viral loads compare between these groups?

Response-2: Thanks for providing valuable advice. We have accomplished the experiments to compare the viral loads between different groups. **Figure R2** shown in the following represents the absolute quantitative PCR standard curve for quantifying the copies of HTNV S segments, the process of which was previously reported (Yi et al., 2013). We found that viral loads increased along with infection progress, while mPMφs and hMo seemed to be more susceptible to HTNV replication (Figure 2F as shown in the following). These have been added to the revised manuscript.

Figure R2. Preparation of HTNV-S standard and absolute quantitative PCR

Figure 2F. Viral Load Comparison between Different Macrophages

3. S3 E seems to belong on S4 given that it is the same line of experimentation with RAW/THP cells.

Response-3: Thanks for providing valuable advice. S3E did belong to the S4, and these figures were orderly organized in our revised manuscript (original S3E---revised S5A-i, original S4---revised S5).

4. Figure S4D also not easy to see, needs zoomed in insets.

Response-4: Thanks for your valuable suggestion.

Very thankful for your question. We have rearranged all the figures and figure legends, and this picture has been enlarged in the revised manuscript for clear presentation (for details, see above, the Response to Question-4 in the general question part). We hope this could settle your question. Here is the revised picture of Figure S5F (that was the original S4D).

5. Figure 3 some fonts are so small as to not be legible eg 3B,D legends

Response-5: Thanks for your kind help.

We have rearranged all the figures and figure legends, and this picture has been enlarged in the revised manuscript for clear presentation (for details, see above, the Response to Question-4 in the general question part). We hope this could settle your question. Here are the revised picture of Figure 3B, as shown in the following, and the original Figure 3D was rearranged to the Figure S6D in the revised paper.

6. Figure 3J- can't appreciate this pathology-too small.

Response-6: Thanks for your valuable advice.

We have rearranged all the figures and figure legends, and this picture has been enlarged in the revised manuscript for clear presentation (for details, see above, the Response to Question-4 in the general question part). We hope this could settle your question. Here are the revised picture of Figure 3G (that was original Figure 3J).

7. Regarding figure sizes- Figure S5B is how all of the path should be shown- total tissue with zoom in to see the details.

Response-7: Thanks for your valuable advice.

Based the model shown as the original Figure S5B, we have rearranged all the figures and figure legends, and this picture has been enlarged in the revised manuscript for clear presentation (for details, see above, the Response to Question-4 in the general question part). We hope this could settle your question.

The original Figure S5B was adjusted to be clear and compact as shown in the following, which was titled as Figure S7B in the revised article.

Specific Questions

Notch pathway studies

1. Again, can't see Figure 4D or E without 300% magnification.

Response-1: Thanks for your valuable advice.

Figure 4D and 4E were adjusted in the revised Figure S9D and S10A, respectively, which were shown in the followings. Additionally, we have rearranged all the figures and figure legends, and these pictures have been enlarged in the revised manuscript for clear presentation (for details, see above, the Response to Question-4 in the general question part). We hope this could settle your question.

(ii)

Nucleus + NICD

Nucleus + F4/80

2. Figure 4 J and K are pointless if you can't magnify the image sufficiently for the reader to see it.

Response-2: Thanks for your valuable advice.

Related figures that represent the macrophage functions, including Figure 4J and K, were

independently displayed as Figure S11 in the revised paper, which were shown in the followings. Additionally, we have rearranged all the figures and figure legends, and these pictures have been enlarged in the revised manuscript for clear presentation (for details, see above, the Response to Question-4 in the general question part). We hope this could settle your question.

Figure S11. The Notch Pathway Contributes to the Late-inactivation of Inflammatory Macrophages for Murine, Related to Figure 4

(A) The mRNA measurement of WT and RBP-J^{CKO} mBMDMs at 36 hpi by qRT-PCR for the M1 or M2 related genes (MOI=1, n=4 in each group).

(B) Representative flow cytometry histograms (right) and their statistical data (left) of the WT and RBP-J^{CKO} mBMDMs showing the alteration of CD80/86 expression (36 hpi, MOI=1, n=4 in each group).

(C) The phagocytic capacity of WT and RBP-J^{CKO} mBMDMs was evaluated by calculating the percentage of FAM⁺ cells (n=4 in each group). Scale bars, 100 μ m.

(D) The chemotaxis ability of WT and RBP-J^{CKO} mBMDMs evaluated by calculating the infiltrated cells with transwell experiments (n=4 in each group). Scale bars, 100 μ m.

(E) iNOS mRNA expression detected by qRT-PCR and cellular ROS production examined by H2DCFDA staining (n=4 in each group).

(F) Metabolic phenotype shift (i), ECAR (ii) and OCR (iii) assays by Seahorse for the WT and RBP-J^{CKO} mBMDMs from 0 to 36 hpi (MOI=1, n=4 in each group).

(G) The mitochondrial morphology of WT and RBP-J^{CKO} mBMDMs evaluated with transmission electron microscopy (TEM). Left, TEM imaging; Right, calculation for the mitochondrial number and related damage percentage of each cell (25 cells were computed for each group).

(H) Serum cytokine concentration (ELISA) (i) and viral load (qRT-PCR) (ii) at 7 dpi were measured by ELISA or qRT-PCR, respectively, from Figure 4E (n=4 in each group).

(I) Serum TNF α concentration (ELISA) (i) and viral load (qRT-PCR) (ii) at 3 dpi or 7 dpi from Figure 4F (i.p., 8×10^5 TCID₅₀/g, n=4 in each group).

Data are shown as the mean \pm SEM and are representative of three independent experiments. Analysis was performed using the unpaired Student's *t* test. * $p < 0.05$, ** $p < 0.01$, *** $p < 0.001$.

3. The most important piece of the RBP-Jcko mouse data (the survival curves) and viral loads are buried in S7. Please check doses of HTNV noted in this legend. I can't read what is on the figure, but the legend has the same thing twice for Fii and Fiii. Also please show viral loads for adult mouse models. This is just as important as showing it for the neonatal model, are the adult mice that succumb doing so with high viral loads or not?

Response-3: Thanks for your constructive suggestion. Three points were listed:

(1) The RBP-Jcko mouse data (the survival curves) were organized in Figure 4 (as shown in the following) in the revised paper, and other results were rearranged in Figure S11 (as shown above, Response-2);

(2) The HTNV challenging doses were different, which was emphasized and corrected in the revised paper:

① “To evaluate whether this process was beneficial *in vivo*, neonatal and adult mouse models were utilized. RBP-J^{CKO} suckling mice showed an early onset of disease than the WT mice (Figure 4E), which was associated with more severe inflammatory responses but not with the viral load (Figure S11H). Although there was no significant difference between the survival curves at low dosages of HTNV, a significant collapse in survival rate was found in the RBP-J^{CKO} group when challenged with higher viral dosages (Figure 4F)” (the result part, line 494 to 499, shown in the yellow shadow, in the revised paper).

②“(E) Survival data for HTNV-challenged 4-day neonatal mice in the WT or RBP^{CKO} group (i.p., 8×10^5 TCID₅₀/g (i). (F) Survival data for 8-week-old adult mice in the WT or RBP^{CKO} group with increasing HTNV challenge doses (i.p., 8×10^5 TCID₅₀/g, 8×10^6 TCID₅₀/g, or 8×10^7 TCID₅₀/g)” (the figure legends, line 1436 to 1439, shown in the yellow shadow, in the revised paper).

(3) Detection of serum viral loads for adult mouse models had been performed, and we found

that significantly increased hantaviral replication at 7 dpi (Figure S11I) in the RBP-J^{CKO} group, as shown in the following. This indicated that the dysfunction of macrophages in RBP-J^{CKO} mice could contribute to the aggravated immune damage as well as delayed clearance of viruses. These data were added to the revised manuscript (line 501 to 502, shown in the yellow shadow, in the revised paper).

4. Again, there is no way a reader can see anything in figure S7Giii. Those spleens are mere spots when printed on a page. Eg. figure 4Q is good. I can actually see the spleen architecture.

Response-4: Thanks for your valuable advice.

Figure S7Giii and 4Q were adjusted in the revised Figure 4H and 4I, respectively, which were shown in the followings. Additionally, we have rearranged all the figures and figure legends, and these pictures have been enlarged in the revised manuscript for clear presentation (for details, see above, the Response to Question-4 in the general question part). We hope this could settle your question.

G**H****(i)****(ii)**
5. Wow, human notch signaling data are also buried in S7. These data are more significant in supporting your overarching hypothesis than everything in figure 4J-R.

Response-5: Thanks for your valuable advice.

The human notch signaling data were important, and we have rearranged these results in Figure S12 and S13 (shown as the followings) in the revised paper. Considering the limitations of figures in the main text, these data had been organized in the supplementary part. Of course, this is of great significance, and we have emphasized it in the main text:

“Last, another noteworthy question was raised, namely, whether such an activation pattern and related regulatory model upon HTNV infection in mice also functioned in human beings. Most Notch signalling-related genes and proteins increased post HTNV infection in hMDMs from 0 to 36 hpi (Figure S12A and S12B), during which the NICD was continuously generated and translocated into the nucleus (Figure S12C and S12D). These results indicated that the Notch pathway was completely activated in human macrophages throughout the infection stage. Hindering NICD generation with DAPT could specifically constrain the phosphorylation of p65 rather than p-JNK or p-ERK, which would also facilitate HTNV replication at the late infection stage (36 to 48 hpi) (Figure S12E), suggesting that Notch signalling might consolidate the human M1 polarization process. Consistently, DAPT restrained the secretion of various proinflammatory cytokines at 48 hpi (Figure S12F), during which the expression of manifold M1-related genes was downregulated while M2-related genes were strengthened (Figure S12G and S12H). Furthermore, we found that the activation level of Notch signalling in monocytes was associated with disease severity (Figure S13). These data collectively demonstrated that Notch signalling showed a distinct activation pattern in humans versus mice, which exerted opposite effects on the macrophage reprogramming process” (the result part, line 509 to 522, shown in the yellow shadow, in the revised paper).

Figure S12. The Notch Pathway Regulates the Activation of Inflammatory Macrophages for Humans, Related to Figure 4

(A) qRT-PCR analysis of Notch pathway-related genes in HTNV-infected hMDMs (MOI=1).

(B) Immunoblot analysis of the Notch pathway in HTNV-infected hMDMs (MOI=1).

(C) Immunoblot analysis of the NICD amount in the cytoplasm or nucleus in HTNV-infected mBMDMs (MOI=1).

(D) Immunofluorescent analysis of NICD localization in HTNV-infected hMDMs (MOI=1).

(E) Immunoblot analysis of phosphorylation of p65, ERK or JNK in HTNV-infected hMDMs that were pretreated with DMSO or DAPT (50 μ mol/L) for 12 hr.

(F) Supernatant cytokine concentration from mock or HTNV-infected hMDMs, (MOI=1) at 48 hpi, which were treated with DMSO or DAPT (50 μ mol/L) at 12 hr before the virus challenge.

(G and H) qRT-PCR analysis of M1- (G) or M2-related (H) gene expression in hMDMs from (F).

Data are shown as the mean \pm SEM and are representative of three independent experiments. Analysis was performed using the unpaired Student's *t* test. * $p < 0.05$, ** $p < 0.01$, *** $p < 0.001$.

Figure S13. The Expression of Notch Pathway-related Genes Measured by qRT-PCR in Monocytes Correlates with the HFRS Severity Related to Figure 4

The expression levels of Notch receptors, ligands and target genes in monocytes from healthy individuals and HFRS patients with various disease severity were evaluated by qRT-PCR. Data are shown as the median, quartile, and standard deviation. Analysis was performed using the unpaired Student's *t* test. * $p < 0.05$, ** $p < 0.01$, *** $p < 0.001$; ns, no significance.

6. I'm not sure what I'm looking at for Fig 5C. I assume that if there is differential expression between WT and RBPJcko mice at the loci indicated with arrows, we should be able to see a difference? I don't. Maybe b/c it is too small?

Response-6: Thank you for your precious comments and advice.

Considering that Fig 5C could not clearly display the lncRNA alteration on different chromosomes (tremendous background gene expression would make it hard to recognize the changes of several lncRNAs) and might lead to the misconception, we have deleted these data in the revised manuscript.

7. I understand the inclusion of DENV infected mBMDM for figure 5 but a sentence explaining why it was chosen would be a nice intro to the expt. Also, I don't see it noted in the methods about his virus or its use. Likewise, no info on use of other viruses- VZV EV71, SeV, HSV...

Response-7: Thank you for this suggestion.

DENV2 also could infect mice but not induce clinical symptoms, which might share some similar characteristics with HTNV during the infection process. The brief introduction was used in the revised manuscript (line 534 to 536, shown in the yellow shadow, in the revised paper).

The information of other viruses was added in the methods part (line 1042 to 1051, shown in the yellow shadow, in the revised paper):

▪ ***Viral Infection***

HTNV strain 76-118 (24th generation), as well as DENV2 (31th generation), SeV (12th generation), VSV (17th generation), EV71 (21th generation) and HSV-2 (9th generation), was propagated in mice brain and then in Vero E6 cells as we previously described (Ma et al., 2017). The infection of HTNV, DENV2, SeV, VSV, EV71 and HSV-2 was performed as previously described (Cao et al., 2022; Chen et al., 2022; Han et al., 2017; Ma et al., 2017; Tessema et al., 2022; Ye et al., 2020). In brief, Cultured cells were infected with the indicated multiplicities of infection (MOI) of HTNV or other viruses. After 2 hr, the virus-containing medium was discarded, and the cells were washed thoroughly with sterile medium and replaced with culture medium. As a control, cells were incubated with culture supernatant from uninfected Vero E6 cells, which were referred to as mock-infected cells.

8. I've stopped commenting on each of the super small figures but this continued to be a problem throughout.

Response-8: Thank you for sparing your precious time to review our manuscript.

We have rearranged all the figures and figure legends, and these pictures have been enlarged in the revised manuscript for clear presentation (for details, see above, the Response to Question-4 in the general question part). We hope this could settle your question.

References

- Angulo, J., Martínez-Valdebenito, C., Marco, C., Galeno, H., Villagra, E., Vera, L., Lagos, N., Becerra, N., Mora, J., Bermúdez, A., *et al.* (2017). Serum levels of interleukin-6 are linked to the severity of the disease caused by Andes Virus. *PLoS Negl Trop Dis* *11*, e0005757.
- Cao, L., Luo, Y., Guo, X., Liu, S., Li, S., Li, J., Zhang, Z., Zhao, Y., Zhang, Q., Gao, F., *et al.* (2022). SAFA facilitates chromatin opening of immune genes through interacting with anti-viral host RNAs. *PLoS Pathog* *18*, e1010599.
- Chen, J., Li, Z., Guo, J., Xu, S., Zhou, J., Chen, Q., Tong, X., Wang, D., Peng, G., Fang, L., *et al.* (2022). SARS-CoV-2 nsp5 Exhibits Stronger Catalytic Activity and Interferon Antagonism than Its SARS-CoV Ortholog. *J Virol* *96*, e0003722.
- Cole, S.L., Dunning, J., Kok, W.L., Benam, K.H., Benlahrech, A., Repapi, E., Martinez, F.O., Drumright, L., Powell, T.J., Bennett, M., *et al.* (2017). M1-like monocytes are a major immunological determinant of severity in previously healthy adults with life-threatening influenza. *JCI Insight* *2*, e91868.
- Han, P., Ye, W., Lv, X., Ma, H., Weng, D., Dong, Y., Cheng, L., Chen, H., Zhang, L., Xu, Z., *et al.* (2017). DDX50 inhibits the replication of dengue virus 2 by upregulating IFN- β production. *Arch Virol* *162*, 1487-1494.
- Kim, J.A., Kim, W.K., No, J.S., Lee, S.H., Lee, S.Y., Kim, J.H., Kho, J.H., Lee, D., Song, D.H., Gu, S.H., *et al.* (2016). Genetic Diversity and Reassortment of Hantaan Virus Tripartite RNA Genomes in Nature, the Republic of Korea. *PLoS Negl Trop Dis* *10*, e0004650.
- Li, X., Du, N., Xu, G., Zhang, P., Dang, R., Jiang, Y., and Zhang, K. (2018). Expression of CD206 and CD163 on intermediate CD14(++)CD16(+) monocytes are increased in hemorrhagic fever with renal syndrome and are correlated with disease severity. *Virus Res* *253*, 92-102.
- Ma, H., Han, P., Ye, W., Chen, H., Zheng, X., Cheng, L., Zhang, L., Yu, L., Wu, X., Xu, Z., *et al.* (2017). The Long Noncoding RNA NEAT1 Exerts Antihantaviral Effects by Acting as Positive Feedback for RIG-I Signaling. *J Virol* *91*.
- Mairpady Shambat, S., Gómez-Mejía, A., Schweizer, T.A., Huemer, M., Chang, C.C., Acevedo, C., Bergada-Pijuan, J., Vulin, C., Hofmaenner, D.A., Scheier, T.C., *et al.* (2022). Hyperinflammatory environment drives dysfunctional myeloid cell effector response to bacterial challenge in COVID-19. *PLoS Pathog* *18*, e1010176.
- No, J.S., Kim, W.K., Cho, S., Lee, S.H., Kim, J.A., Lee, D., Song, D.H., Gu, S.H., Jeong, S.T., Wiley, M.R., *et al.* (2019). Comparison of targeted next-generation sequencing for whole-genome sequencing of Hantaan orthohantavirus in Apodemus agrarius lung tissues. *Sci Rep* *9*, 16631.
- Saha, B., Kodys, K., and Szabo, G. (2016). Hepatitis C Virus-Induced Monocyte Differentiation Into Polarized M2 Macrophages Promotes Stellate Cell Activation via TGF- β . *Cell Mol Gastroenterol Hepatol* *2*, 302-316 e308.

Saksida, A., Wraber, B., and Avšič-Županc, T. (2011). Serum levels of inflammatory and regulatory cytokines in patients with hemorrhagic fever with renal syndrome. *BMC Infect Dis* *11*, 142.

Sándor, N., Lukácsi, S., Ungai-Salánki, R., Orgován, N., Szabó, B., Horváth, R., Erdei, A., and Bajtay, Z. (2016). CD11c/CD18 Dominates Adhesion of Human Monocytes, Macrophages and Dendritic Cells over CD11b/CD18. *PLoS One* *11*, e0163120.

Tessema, M.B., Farrukee, R., Andoniou, C.E., Degli-Esposti, M.A., Oates, C.V., Barnes, J.B., Wakim, L.M., Brooks, A.G., Londrigan, S.L., and Reading, P.C. (2022). Mouse Mx1 Inhibits Herpes Simplex Virus Type 1 Genomic Replication and Late Gene Expression In Vitro and Prevents Lesion Formation in the Mouse Zosteriform Model. *J Virol*, e0041922.

Tian, H., and Stenseth, N.C. (2019). The ecological dynamics of hantavirus diseases: From environmental variability to disease prevention largely based on data from China. *PLoS Negl Trop Dis* *13*, e0006901.

Tian, H., Yu, P., Cazelles, B., Xu, L., Tan, H., Yang, J., Huang, S., Xu, B., Cai, J., Ma, C., *et al.* (2017). Interannual cycles of Hantaan virus outbreaks at the human-animal interface in Central China are controlled by temperature and rainfall. *Proc Natl Acad Sci U S A* *114*, 8041-8046.

Tian, H.Y., Yu, P.B., Luis, A.D., Bi, P., Cazelles, B., Laine, M., Huang, S.Q., Ma, C.F., Zhou, S., Wei, J., *et al.* (2015). Changes in rodent abundance and weather conditions potentially drive hemorrhagic fever with renal syndrome outbreaks in Xi'an, China, 2005-2012. *PLoS Negl Trop Dis* *9*, e0003530.

Wang, K., Ma, H., Liu, H., Ye, W., Li, Z., Cheng, L., Zhang, L., Lei, Y., Shen, L., and Zhang, F. (2019). The Glycoprotein and Nucleocapsid Protein of Hantaviruses Manipulate Autophagy Flux to Restrain Host Innate Immune Responses. *Cell Rep* *27*, 2075-2091 e2075.

Xiao, H., Tong, X., Huang, R., Gao, L., Hu, S., Li, Y., Gao, H., Zheng, P., Yang, H., Huang, Z.Y.X., *et al.* (2018). Landscape and rodent community composition are associated with risk of hemorrhagic fever with renal syndrome in two cities in China, 2006-2013. *BMC Infect Dis* *18*, 37.

Yang, Y., Li, M., Ma, Y., Ye, W., Si, Y., Zheng, X., Liu, H., Cheng, L., Zhang, L., Zhang, H., *et al.* (2022). LncRNA NEAT1 Potentiates SREBP2 Activity to Promote Inflammatory Macrophage Activation and Limit Hantaan Virus Propagation. *Front Microbiol* *13*, 849020.

Ye, W., Yao, M., Dong, Y., Ye, C., Wang, D., Liu, H., Ma, H., Zhang, H., Qi, L., Yang, Y., *et al.* (2020). Remdesivir (GS-5734) Impedes Enterovirus Replication Through Viral RNA Synthesis Inhibition. *Front Microbiol* *11*, 1105.

Yi, J., Xu, Z., Zhuang, R., Wang, J., Zhang, Y., Ma, Y., Liu, B., Zhang, Y., Zhang, C., Yan, G., *et al.* (2013). Hantaan virus RNA load in patients having hemorrhagic fever with renal syndrome: correlation with disease severity. *J Infect Dis* *207*, 1457-1461.

Zhang, Y., Zhang, C., Zhuang, R., Ma, Y., Zhang, Y., Yi, J., Yang, A., and Jin, B. (2015). IL-33/ST2 correlates with severity of haemorrhagic fever with renal syndrome and regulates the inflammatory response in Hantaan virus-infected endothelial cells. *PLoS Negl Trop Dis* *9*, e0003514.

REVIEWER COMMENTS

Reviewer #1 (Remarks to the Author):

In this revision the authors have constructively addressed the original critiques and the inclusion of new information/interpretations definitely helps.

However, as a reader I still feel overwhelmed by the information presented. For example, it is laudable that every reciprocal IP is shown in the figures and it seems to be strange to be also critiquing researchers for doing things too well! but it still leaves this reviewer wondering whether some of this information could be moved into supplementary.

There really is more than 1 paper here and I worry that the impact of the work will suffer because of the sheer volume of information.

However, these are mainly stylistic and presentation issues - I think the data is generally robust and presents an interesting story.

Rebuttal Letter-2

Responses to the Reviewer #2

General Questions (from page 1 to 4 in this letter)

Question-1:

1. This manuscript requires extensive English editing for grammar.

Reply-1:

Very thankful for taking your precious time to review our manuscript.

Here, I represent all the authors to express our genuine appreciation for your valuable comments.

We have carefully revised the manuscript according to your constructive suggestions, and have done our best to reply all the questions point by point.

Detailed responses are shown as followings.

We have sent our manuscript to the American Journal of Experts (AJE) for language polishing, and the English grammar and other mistakes have been corrected (verification code 7790-80BD-DED5-B7BE-9019). We hope that the revised manuscript could reach the standard in terms of language.

Question-2:

2. Insufficient methods are provided for multiple experiments.

Reply-2:

More details about the methods, including the virus information and mice experiments, have been added in the revised manuscript, and we are sure now almost all the crucial experiment protocols were concluded (in the method part of the revised paper, from page 22 to 33).

Additionally, partial experiments were explained in the figure legends to make it clear.

Moreover, the specific reagents were all listed in the “The Key Resources Table” (Table S6).

We hope that these modifications could make the experimental design easy to understand.

Question-3:

3. Within each section there are methodological issues with data analysis or presentation and these will be detailed below.

Reply-3:

The methodological issues were handled one by one based on your suggestions, which will be

explained in the following responses.

Question-4:

4. Throughout the entire manuscript there are numerous figures that are just too small for the reader to see, there is no way to assess the data if you can't visualize it.

Reply-4:

We have noticed that too much data in the main figures will make the paper somewhat hard to read, and the figure annotations are too small to recognize.

To tackle this issue, we have rearranged all the figures and figure legends.

The modifications are listed as follows:

(1) The crucial data in the main text have been distilled, and now each figure contains 8 to 10 small pictures.

(2) The figure annotations have been adjusted to a suitable size for reading.

(3) The main results, as well as the supplementary data, have been put in order based on one certain theme, which might be easy to understand the research.

(4) The result part has also been modified to be more concise. Partial data have been deleted or moved to the supplementary figures based on the comments from the reviewers. We hope that all these alterations could make our study easy to read.

Here are the revised themes and related figures.

Results	Themes	Figures
1	M1-mediated cytokine storm promotes HFRS pathogenesis.	Fig. 1; Fig. S1 - S3
2	The p65-related M1 activation and inflammation are controlled in HTNV-infected rodents.	Fig. 2; Fig. S4 and S5
3	The late-phase inactivation of M1 protects rodents from secondary bacterial sepsis.	Fig. 3; Fig. S6 and S7
4	Murine Notch signaling blocks p65 activation and rewires the macrophage phenotype.	Fig. 4; Fig. S8 – S11
5	Human Notch activation pattern differs from rodents and mainly enhances the M1-mediated inflammation.	Fig. 5; Fig. S12
6	Murine-specific lncRNAs downstream of Notch signaling retrain M1 polarization.	Fig. 6; Fig. S13 – S15
7	Lnc-ip65 obstructs M1 polarization by interacting with and inhibiting p65 phosphorylation.	Fig. 7; Fig. S16 and S17
8	Lnc-ip65 deficiency aggravates systemic inflammation and	Fig. 8;

	sensitizes mice to HTNV infection.	Fig. S18 and S20
9	NICD activates NF- κ B signaling by recruiting IKK β and p65, which is blocked by lnc-ip65.	Fig. S21 and S23
10	HTNV NP facilitates Notch-mediated M1 activation and exacerbates disease progression.	Fig. 9

Question-5:

5. It would be super helpful to the reader if you would give just a little more info about how an experiment was done ie in which cell type, model, animal etc as you describe the results. For example: Line 608 if you ended this sentence in "...at various time points in mBMDM" it would save the reader some time searching for it.

Reply-5:

Thanks for your constructive suggestion. To reckon with this issue, all the annotations are updated in the figures, which could clearly and directly show the detecting time points or cell lines. Additionally, related explanations are also optimized in the figure legends or methods part.

Example-1 (see the revised Figure 2G)

Example-2 (see the revised Figure 7A-7C)

Example-3 (see the revised Figure S11A-S11E)

Question-6:

6. Passage history of the HTNV used in this study should be noted.

Reply-6:

The passage history of HTNV and other viruses is explained in the revised manuscript.

“HTNV strain 76-118 (24th generation), as well as DENV2 (31th generation), SeV (12th generation), VSV (17th generation), EV71 (21th generation) and HSV-2 (9th generation), was propagated in mice brain and then in Vero E6 cells as we previously described” (see the revised text, Methods- “*Viral Infection*”, line 828-835).

(Human data section) Question-1:

1. The gating strategy shown in S1A has a log scale for what is reported as SSC-H this seems unusual as SSC and FSC are typically on a linear axis and indeed the authors have linear scales for SSC and FSC in the other subparts of S1. This needs to be checked and corrected.

(Human data section) Response-1:

We are extremely grateful to you for pointing out this problem, and this mistake has been corrected in the revised article (Figure S1A) as the following shown. And we have checked the raw data, and found the “log” scale annotation in the original pictures was caused by the default analysis of the FlowJo V10 when it read the data from the BD FACSCalibur Machine. The SSC-H scale has been adjusted with the linear instead of the log unit.

Revised Figure S1A

(Human data section) Question-2:

2. There is nothing upstream of S1 flow plots to explain the gating strategy used. This needs to be shown.

(Human data section) Response-2:

Thanks very much for your comment. We are not sure that whether we get the point, for that generally the SSC (side scatter measures the subcellular components) and FSC (forward scatter measures the cell size) analysis were the most initial process to select the target cells. In terms of the gating strategy for M1- or M2-like monocytes, the sketch map was shown on the left of the flow cytometry results (Figure S1A) as shown in the following.

(Human data section) Question-3:

S1C why are total lymphocytes selected for Th2 and Th17 gating but CD4+ are selected for Treg and Th1 gating? Typically, total CD4's are selected then they are plotted on the subsequent plots to define subtypes.

(Human data section) Response-3:

Thank you for the suggestion. The gating strategy for Th17 has been corrected. All Th subsets (Th17 included) have been enumerated from gated CD4+ T cells (see revised Figure S1D), and correspondingly, related flow cytometry data of Th17 are re-analyzed (see revised Figure S1E and S3E).

(1) Revised Figure S1D:

(2) Revised Figure S1E:

(3) Revised Figure S3E:

(Human data section) Question-4:

4. An explanation needs to be provided for why monocytes are defined differently for each analysis- in one CD14/CD16 are used, in another CD11b/11c++ and in a third only CD11b+.

(Human data section) Response-4:

This is a valuable question to discuss that we did not explain in the original article.

Complement receptors CR3 (CD11b/CD18) and CR4 (CD11c/CD18) belong to the family of beta2 integrins and are expressed mainly in human monocytes ¹. CD11b+ is relatively sufficient to target the monocytes in PBMCs according to current research. However, it should be noted the expression of CD11b on monocytes could be altered under viral infection conditions ². To exclude such influences, CD11b+ and CD11c+ are mostly used to identify monocytes in our study. CD14 and CD16 are used to directly analyze the M1- and M2-like monocytes from the PBMCs as previously reported ³. This gating strategy was emphasized in the revised paper (see the revised Figure S1A and S1B).

(Human data section) Question-5:

5. What is the reasoning behind comparing HNTV patients to JEV, HBV and HCV? JEV is at least an acute RNA viral infection so might be relevant, but HBV and HCV are chronic DNA infections so they seem inconsequential. Also, there is no comment on the timing of sample acquisition with respect to symptom onset for any of these viral infections. Since we know immune responses change over time during any infection this is quite pertinent.

(Human data section) Response-5:

(1) As for the comparison of HTNV infection with other viruses:

Firstly, in terms of the infection features, JEV was used as a control for acute and HBV or HCV was used for chronic infection.

Secondly, in terms of the role of monocytes on the pathogenesis process post infection, previous studies have shown that M1-like (marked by CD14⁺⁺ CD16⁺) or M2-like (marked by CD14⁺ CD16⁺⁺) monocytes are the major immunological determinants for life-threatening influenza or chronic viral hepatitis, respectively ^{3,4}, but whether monocyte activation patterns affect HFRS pathogenesis is unclear. To narrow this gap, peripheral blood mononuclear cells (PBMCs) from patients with distinct viral infections were collected, and the monocyte subset was examined and analyzed.

Thirdly, there also existed the objective limitation, that is only these patients' samples were accessible for our research.

(2) As for the timing of sample acquisition:

In Figure 1A and S1A, HFRS patients, the febrile phase, n=12; Japanese encephalitis patients, the febrile phase, n=15; hepatitis B patients, confirmed chronic infection for more than one-year, inactive phase without liver cirrhosis and antiviral therapy, n=18; hepatitis C patients, confirmed chronic infection for more than one-year, inactive phase without liver cirrhosis and antiviral therapy, n=18. This has been explained in the revised manuscript (see the related figure legends).

(Human data section) Question-6:

6. Define time post onset for each cohort (Febrile, Hypotensive, Oliguric, Diuretic, conval) shown in Figure 1. See comment above about the importance of timing.

(Human data section) Response-6:

For the infection stage of HTNV, we have defined the time post six onsets of HFRS for each cohort in the revised text, and to make this clear, a schematic diagram was added to Figure 1

(Figure 1B) as shown in the following. In general, samples were collected at 3–6 days for the febrile or hypotension stage, 7–12 days for the oliguric stage, 13–18 days for the diuretic stage and after 18 days for the convalescent stage. The phase within 8 days from the fever onset to the early oliguric stage was typically defined as the acute or early phase of the disease. Details for classification of disease severity and exclusion criteria were previously described ^{5, 6}. Additionally, to control for potential confounders, we excluded HFRS patients with autoimmune diseases, viral hepatitis, hematological diseases, diabetes, cardiovascular diseases, and other kidney or liver diseases. This has been explained in the Method parts of the revised manuscript (“Study Participants”, line 734 to 752).

Revised Figure 1B

(Human data section) Question-7:

7. There is no description of how human blood was processed for the various flow cytometric assays. Typically, secretion inhibitors are used to capture ICS events, but this isn’t mentioned. Were the cells processed and stained immediately or rested in vitro?

(Human data section) Response-7:

Thank you for this technique question. The monocytes or T cells from HFRS patients underwent intracellular cytokine staining immediately, while for the *in vitro* experiments, the secretion inhibitors are used to precisely measure cytokine production. This has been explained in the revised manuscript (see the revised methods parts, line 753 to 761 and line 932 to 949).

(1) (the revised paper, line 753 to 761)

“The clinical blood samples of healthy individuals between the ages of 20 and 35 years were obtained from the Blood Transfusion Department of Tangdu Hospital (n=55) in agreement with institutional ethics regulations. To obtain human **monocyte-derived macrophages (hMDM)**, peripheral blood mononuclear cells (PBMCs) were first enriched by Ficoll (TBDscience) from peripheral blood density

gradient centrifugation. Then, human monocytes were magnetically purified from PBMCs with negative selection beads (EasySep™ Human Monocyte Isolation Kit, StemCell). Finally, monocytes were primed with recombinant human macrophage colony-stimulating factor (M-CSF) (15 ng/ml, PeproTech) with medium exchange every other day for a week to generate hMDM. Alternatively, the PBMCs were laid into Petri dishes for 4 hr, and the supernatant cells were collected to acquire the **monocyte-removed PBMCs.**”

(2) (the revised paper, line 932 to 949)

“Flow Cytometry (FCM)

The monocytes or T cells from HFRS patients underwent intracellular cytokine staining immediately, while for the *in vitro* experiments, the secretion inhibitors are used to precisely measure cytokine production. Generally, FcγII/III receptors of monocytes and macrophages were blocked with anti-CD16/32 antibody (BD Bioscience) before staining the cell surface markers, and brilliant stain buffer (BD Bioscience) was applied before staining intracellular cytokines. For the TFs (FoxP3, RORγt, GATA3 and T-bet) in T cells from healthy or patient PBMCs, the BD Pharmingen™ Transcription Factor Buffer Set was applied. The cells were manipulated in FCM buffer during the flow cytometry assays, which included PBS containing 2% FBS (Gibco) and 2 mM EDTA. For *in vitro* assays, cells were enzymatically detached with Trypsin-EDTA solution (Solarbio) and subsequently washed and processed with FCM buffer. For the flow cytometry detection of macrophages in spleens, the single-cell suspension of the spleen tissue was acquired through gentle grinding and filtration with a 70 μm cell strainer, and the erythrocytes were lysed with RBC lysis buffer (Gibco). The main procedure was as follows: Cell acquisition→ Fc receptor block→ Surface marker staining→ Permeabilization and fixation→ brilliant stain buffer treatment→ intracellular iNOS or cytokine staining→ Compensation adjustment with beads→ Samples were analyzed with a BD FACSCalibur™ 3-laser flow cytometer or BD FACSCanto 10-laser flow cytometer. Finally, the data were processed with FlowJo v10 (TreeStar). Respective antibodies are shown in the Key Resources Table (Table S6).”

(Human data section) Question-8:

8. It is not clear how the cytokine data presented in Figure 1G were analyzed to cohort them into red/green/blue/black. What threshold was used to put a given cytokine in each category?

(Human data section) Response-8:

Our deepest gratitude goes to you for your careful work and thoughtful suggestions that have helped improve this paper substantially. Indeed, the heatmap display and the statistical analysis were performed severally, which means this picture contains two kinds of information.

More details about this assay are shown in the **Methods-** “*Bio-Plex Multiplex Immunoassay*” (see the revised paper, line 950-960).

(Human data section) Question-9:

9. Line 251 “various clinical symptoms experienced by patients” are attributed to TNF and IL-8 levels- what clinical symptoms? Were they quantitated in some way to determine if they correlated with cytokine levels? Along these same lines, what criteria were used to consider patients as “severe/critical” vs mild/mod”

(Human data section) Response-9:

Very thankful for your question. Indeed, it was speculated that “various clinical symptoms experienced by patients” are attributed to TNF and IL-8 levels according to previous researches 7, 8, 9. Related descriptions have been deleted in the revised manuscript in case of being misunderstood.

As for the criteria, mild HFRS was defined as mild renal failure without an obvious oliguric stage and moderate for obvious symptoms of uremia, effusion (bulbar conjunctiva), hemorrhage (skin and mucous membrane), and renal failure with a typical oliguric stage. While patients with severe uremia, effusion (bulbar conjunctiva and either peritoneum or pleura), hemorrhage (skin and mucous membrane), and renal failure with oliguria (urine output, 50–500 mL/day) for ≤ 5 days or anuria (urine output, < 50 mL/day) for ≤ 2 days were defined as severe HFRS, and critical patients were considered as those with ≥ 1 of the following symptoms: refractory shock, visceral hemorrhage, heart failure, pulmonary edema, brain edema, severe secondary infection, and severe renal failure with either oliguria (urine output, 50–500 mL/day) for > 5 days, anuria (urine output, < 50 mL/day) for > 2 days, or a blood urea nitrogen level of > 42.84 mmol/L. Details for classification of disease severity and exclusion criteria were previously described 5, 6.

This has been explained in the revised manuscript (line 746 to 747).

(Human data section) Question-10:

10. It is mentioned that the lack of detection of IFN γ in HFRS patient is evidence of lack of Th1 activity- this should be considered carefully. MIG (cytokine induced by IFN γ levels), CXCL-10 and CXCL-11 were elevated in HFRS patients with severe disease, and they are all markers of IFN γ activity. Given the limited sampling of humans, the IFN peak could have

been missed or occurred at a tissue level and not in the bloodstream. All humans studies are limited by the fact that only blood is routinely sampled.

(Human data section) Response-10:

Very thankful for your question. Related descriptions about that lack of detection of IFN γ in HFRS patient is evidence of lack of Th1 activity were deleted in the revised manuscript. Indeed, this might be caused by the limitation of the sample number as you mentioned.

(Human data section) Question-11:

A complicated PMBC/HUVEC/293 co-culture system is established and not well explained. Why are 293 cells even needed in this system? Can't you measure the resistance generated by the HUVEC layer in the transwell without having cells below? It also seems that what is being measured is HTNV infection in PMBCs with or without monos and you don't need transwell for this. What is meant by red/green/blue in this context in S2? It is compared to the human data in Fig 1 but red/green/blue had definitions related to disease severity which is clearly not possible in this in vitro co-culture system.

(Human data section) Response-11:

(1) To confirm the role of monocytes/macrophages in HFRS pathogenesis, a cell coculture system was established as we previously reported (*Yang Y, et al. LncRNA NEAT1 Potentiates SREBP2 Activity to Promote Inflammatory Macrophage Activation and Limit Hantaan Virus Propagation. Front Microbiol 13, 849020 (2022).*), in which the PBMC and HUVEC were laid on the upper layer to mimic the endovascular environment, and HEK293 laid on the bottom layer to mimic the tissue immune responses against HTNV infection (see the revised paper from line 133 to 136 and Figure S3A-i).

This coculture system was applied for the crosstalk between HUVECs and HEK293 cells might mimic the *in vivo* infection environment of the kidney tissue. In fact, the TER of HUVECs could be detected based on the transwell system without HEK293, but this could not reflect the real infection situation as the cytokines secreted by tissue cells also matter a lot in activating immune responses, which might affect the vascular permeability post HTNV infection.

(2) The 'color' scheme statements have been replaced by ".....the production of most inflammatory cytokines was remarkably inhibited once the monocytes were removed (Figure S3D)" in the revised main text (line 139-140) and "Heatmap depicting 40 cytokines in the supernatants at 48 hpi. Upregulated

cytokines were marked red” in the revised figure legend.

(Human data section) Question-12:

12. Line 298- “negative screening technology” used to purify monocytes- I think you mean negative selection, correct?

(Human data section) Response-12:

Very thankful for your kind help in correcting the description. Yes, it means the “negative selection”, and this has been modified in the revised manuscript (line 143 to 144). Other parts of this article have been carefully checked, and related description has also been adjusted (e.g., “negative screening beads” was rectified as “negative selection beads” in line 757).

(1) The revised manuscript (line 143 to 144):

“To exclude the effects of other immune cells in PBMCs, monocytes were exclusively collected through negative selection.”

(2) The revised manuscript (line 757 to 758):

“Then, human monocytes were magnetically purified from PBMCs with negative selection beads (EasySep™ Human Monocyte Isolation Kit, StemCell).”

(Human data section) Question-13:

13. Correlation does not equal causation. This error is made repeatedly with respect to discussion of human clinical data- which is only able to show correlation/association.

(Human data section) Response-13:

Very thankful for your kind help in pointing out this error. In the revised paper, all the related descriptions have been rectified with the words as “suggesting that they might lead to ……” and “These data indicated that excessive inflammation in HFRS might be incriminated with the dysregulated monocyte patterns” (see the revised paper, line 128-132). Neutral narration is more rigorous.

Specific Questions (Natural mouse reservoir studies, from page 14 to 18 in this letter)

(Natural mouse reservoir studies) Question-1:

1. I cannot find a description in the methods regarding collection, manipulation or processing of field mice. Was there institutional or community level approval for this?

(Natural mouse reservoir studies) Response-1:

Very thankful for the reminder. The epidemiological data of the field mice were collected from

the Shaanxi Provincial Notifiable Disease Surveillance System in collaboration with the research team of Pengbo Yu, which was authorized by the government ¹⁰. For the capture, traps were placed outdoors (set as 4 parallel lines of 25 traps each and spaced intervals of 5 meters). Rodents were removed from the traps once captured for further investigation. According the standard measurement for HTNV infection in rodents ^{11,12,13}, the lung tissues were acquired for analysis. This has been explained in the revised manuscript (Method- “*Animal Models*”, line 762 to 768). And we thanked the research team of Pengbo Yu in the ACKNOWLEDGMENTS.

(Natural mouse reservoir studies) Question-2:

2. Fig 2A is too small to be read.

(Natural mouse reservoir studies) Response-2:

According to the editorial comments, this map has been removed and the related data are exhibited in Table S1.

Table S1. The incidence and mortality rate of HFRS in China, Related to Figure 2 (shown as a word file in the re-submitted compressed file entitled “Sup Tables.rar”).

(Natural mouse reservoir studies) Question-3:

3. Where do the data from Figure 2Ai come from? No reference is provided for this prevalence data.

(Natural mouse reservoir studies) Response-3:

Very thankful for the reminder. Original Figure 2A-i showed Geographic heatmap depicting HFRS incidence in China from 2016 to 2017, and the epidemiological data of HFRS in China were acquired from the Chinese Center for Disease Control and Prevention (CDCDC) (Available at: <http://www.stats.gov.cn/>).

According to the editorial comments, this map has been removed and the related data are exhibited in Table S1 shown as follows.

Table S1. The incidence and mortality rate of HFRS in China, Related to Figure 2

Regions	2016				2017			
	Reported Cases	Incidence Rate	Death Toll	Mortality Rate	Reported Cases	Incidence Rate	Death Toll	Mortality Rate
*Shaanxi	933	2.4599	1	0.0026	2064	5.4129	14	0.0367
Heilongjiang	1200	3.148	2	0.0052	1119	2.9455	2	0.0053
Liaoning	863	1.9692	3	0.0068	1078	2.4624	3	0.0069
Jilin	515	1.8705	2	0.0073	574	2.1003	3	0.011
Shandong	985	1.0003	12	0.0122	1252	1.2587	10	0.0101
Jiangxi	676	1.4806	5	0.011	571	1.2434	1	0.0022
Fujian	366	0.9534	0	-	388	1.0015	1	0.0026
Hunan	588	0.8678	3	0.0044	632	0.9264	4	0.0059
Hubei	236	0.4033	5	0.0085	505	0.8581	6	0.0102
Zhejiang	345	0.6229	0	-	351	0.6279	1	0.0018
Hebei	434	0.5845	1	0.0013	447	0.5983	0	-
Yunnan	216	0.4555	0	-	229	0.48	0	-
Jiangsu	330	0.4137	8	0.01	375	0.4688	3	0.0038
Henan	176	0.1857	1	0.0011	435	0.4563	11	0.0115
Anhui	207	0.3369	0	-	273	0.4406	2	0.0032
Guangdong	410	0.3779	3	0.0028	443	0.4028	0	-
Inner Mongolia	106	0.4221	1	0.004	93	0.369	0	-
Sichuan	110	0.1341	1	0.0012	260	0.3147	1	0.0012
Tianjin	31	0.2004	0	-	32	0.2048	0	-
Guizhou	59	0.1672	0	-	58	0.1632	0	-
Gansu	19	0.0731	0	-	20	0.0766	0	-
Shanxi	8	0.0218	0	-	26	0.0706	2	0.0054
Hainan	3	0.0329	0	-	3	0.0327	0	-
Beijing	9	0.0415	0	-	7	0.0322	0	-
Tibet	1	0.0309	0	-	1	0.0302	0	-
Ningxia	4	0.0599	0	-	2	0.0296	0	-
Guangxi	5	0.0104	0	-	14	0.0289	0	-
Chongqing	13	0.0431	0	-	5	0.0164	0	-
Xinjiang	1	0.0042	0	-	3	0.0125	0	-
Shanghai	4	0.0166	0	-	2	0.0083	0	-
Taiwan	4	0.0056	0	-	0	-	0	-
Total	8853	0.6458	48	0.0035	11262	0.8162	64	0.0046

* *A. agrarius* mice were captured in the field of Hu County in the Weihe Plain (106–110°E, 34–36°N, Shaanxi Province, in which there was the highest incidence rate of HFRS in China (2017). The epidemiological data of HFRS in China were acquired from the Chinese Center for Disease Control and Prevention (CDCDC) (Available at: <http://www.stats.gov.cn/>).

(Natural mouse reservoir studies) Question-4:

4. Why was anti-HTNV antibody measured from the lungs and not the blood of mice? What

samples were collected from these animals? (see point 1 in this section)

(Natural mouse reservoir studies) Response-4:

Very thankful for your question.

On the one hand, as we mentioned above, according the standard measurement for HTNV infection in rodents ^{11, 12, 13}, the lung tissues were acquired for analysis.

On the other hand, we found that *A. agrarius* mouse lung tissue was more susceptible to HTNV infection than liver or kidney (Figure S4A), which was consistent with previous studies ^{14, 15} and ensured the authenticity of our testing system.

All these alterations have been updated in the revised manuscript in Result 2.

(Natural mouse reservoir studies) Question-5:

5. Figure S3A- IHC images cannot be visualized at current magnification, a zoomed in inset of each should be included.

(Natural mouse reservoir studies) Response-5:

Very thankful for your suggestion.

For the figures that are ambiguous, the original complex figures have been disassembled into different parts to make every immunofluorescent, flow cytometry or statistical picture clearly presented (main results with Figure 1 to 9, and supplementary materials with Figure S1 to S24). The figures are re-organized and re-labelled based on the logic requirements of the reviewers and editors.

This figure has been adjusted in the revised paper (Figure S4A), which means a zoomed in inset of each is included as the following shown.

(Natural mouse reservoir studies) Question-6:

6. For studies in field mice, what assay was used to measure HTNV S- IHC? Antigen capture? qRT-PCR?

(Natural mouse reservoir studies) Response-6:

Very thankful for your question. Natural infection phases of HTNV in *A. agrarius* mice defined based on the assessment of HTNV-S (qRT-PCR) and anti-NP IgG (ELISA) in mouse lung tissue. This has been explained as follows post revision.

“Based on the viral RNA level and host antibody production, the rodent disease phases were classified as HTNV infection negative stage (HINS), early stage (HIES), progressive stage (HIPS), and clearance stage (HICS) (Figure 2A)” (see the revised paper, line 153 to 155).

(Natural mouse reservoir studies) Question-7:

7. Figure 2B-please state in text what sample you are measuring cytokines in.

(Natural mouse reservoir studies) Response-7:

Very thankful for your question. The samples were the lung tissues as we explained above, and this was emphasized in the revised manuscript:

“Cytokine production measured by ELISA in lung tissues” (the revised figure legends, line 596);

(Natural mouse reservoir studies) Question-8:

8. Nothing can be seen in Figure 2B. This data cannot be interpreted in print. I had to zoom in to 300% on the PDF to see this.

(Natural mouse reservoir studies) Response-8:

Very thankful for your advice.

The original complex figures have been disassembled into different parts to make every immunofluorescent, flow cytometry or statistical picture clearly presented (main results with Figure 1 to 9, and supplementary materials with Figure S1 to S24).

This picture has been zoomed in for clear presentation in the revised article (Figure 2B). We hope this could settle your question.

Other Specific Questions-1 (rely from page 18 to 23 in this letter)

Laboratory mouse model studies showing in vitro p65 differences between human and mouse and late in vivo anti-inflammatory state in mice.

(Other Specific Questions-1) Question-1:

1. Please state in the text what species of mice are being used in these studies. The methods say

all mice are BL/6 (clearly not since you use field mice too) the legend for S3 says “WT“ or RIG-/- (the latter are not mentioned at all in the methods nor are the various mice used later in the paper).

Response-1: Thanks for providing the precious suggestion. Except for the *A. agrarius* captured from the field (also termed as the field mice), the strain of mice in this study, including the wild type or transgenic mice, is BL/6J. RIG-I deficient (RIG-I^{-/-}) C57BL/6J mice were constructed and identified from the Experimental Animal Center of Air Force Medical University. These have been emphasized in the revised manuscript (the Methods- “*Animal Models*”, line 762 to 779; also listed in the Table S6, the Key Resources Table in This Study).

(Other Specific Questions-1) Question-2:

2. Figure 2E/F- how do HTNV viral loads compare between these groups?

Response-2: Thanks for providing the valuable advice. We have accomplished the experiments to compare the viral loads between different group. The following figure showed the absolute quantitative PCR standard curve for quantify the copies of HTNV S segments, the process of which was previously reported ⁵.

(Other Specific Questions-1) Question-3:

3. S3 E seems to belong on S4 given that it is the same line of experimentation with RAW/THP cells.

Response-3: Thanks for providing the valuable advice. S3E did belong the S4, and these figures were orderly organized in our revised manuscript (original S3E---revised Figure S5A-i, original S4---revised Figure S5).

(Other Specific Questions-1) Question-4:

4. Figure S4D also not easy to see, needs zoomed in insets.

Response-4: Thanks for your valuable suggestion. We have rearranged all the figures and figure legends, and this picture has been enlarged in the revised manuscript for clear presentation. We hope this could settle your question. Here are the revised picture of Figure S5F (that was the original S4D).

(Other Specific Questions-1) Question-5:

5. Figure 3 some fonts are so small as to not be legible eg 3B,D legends

Response-5: Thanks for your kind help.

We have rearranged all the figures and figure legends, and this picture has been enlarged in the revised manuscript for clear presentation. We hope this could settle your question.

The original Figure 3B and 3D contents have been moved to the revised Figure S6B and S6C, shown in the following in a clearer pattern.

- Revised Figure S6
-

(Other Specific Questions-1) Question-6:

6. Figure 3J- can't appreciate this pathology-too small.

Response-6: Thanks for your valuable advice.

The original complex figures have been disassembled into different parts to make every immunofluorescent, flow cytometry or statistical picture clearly presented (main results with Figure 1 to 9, and supplementary materials with Figure S1 to S24). The figures are re-organized and re-labelled based on the logic requirements of the reviewers and editors.

We hope this could settle your question. Here are the revised picture of Figure 3G (that was

original Figure 3J).

- Revised **Figure 3G**

(Other Specific Questions-1) Question-7:

7. Regarding figure sizes- Figure S5B is how all of the path should be shown- total tissue with zoom in to see the details.

Response-7: Thanks for your valuable advice.

Based the model shown as the original Figure S5B, we have rearranged all the figures and figure legends, and this picture has been enlarged in the revised manuscript for clear presentation. We hope this could settle your question.

The original Figure S5B was adjusted to be clear and compact as shown in the following, which was titled as **Figure S7B** in the revised article.

- Revised **Figure S7B**

Other Specific Questions-2 (Notch pathway studies, from page 23 to 29 in this letter)

(Other Specific Questions-2, Notch pathway studies) Question-1:

1. Again, can't see Figure 4D or E without 300% magnification.

Response-1: Thanks for your valuable advice.

Figure 4D and 4E were adjusted in the revised Figure S9D and S10A, respectively, which were shown in the followings. Additionally, we have rearranged all the figures and figure legends, and these pictures have been enlarged in the revised manuscript for clear presentation. We hope this could settle your question.

• Revised **Figure S9D**

• Revised **Figure S10A**

(Other Specific Questions-2, Notch pathway studies) **Question-2:**

2. Figure 4 J and K are pointless if you can't magnify the image sufficiently for the reader to

see it.

Response-2: Thanks for your valuable advice.

Related figures that represent the macrophage functions, including Figure 4J and K, were independently displayed as Figure S11 in the revised paper, which were shown in the followings. Additionally, we have rearranged all the figures and figure legends, and these pictures have been enlarged in the revised manuscript for clear presentation. We hope this could settle your question.

(Other Specific Questions-2, Notch pathway studies) Question-3:

3. The most important piece of the RBP-J^{cko} mouse data (the survival curves) and viral loads are buried in S7. Please check doses of HTNV noted in this legend. I can't read what is on the figure, but the legend has the same thing twice for Fii and Fiii. Also please show viral loads for adult mouse models. This is just as important as showing it for the neonatal model, are the adult mice that succumb doing so with high viral loads or not?

Response-3: Thanks for your constructive suggestion. Three points were listed:

(1) The RBP-J^{cko} mouse data (the survival curves) were organized to Figure 4E and 4F in the revised paper, and other results were rearranged in Figure S11;

(2) The HTNV challenge doses were different, which was emphasized and corrected in the revised paper.

- The revised result part, line 275 to 282:

“To evaluate whether this process was beneficial *in vivo*, neonatal and adult mouse models were utilized. RBP-J^{CKO} suckling mice showed an early onset of disease than the WT mice (Figure 4E), which was associated with more severe inflammatory responses but not viral loads (Figure S11H). Intriguingly, RBP-J^{CKO} adult mice were susceptible to high-dose HTNV challenge (Figure 4F and 4G), and they also showed aggravated TNF α responses and uncontrolled hantaviral replication at 7 dpi (Figure S11I). Additionally, worsen pathological changes in the RBP-J^{CKO} spleens at 7 dpi, which were accompanied by hyperactivation of p65 (Figure 4H and 4I).”

- The revised figure legends, line 630 to 631:

“(E) Survival data for HTNV-challenged 4-day neonatal mice (i.p., 8×10^5 TCID₅₀/g (i). (F) Survival data for HTNV-challenged 8-week-old adult mice doses (i.p., 8×10^5 to 8×10^7 TCID₅₀/g).”

(3) Detection of serum viral loads for adult mouse models had been performed, and we found that a significantly increased hantaviral replication at 7 dpi (Figure S11I-ii) in the RBP-J^{CKO}

group, as shown in the following. This indicated that the dysfunction of macrophages in RBP-J^{CKO} mice could contribute to the aggravated immune damages as well as delayed clearance of viruses. These data were added to the revised manuscript.

(Other Specific Questions-2, Notch pathway studies) Question-4:

4. Again, there is no way a reader can see anything in figure S7Giii. Those spleens are mere spots when printed on a page. Eg. figure 4Q is good. I can actually see the spleen architecture.

Response-4: Thanks for your valuable advice.

Figure S7Giii and 4Q were adjusted in the revised Figure 4H and 4I, respectively, which were shown in the followings. Additionally, we have rearranged all the figures and figure legends, and these pictures have been enlarged in the revised manuscript for clear presentation. We hope this could settle your question.

• Revised Figure 4G to 4I

(Other Specific Questions-2, Notch pathway studies) Question-5:

5. Wow, human notch signaling data are also buried in S7. These data are more significant in supporting your overarching hypothesis than everything in figure 4J-R.

Response-5: Thanks for your valuable advice.

The human notch signaling data are very important, and we have rearranged these results in Figure 5 and S12 (shown as the followings) in the revised paper.

Moreover, we emphasize this part with an independent result part as follows.

• Revised Result (the revised main text, line 286-299)

Notch Activation Pattern Differs in Human Macrophages and Promotes M1-mediated Inflammation

Another noteworthy question was whether there existed a discrepant Notch activation pattern in human

macrophages. Most Notch-related genes and proteins increased post HTNV infection in hMDM from 0 to 36 hpi (Figure 5A and 5B), during which the NICD was continuously generated and translocated into the nucleus (Figure 5C and 5D). These results indicated that the Notch pathway was completely activated in human macrophages throughout the infection stage. Hindering NICD generation with DAPT could constrain the phosphorylation of p65 rather than p-JNK or p-ERK, which would also facilitate HTNV replication at the late infection stage (36 to 48 hpi) (Figure 5E), suggesting that Notch signaling might consolidate the human M1 polarization process. Consistently, DAPT restrained the secretion of various proinflammatory cytokines at 48 hpi (Figure 5F), during which the expression of manifold M1-related genes was downregulated while M2-related genes were strengthened (Figure 5G and 5H). Furthermore, we found that the activation level of Notch signaling in monocytes was associated with disease severity (Figure S12). These data collectively demonstrated that Notch signaling showed a distinct activation pattern in humans versus mice, which promoted M1-mediated cytokine storm in HFRS patients.

(Other Specific Questions-2, Notch pathway studies) Question-6:

6. I'm not sure what I'm looking at for Fig 5C. I assume that if there is differential expression between WT and RBPJcko mice at the loci indicated with arrows, we should be able to see a difference? I don't. Maybe b/c it is too small?

Response-6: Thank you for your precious comments and advice.

Considering that the Fig 5C could not clearly display the lncRNA alteration on different chromosomes (tremendous background gene expression would make it hard to recognize the changes of several lncRNAs) and might lead to misconception, we have deleted these data in the revised manuscript.

(Other Specific Questions-2, Notch pathway studies) Question-7:

7. I understand the inclusion of DENV infected mBMDM for figure 5 but a sentence explaining why it was chosen would be a nice intro to the expt. Also, I don't see it noted in the methods about his virus or its use. Likewise, no info on use of other viruses- VZV EV71, SeV, HSV...

Response-7: Thank you for this suggestion.

DENV2 also could infect mice but not induce clinical symptoms, which might share some similar characteristics with HTNV during the infection process.

The information of other viruses was added in the methods part (see the revised text, Methods-*"Viral Infection"*, line 828-835).

(Other Specific Questions-2, Notch pathway studies) Question-7:

8. I've stopped commenting on each of the super small figures but this continued to be a problem throughout.

Response-8: Thank you for sparing your precious time to review our manuscript.

The original complex figures have been disassembled into different parts to make every immunofluorescent, flow cytometry or statistical picture clearly presented (main results with Figure 1 to 9, and supplementary materials with Figure S1 to S24). The figures are re-organized and re-labelled based on the logic requirements of the reviewers and editors.

Reference

1. Sándor N, *et al.* CD11c/CD18 Dominates Adhesion of Human Monocytes, Macrophages and Dendritic Cells over CD11b/CD18. *PLoS One* **11**, e0163120 (2016).
2. Mairpady Shambat S, *et al.* Hyperinflammatory environment drives dysfunctional myeloid cell effector response to bacterial challenge in COVID-19. *PLoS Pathog* **18**, e1010176 (2022).
3. Cole SL, *et al.* M1-like monocytes are a major immunological determinant of severity in previously healthy adults with life-threatening influenza. *JCI Insight* **2**, e91868 (2017).
4. Saha B, Kodys K, Szabo G. Hepatitis C Virus-Induced Monocyte Differentiation Into Polarized M2 Macrophages Promotes Stellate Cell Activation via TGF- β . *Cell Mol Gastroenterol Hepatol* **2**, 302-316 e308 (2016).
5. Yi J, *et al.* Hantaan virus RNA load in patients having hemorrhagic fever with renal syndrome: correlation with disease severity. *J Infect Dis* **207**, 1457-1461 (2013).
6. Zhang Y, *et al.* IL-33/ST2 correlates with severity of haemorrhagic fever with renal syndrome and regulates the inflammatory response in Hantaan virus-infected endothelial cells. *PLoS Negl Trop Dis* **9**, e0003514 (2015).
7. Saksida A, Wraber B, Avšič-Županc T. Serum levels of inflammatory and regulatory cytokines in patients with hemorrhagic fever with renal syndrome. *BMC Infect Dis* **11**, 142 (2011).
8. Li X, *et al.* Expression of CD206 and CD163 on intermediate CD14(++)CD16(+) monocytes are increased in hemorrhagic fever with renal syndrome and are correlated with disease severity. *Virus Res* **253**, 92-102 (2018).
9. Angulo J, *et al.* Serum levels of interleukin-6 are linked to the severity of the disease caused by Andes Virus. *PLoS Negl Trop Dis* **11**, e0005757 (2017).
10. Tian H, *et al.* Interannual cycles of Hantaan virus outbreaks at the human-animal interface in Central China are controlled by temperature and rainfall. *Proc Natl Acad Sci U S A* **114**, 8041-8046 (2017).

11. Xiao H, *et al.* Landscape and rodent community composition are associated with risk of hemorrhagic fever with renal syndrome in two cities in China, 2006-2013. *BMC Infect Dis* **18**, 37 (2018).
12. Tian HY, *et al.* Changes in rodent abundance and weather conditions potentially drive hemorrhagic fever with renal syndrome outbreaks in Xi'an, China, 2005-2012. *PLoS Negl Trop Dis* **9**, e0003530 (2015).
13. Tian H, Stenseth NC. The ecological dynamics of hantavirus diseases: From environmental variability to disease prevention largely based on data from China. *PLoS Negl Trop Dis* **13**, e0006901 (2019).
14. No JS, *et al.* Comparison of targeted next-generation sequencing for whole-genome sequencing of Hantaan orthohantavirus in *Apodemus agrarius* lung tissues. *Sci Rep* **9**, 16631 (2019).
15. Kim JA, *et al.* Genetic Diversity and Reassortment of Hantaan Virus Tripartite RNA Genomes in Nature, the Republic of Korea. *PLoS Negl Trop Dis* **10**, e0004650 (2016).

REVIEWER COMMENTS

Reviewer #4 (Remarks to the Author):

Human data Question 11.

I agree with the previous reviewer that the co-culture system is unnecessarily complicated. While using primary PBMC's, the authors also use HUVEC as a model for the endovascular environment. While HUVEC are endothelial cells, they are isolated from umbilical cord which is not a target organ. In addition, orthohantaviruses primarily target microvascular EC's. To mimic the environment in the kidney, renal microvascular EC should be used. While HEK293 originated from kidney, these are from fetal origin and immortalized, thus not really a representative cell with no data on whether this cell line responds to HTNV similar to infection in human cases.

Natural mouse reservoir question 6:

It is not clear what the division of early, progressive and clearance stage is based on. Please specify how these stages were determined, and what data this is based on.

Rebuttal Letter

Responses to the Reviewer #4

Question-1

Human data Question 11.

I agree with the previous reviewer that the co-culture system is unnecessarily complicated. While using primary PBMC's, the authors also use HUVEC as a model for the endovascular environment. While HUVEC are endothelial cells, they are isolated from umbilical cord which is not a target organ. In addition, orthohantaviruses primarily target microvascular EC's. To mimic the environment in the kidney, renal microvascular EC should be used. While HEK293 originated from kidney, these are from fetal origin and immortalized, thus not really a representative cell with no data on whether this cell line responds to HTNV similar to infection in human cases.

Reply-1

Very thankful for taking your precious time to review our manuscript. Here, I represent all the authors to express our genuine appreciation for your valuable comments.

We agreed that HUVECs are not the ideal cell model to assess the vascular permeability alteration, although previous researches performed related experiments with HUVECs during hantavirus infection (*PLoS Pathog*, 2013; 9(7): e1003470; *J Virol*, 2008 Jun; 82(12): 5797-806.). Hence, the data of co-culture system (original Figure S3), as well as the corresponding description in the result and method parts, have been removed in the revised paper. Additionally, we discussed this issue as a limitation of our research (page 16, from line 526 to 533 in the revised paper; shown in yellow shadows). Based on your precious advice, we are now trying to construct kidney organoids with capillaries, which might be a suitable model to investigate the hantavirus pathogenesis. Of course, this is another intriguing story.

Revision (page 16, from line 526 to 533 in the revised paper):

526 The typical pathology feature of HFRS is extensively increased capillary permeability triggered by
527 cytokine storm and hyperactivation of immune cells, while to date no ideal co-culture systems are available
528 to measure how HTNV aggravates endovascular damage with complicated immune micro-environments⁵.
529 ¹⁰². Although we found that hyperactivation of inflammatory monocyte/macrophage might contribute to
530 the endovascular dysfunction with a transwell co-culture system containing both primary human umbilical
531 vein endothelial cells and monocytes (data not shown) as we previously established¹⁰³, further experiments
532 with renal microvascular endovascular cells to mimic the environment in the kidney should be developed
533 to verify the crucial role of MPS in HTNV-induced endovascular dysfunction.

Question-2

Natural mouse reservoir question 6:

It is not clear what the division of early, progressive and clearance stage is based on. Please specify how these stages were determined, and what data this is based on.

Reply-2

Thank you for your question. To better illuminate the natural infection process of hantaviruses in field mice, the disease phases were classified as HINS (HTNV S/viral RNA: negative, anti-NP IgG/antibodies against viral protein: negative), HIES (HTNV S: positive, anti-NP IgG: negative), HIPS (HTNV S: positive, anti-NP IgG: positive), and HICS (HTNV S: negative, anti-NP IgG: positive) according to the assessment results of viral RNA and host anti-hantaviral antibody in lungs (Figure 2A). The lung tissues of captured A. agrarius mice were acquired for viral RNA or anti-NP IgG detection by qRT-PCR or ELISA, respectively (details were shown in the following). The viral RNA assessment was evaluated as positive if the Cq (quantification cycle) value of HTNV S segments was lower than 30, as we described previously. The anti-NP IgG assessment was evaluated as positive if the P/N (positive versus negative) value was higher than 2.1, as we described previously. This has been clearly explained in the revised paper (page 23, from line 754 to 763).

Revision (page 23, from line 754 to 763):

754 included for analysis. **To better illuminate the natural infection process of hantaviruses in field mice, the**
755 **disease phases were classified as HINS (HTNV S/viral RNA: negative, anti-NP IgG/antibodies against viral**
756 **protein: negative), HIES (HTNV S: positive, anti-NP IgG: negative), HIPS (HTNV S: positive, anti-NP**
757 **IgG: positive), and HICS (HTNV S: negative, anti-NP IgG: positive) according to the assessment results of**
758 **viral RNA and host anti-hantaviral antibody in lungs (Figure 2A). The lung tissues of captured A. agrarius**
759 **mice were acquired for viral RNA or anti-NP IgG detection by qRT-PCR or ELISA, respectively (details**
760 **were shown in the following). The viral RNA assessment was evaluated as positive if the Cq (quantification**
761 **cycle) value of HTNV S segments was lower than 30, as we described previously¹⁰⁴. The anti-NP IgG**
762 **assessment was evaluated as positive if the P/N (positive versus negative) value was higher than 2.1, as we**
763 **described previously¹⁰⁸.**

Here are partial raw data for HTNV RNA or antibodies detection captured A. agrarius mice:

Table with columns: 之前编号, 检测编号, 居民区/野外, 原始编号, 是否送公司染色, 样本 (鼠肺, 鼠血, 鼠胃, 鼠肝, 抗体小鼠), 检测结果 (HIV核酸), and various assay results (TNFa, IL-1β, IFNα, MCP1, IP-10, IL-10, TNFα, IL-1β, IFNα, MCP1, IP-10, pg/ml).

REVIEWERS' COMMENTS

Reviewer #4 (Remarks to the Author):

The authors have sufficiently addressed my comments.

A point-by-point response to the reviewers' comments

REVIEWERS' COMMENTS

Reviewer #4 (Remarks to the Author):

The authors have sufficiently addressed my comments.

Reply:

Thank you for your kind help.